# NEWTONBENCH: BENCHMARKING GENERALIZABLE SCIENTIFIC LAW DISCOVERY IN LLM AGENTS

**Tianshi Zheng**[*1]**, Kelvin Kiu-Wai Tam**[*1]**, Newt Hue-Nam K. Nguyen**[*1]
**Baixuan Xu**[1]**, Zhaowei Wang**[1]**, Jiayang Cheng**[1]**, Hong Ting Tsang**[1]**, Weiqi Wang**[1]
**Jiaxin Bai**[1]**, Tianqing Fang, Yangqiu Song**[†1]**, Ginny Y. Wong**[2]**, Simon See**[2]
[1]The Hong Kong University of Science and Technology, [2]NVIDIA
{tzhengad, kwtamai, khnnguyen}@connect.ust.hk, yqsong@cse.ust.hk
{gwong, ssee}@nvidia.com
**Code and Data:** https://github.com/HKUST-KnowComp/NewtonBench

## ABSTRACT

Large language models (LLMs) are emerging as powerful tools for scientific law discovery, a foundational challenge in AI-driven science. However, existing benchmarks for this task suffer from a fundamental methodological trilemma, forcing a trade-off between scientific relevance, scalability, and resistance to memorization. Furthermore, they oversimplify discovery as static function fitting, failing to capture the authentic scientific process of uncovering embedded laws through the interactive exploration of complex model systems. To address these critical gaps, we introduce NEWTONBENCH, a benchmark comprising 324 scientific law discovery tasks across 12 physics domains. Our design mitigates the evaluation trilemma by using *counterfactual law shifts*—systematic alterations of canonical laws—to generate a vast suite of problems that are scalable, scientifically relevant, and memorization-resistant. Moreover, we elevate the evaluation from static function fitting to interactive *model discovery*, requiring agents to experimentally probe simulated complex systems to uncover hidden principles. Our extensive evaluation of 11 state-of-the-art LLMs reveals a clear but fragile capability for discovery in frontier models: this ability degrades precipitously with increasing system complexity and exhibits extreme sensitivity to observational noise. Notably, we uncover a paradoxical effect of tool assistance: providing a code interpreter can hinder more capable models by inducing a premature shift from exploration to exploitation, causing them to satisfice on suboptimal solutions. These results demonstrate that robust, generalizable discovery in complex, interactive environments remains the core challenge for the future of automated science. By providing a scalable, robust, and scientifically authentic testbed, NEWTONBENCH offers a crucial tool for measuring true progress and guiding the development of next-generation AI agents capable of genuine scientific discovery.

## 1 INTRODUCTION

Scientific law discovery, the process of distilling complex natural phenomena into concise, predictive mathematical principles, represents a cornerstone of human intellectual achievement, having consistently catalyzed paradigm shifts across the sciences (Langley et al., 1987; Popper, 2002). This endeavor demands a sophisticated interplay of abilities central to the scientific method: formulating hypotheses, methodical experimentation, and synthesizing empirical findings into universal laws (Peirce, 1958; Schmidt & Lipson, 2009).

The rapid advancement of Large Language Models (LLMs) has brought this grand challenge into sharp focus (OpenAI, 2024; Google, 2025). Exhibiting remarkable emergent capabilities in mathematical reasoning (Ahn et al., 2024), logical reasoning (Yu et al., 2024), and agentic planning (Huang et al., 2024b), these models appear to possess foundational skills for scientific inquiry. This

---

[*]Equal Contribution.
[†]Corresponding Author.

Table 1: Comparison of benchmarks for scientific law discovery. The "Sci. Rel." column (scientific relevance) indicates if problems are derived from real-world scientific laws (versus being synthetic). The "Mem.-Free" column indicates whether the benchmark *fully* prevents memorization.

| Benchmark | # Problems | Discovery Scope | Sci. Rel. | Mem.-Free | Scalable | Difficulty | Data Acquisition |
|---|---|---|---|---|---|---|---|
| *Traditional Real-World Laws* | | | | | | | |
| AI Feynman (Udrescu & Tegmark, 2020) | 120 | Function | ✓ | ✗ | ✗ | Partially Tunable | Passive Observation |
| EmpiricalBench (Cranmer, 2023) | 9 | Function | ✓ | ✗ | ✗ | Fixed | Passive Observation |
| *Out-of-Distribution Laws* | | | | | | | |
| LLM-SR (Shojaee et al., 2025a) | 4 | Function | ✓ | ✓ | ✗ | Fixed | Passive Observation |
| EvoSLD (Lin et al., 2025) | 5 | Function | ✓ | ✓ | ✗ | Fixed | Passive Observation |
| *Transformed Laws* | | | | | | | |
| LSR-Transform (Shojaee et al., 2025b) | 111 | Function | ✓ | ✗ | ✓ | Fixed | Passive Observation |
| *Synthetic Laws* | | | | | | | |
| LSR-Synth (Shojaee et al., 2025b) | 128 | Function | ✗ | ✓ | ✓ | Fixed | Passive Observation |
| PhysSymbol (Liu et al., 2025) | 5,000 | Function | ✗ | ✓ | ✓ | Tunable | Passive Observation |
| NEWTONBENCH (Ours) | **324** | **Model System** | ✓ | ✓ | ✓ | **Multi-dimensional** | **Active Exploration** |

has sparked widespread curiosity (Armitage, 2025; Fang et al., 2025), crystallized by provocative questions such as whether an LLM could "*rediscover Newton's laws from its own intelligence*" (He, 2023). However, answering such questions requires moving beyond anecdotal observations. To foster and guide progress toward automated science, it is critical to develop a robust testbed to rigorously benchmark the true abilities and inherent limitations of current LLMs in this domain.

Nevertheless, our closer inspection reveals that existing benchmarks for LLM-driven scientific law discovery exhibit substantive shortcomings that prevent a robust evaluation (as illustrated in Table 1). Current approaches fall into three paradigms, each with a critical limitation. First, the **Out-of-Distribution (OOD) Laws** paradigm uses complex, lesser-known principles to ensure relevance and memorization resistance, but their scarcity makes it **impractical to scale**. Second, the **Transformed Laws** paradigm provides scalability by rewriting known principles, yet remains **vulnerable to memorization**, as models may simply recall the original law rather than reason from the provided data (as evidenced in Appendix E.1). Finally, the **Synthetic Laws** paradigm achieves maximum scalability and is inherently memorization-free, but at the cost of scientific relevance, as it relies on algorithmically generated equations that **lack correspondence to real-world phenomena**. Collectively, these paradigms expose a **fundamental methodological trilemma**: a forced trade-off between scientific relevance, scalability, and resistance to memorization.

While the trilemma highlights the trade-offs *between* existing paradigms, a more fundamental limitation *underlies* them all: they frame scientific discovery as a problem of **static function discovery**, where the goal is simply to fit a mathematical formula to variables in a tabular dataset. This static framing is a stark departure from the authentic scientific process, which relies on the **interactive exploration of complex model systems** (Cartwright, 1983; de Silva et al., 2020)—for instance, J.J. Thomson's inference of the e/m ratio from cathode ray manipulation. Therefore, to meaningfully benchmark LLMs, evaluations must move beyond static function fitting and embrace the more scientifically authentic challenge of model discovery in interactive environments (Fang et al., 2025).

To address these limitations and facilitate a more rigorous evaluation, we introduce NEWTON-BENCH. Our design is built on two core principles, each targeting a fundamental shortcoming in prior work. First, to resolve the **methodological trilemma**, we introduce the *counterfactual law shift*—a technique that systematically alters the mathematical structure of canonical physical laws, for instance, by modifying operators or exponents (see examples in Figure 1). This generates a vast suite of laws that are **conceptually grounded but physically novel**. This approach simultaneously achieves the **scalability** of synthetic laws, maintains the **scientific relevance** of real-world principles, and ensures **resistance to memorization** by creating problems that cannot be solved by recall, thus forcing models to reason from first principles.

Complementing this approach, our second principle tackles the limitation of **static function discovery**. To this end, we designed NEWTONBENCH as an **interactive, system-oriented environment**. Rather than fitting a formula to a pre-existing table, an agent must actively design experiments by specifying input parameters (e.g., setting the mass of an object or the initial velocity in a simulation) and interpreting the resulting feedback from the virtual environment. Critically, the target law is embedded within a complex model with confounding variables. This design compels the agent to

engage in the **interactive exploration of a complex model system**, elevating the core challenge from simple function fitting to the more scientifically representative task of model discovery.

Furthermore, NEWTONBENCH is designed to precisely probe the limits of an LLM's scientific capabilities. It features **two independent dimensions of difficulty control**: the intrinsic complexity of the target law (tuned via the counterfactual law shift) and the extrinsic complexity of the surrounding experimental system. This allows for a fine-grained analysis of a model's breaking points. To isolate the challenge of discovery itself, we introduce an advanced setting where the LLM is provided with a code execution interface for tasks like numerical regression or hypothesis testing. By removing computational limitations as a primary bottleneck, this setup is designed to push models from being merely **compute-bound** to being truly **discovery-bound**, revealing the genuine frontiers of their scientific reasoning abilities.

Experimental results from 11 state-of-the-art LLMs on NEWTONBENCH reveal a clear but fragile capability for scientific law discovery. While frontier models like GPT-5 and Gemini-2.5-pro demonstrate proficiency in simple, isolated systems, their performance degrades precipitously as the intrinsic complexity of the law or the extrinsic complexity of the experimental system increases. This fragility is further underscored by an extreme sensitivity to observational noise, where even minimal perturbations cause a sharp decline in symbolic accuracy. Furthermore, our analysis uncovers a paradoxical effect of code assistance: providing a code interpreter boosts weaker models by offloading computation, but paradoxically hinders stronger models. We trace this performance degradation to a premature shift from exploration to exploitation, where capable agents over-rely on the tool for local optimization at the expense of discovering the globally correct law. Collectively, these results demonstrate that while LLMs are beginning to develop foundational scientific reasoning skills, achieving robust, generalizable discovery in complex, interactive environments remains the core challenge for the future of automated science.

## 2 TASK FORMALIZATION

We formalize the discovery task in NEWTONBENCH around two core concepts: *Equation* and *Model*. An agent is required to discover a target equation by experimenting within a given model.

**Definition 1** (Equation). *An **equation** $f$ is treated as a mathematical expression that acts as a function, mapping a set of input variables to a single numerical output. It is structurally represented by an **Expression Tree** $\mathcal{T}$, a form of Abstract Syntax Tree (AST).*

The nodes of $\mathcal{T}$ are categorized as follows: 1) **Leaf Nodes** $\mathcal{N}_L$: These represent the operands of the equation. They can be either *input variables* $v \in \mathcal{V}$ (e.g., mass, distance) or *numeric literals* $c \in \mathcal{C}$ (e.g., coefficients, known constants). 2) **Internal Nodes** $\mathcal{N}_I$: These represent mathematical operators $o \in \mathcal{O}$ that are applied to their child nodes. The operators can be unary (e.g., $\sin(x)$, $\sqrt{x}$) or binary (e.g., $x + y$, $x \times y$). The complete set of operators $\mathcal{O}$ used in NEWTONBENCH is detailed in Appendix A.3.2. The evaluation of an equation $f$ for a given assignment of values to its input variables $\mathcal{V}$ is performed via a post-order traversal of its expression tree $\mathcal{T}$. This recursive evaluation process is formally described in Appendix A.3.1.

**Definition 2** (Model). *A **model** $\mathcal{M}$ represents an experimental system and is defined as an ordered sequence of $k$ equations, $\mathcal{M} = (f_1, f_2, \ldots, f_k)$. The model takes a set of system-level input variables $\mathcal{V}_\mathcal{M}$ and produces a **set of final outputs** $\mathcal{Y}_\mathcal{M}$.*

The computation within $\mathcal{M}$ proceeds sequentially through its ordered equations. The inputs for any equation $f_i$ can be drawn from the model's primary inputs $\mathcal{V}_\mathcal{M}$ or the outputs of any preceding equations $\{y_j \mid j < i\}$. The model's final output, $\mathcal{Y}_\mathcal{M}$, is a designated subset of the collected outputs from all equations, i.e., $\mathcal{Y}_\mathcal{M} \subseteq \{y_1, \ldots, y_k\}$.

Based on these definitions, in our benchmark, one equation $f_{\text{target}}$ is designated as the hidden physical law for the agent to discover. All other equations in the given model, forming the set of *assisting equations* $\mathcal{F}_{\textbf{assist}} = \{f_1, \ldots, f_k\} \setminus \{f_{\textbf{target}}\}$. The physical laws in assisting equations are provided as known information via prompt to foster discovery.

The task's complexity is determined by the structure of $\mathcal{M}$ (illustrated in Figure 1), which we define across three parallel settings. The simplest setting is **Vanilla Equation**, where the model contains only the target law ($\mathcal{M} = (f_{\text{target}})$). The challenge escalates in the **Simple System** and **Complex**

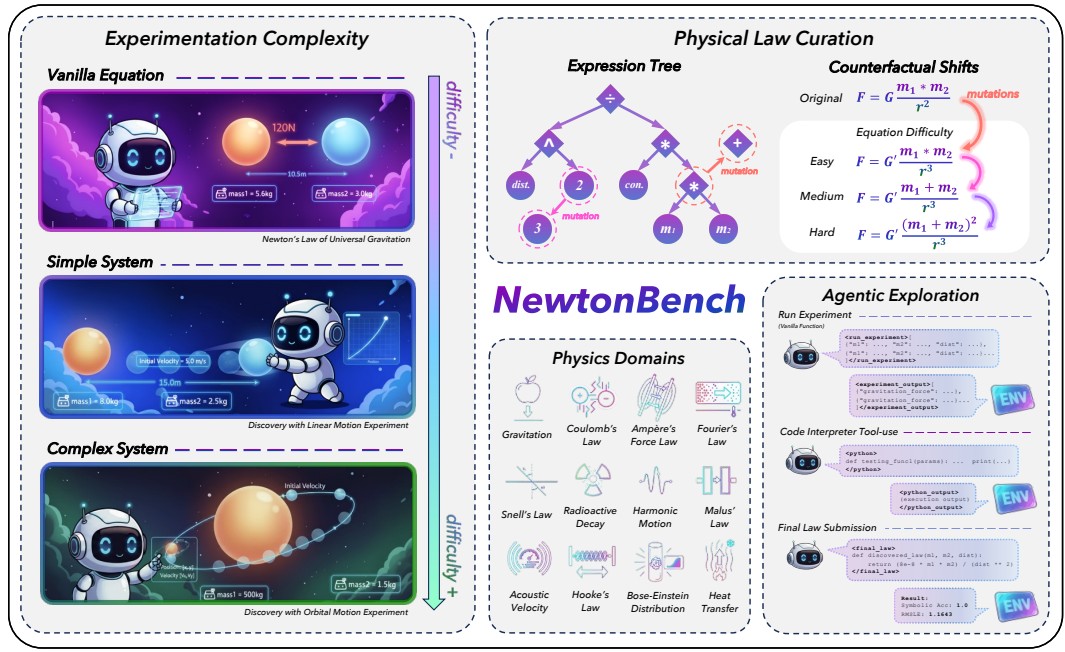

Figure 1: An illustration of core designs in NEWTONBENCH: experimentation with model system, counterfactual shifts in physical laws from various domains, and agentic exploration settings.

**System** settings, where $f_{\text{target}}$ is embedded within a larger model ($k > 1$). In these more advanced model discovery tasks, the agent must leverage assisting equations to disentangle confounding variables and uncover the target law. Details on system configurations are provided in Appendix A.2. The solvability proof of our task settings is provided in Appendix E.2.

## 3 NEWTONBENCH

In this section, we introduce the construction of NEWTONBENCH, detailing its three core components: physical law curation (§3.1), the virtual environment (§3.2), and the evaluation metrics (§3.3), with full implementation details provided in Appendix A.

### 3.1 PHYSICAL LAW CURATION

Our benchmark's physical law curation process involves two primary stages: first, the collection of canonical seed laws, and second, the application of *Counterfactual Law Shifts* to generate novel variations. For the first stage, the selection of our seed laws is guided by three criteria: they must be foundational to their respective fields, span a diverse range of physics domains, and, critically, each must allow for dimensionally consistent mutations (often via adjusting physical constants). The second stage employs *Counterfactual Law Shifts*—analogous to counterfactual reasoning in physics (Elwenspoek, 2012)—to systematically alter these canonical laws. This approach prevents agents from solving tasks by memorizing known laws, effectively creating a novel hypothetical universe for the agent to discover in each task.

Operationally, these shifts are implemented as **mutation operations** (Koza, 1994) on the equation's expression tree. These mutations can alter either the mathematical operators (e.g., changing an addition to a multiplication) or the values of numeric constants (e.g., modifying an exponent to change a quadratic relationship to a cubic one). A critical consequence of such alterations is the potential disruption of dimensional consistency. To preserve the physical coherence of the modified equation, this disruption is offset by systematically adjusting the dimensional units of the embedded

physical constant. This necessity dictates that every target law in our benchmark includes at least one such constant to serve this compensatory role.

Three levels of **equation difficulty** are categorized based on the cumulative extent of mutations applied to a canonical law[1]. *Easy* laws are generated via 1–2 mutations from the original law. *Medium* laws are subsequently derived from *Easy* laws through 1–2 additional mutations, and *Hard* laws are similarly derived from *Medium* laws. We curated 108 shifted laws from 12 canonical laws under strict guidelines to ensure scientific plausibility, conducted by three domain-expert co-authors. The size of our shifted laws is comparable to existing synthetic benchmarks (Shojaee et al., 2025b). Appendix A.3.3 provides the complete set of laws, their mutated forms, and the curation guidelines.

## 3.2 VIRTUAL ENVIRONMENT & TASKS

The LLM agent interacts with this virtual environment through two primary interfaces: an experimental tool to probe the physical model system and a code interpreter to perform complex computations. Each of the 108 shifted laws is instantiated in three model systems (Vanilla Equation, Simple System, Complex System), yielding 324 total tasks.

**Experimental Tool**  A core design principle of our benchmark is the shift from passive data observation to active agentic exploration. We formalize this by framing the experimental system, or model $\mathcal{M}$, as an interactive black-box tool. Instead of analyzing a static dataset, the agent must strategically probe $\mathcal{M}$ to gather evidence about the hidden target equation $f_{\text{target}}$.

The agent interacts with $\mathcal{M}$ by invoking a `<run_experiment>` tool. This interaction follows a precise protocol: 1) The agent proposes a specific assignment of values for the model's input variables, $\mathcal{V}_{\mathcal{M}}$. 2) The environment evaluates the full model $\mathcal{M}$ with these inputs and returns the corresponding set of final model outputs, $\mathcal{Y}_{\mathcal{M}}$.

By iteratively executing these experiments, the agent builds a dataset of input-output pairs to deduce the structure of $f_{\text{target}}$. This task is non-trivial, especially in the *Simple* and *Complex System* settings, as the agent must leverage the provided assisting equations ($\mathcal{F}_{\text{assist}}$) to reason about the model's internal state and isolate the behavior of $f_{\text{target}}$. To facilitate this process, we provide instructional scaffolding in the initial prompt regarding tool syntax and objectives (detailed in Appendix C), while reserving the experimental strategy entirely for the agent's reasoning capabilities.

**Code Assistance**  While LLMs excel at symbolic reasoning, their native computational capabilities can be a bottleneck for precisely determining numerical constants or fitting complex functional forms, particularly for equations involving exponential or logarithmic operations. To ensure our benchmark evaluates scientific discovery rather than raw computational prowess, we introduce an optional *Code Assistance* setting.

In this configuration, agents are provided with a **code interpreter tool**, invoked via `<python>` tags. The agent can write and execute arbitrary Python code and receives the output from the execution environment. Crucially, the agent must autonomously decide how and when to leverage this tool—for instance, to perform numerical computation, regression, or verify a hypothesis. By providing this computational offloading capability, we aim to shift the evaluation from being **compute-bound** to **discovery-bound**, thereby better isolating the agent's scientific reasoning abilities.

## 3.3 EVALUATION METRICS

We evaluate agent performance using two metrics: **Symbolic Accuracy** to assess the correctness of the discovered equation's structure, and **Root Mean Squared Logarithmic Error (RMSLE)** to quantify its predictive fidelity to the data.

**Symbolic Accuracy**  Following standard practice in scientific law discovery, Symbolic Accuracy is a binary metric that verifies if a discovered equation $\hat{f}$ is mathematically equivalent to the ground-truth law $f_{\text{target}}$. The equivalence check intentionally disregards the values of physical constants, as

---

[1]Consequently, equation *difficulty* arises from two factors: equation complexity and the level of distributional shift.

Table 2: Experiment results aggregated over 12 domains by equation difficulty (*easy*, *medium*, *hard*) and system complexity, reported as mean and standard deviation from four runs. Within each agent configuration, the highest score is in **bold** and the second-highest is underlined.

| LLM Agents | Vanilla Equation | | | Simple System | | | Complex System | | | Average | | |
|---|---|---|---|---|---|---|---|---|---|---|---|---|
| | *easy* | *medium* | *hard* | *easy* | *medium* | *hard* | *easy* | *medium* | *hard* | SA(%)↑ | RMSLE↓ | #Tokens (k) |
| *Vanilla Agent* | | | | | | | | | | | | |
| GPT-4.1-mini | 20.1 (±1.389) | 7.6 (±3.495) | 2.8 (±2.268) | 8.3 (±4.536) | 2.8 (±2.268) | 0.0 (±0.000) | 4.2 (±1.604) | 0.0 (±0.000) | 0.0 (±0.000) | 5.1 (±0.777) | 4.3332 (±0.427) | 1.98 |
| GPT-4.1 | 16.7 (±5.556) | 4.2 (±1.604) | 1.4 (±1.604) | 12.5 (±4.811) | 4.9 (±2.660) | 2.1 (±2.660) | 6.9 (±1.604) | 2.8 (±3.208) | 0.7 (±1.389) | 5.8 (±0.812) | 3.8241 (±0.233) | 2.82 |
| o4-mini | 88.9 (±2.268) | 78.5 (±3.495) | 52.8 (±9.886) | 68.1 (±7.349) | 46.5 (±3.495) | 22.2 (±2.268) | 47.2 (±3.928) | 22.9 (±5.258) | 2.8 (±2.268) | 47.8 (±1.753) | 1.2373 (±0.136) | 11.57 |
| GPT-5-mini | 89.6 (±6.159) | 79.9 (±4.167) | 60.4 (±4.167) | 78.5 (±3.495) | 59.0 (±1.389) | 26.4 (±4.811) | 50.0 (±3.928) | 29.2 (±2.778) | 4.9 (±2.660) | 53.1 (±0.564) | 0.7193 (±0.078) | 9.41 |
| GPT-5 | 90.3 (±3.586) | 90.3 (±2.778) | 87.5 (±6.612) | 92.4 (±6.564) | 81.9 (±6.612) | 64.6 (±2.660) | 72.9 (±2.660) | 63.2 (±5.727) | 40.3 (±5.319) | 75.9 (±1.260) | **0.2490** (±0.059) | 19.15 |
| DeepSeek-V3 | 39.6 (±1.389) | 12.5 (±2.778) | 2.1 (±2.660) | 18.1 (±4.811) | 0.7 (±1.389) | 0.0 (±0.000) | 6.9 (±2.778) | 0.0 (±0.000) | 0.0 (±0.000) | 8.9 (±1.319) | 3.0225 (±0.152) | 1.59 |
| DeepSeek-R1 | 88.2 (±4.167) | 72.9 (±3.495) | 36.8 (±6.159) | 75.0 (±11.785) | 41.7 (±5.072) | 12.5 (±1.604) | 40.3 (±7.349) | 20.1 (±10.486) | 2.8 (±3.928) | 43.4 (±1.369) | 1.3839 (±0.172) | 16.46 |
| QwQ-32B | 74.3 (±5.258) | 55.6 (±9.886) | 20.8 (±3.586) | 47.2 (±7.522) | 25.7 (±8.295) | 0.7 (±1.389) | 20.8 (±5.782) | 11.1 (±5.072) | 0.0 (±0.000) | 28.5 (±3.241) | 2.4270 (±0.201) | 14.49 |
| Qwen3-235B | 81.9 (±1.604) | 60.4 (±4.744) | 23.6 (±5.782) | 51.4 (±5.319) | 25.0 (±6.804) | 1.4 (±1.604) | 28.5 (±8.599) | 4.9 (±2.660) | 0.7 (±1.389) | 30.9 (±0.976) | 1.9385 (±0.085) | 13.91 |
| Gemini-2.5-flash | 93.1 (±2.778) | 81.9 (±5.782) | 40.3 (±5.782) | 79.9 (±6.159) | 63.9 (±6.001) | 23.6 (±3.586) | 41.0 (±5.727) | 26.4 (±5.319) | 5.6 (±3.208) | 50.6 (±1.553) | 0.7904 (±0.089) | 32.11 |
| Gemini-2.5-pro | 96.5 (±2.660) | 86.8 (±2.660) | 69.4 (±4.536) | 86.1 (±3.928) | 76.4 (±5.319) | 47.2 (±6.804) | 62.5 (±3.586) | 47.2 (±3.928) | 16.7 (±2.268) | 65.4 (±1.403) | 0.3535 (±0.041) | 21.54 |
| *Agent with Code Assistance* | | | | | | | | | | | | |
| GPT-4.1-mini | 35.4 (±1.389) | 24.3 (±1.389) | 16.7 (±2.268) | 18.8 (±6.159) | 11.1 (±3.928) | 6.9 (±4.811) | 13.2 (±6.944) | 5.6 (±2.268) | 0.7 (±1.389) | 14.7 (±0.636) | 2.7830 (±0.206) | 2.92 |
| GPT-4.1 | 52.1 (±3.495) | 31.9 (±1.604) | 20.1 (±2.660) | 29.2 (±2.778) | 10.4 (±4.167) | 9.0 (±4.167) | 13.2 (±4.167) | 2.1 (±2.660) | 2.1 (±1.389) | 18.9 (±0.772) | 2.5044 (±0.229) | 4.53 |
| o4-mini | 87.5 (±3.586) | 70.8 (±4.811) | 47.2 (±8.178) | 67.4 (±4.744) | 52.1 (±4.744) | 15.3 (±6.991) | 43.1 (±6.991) | 17.4 (±2.660) | 5.6 (±3.208) | 45.1 (±2.038) | 1.2078 (±0.077) | 10.26 |
| GPT-5-mini | 88.2 (±1.389) | 73.6 (±4.811) | 52.1 (±4.744) | 75.0 (±5.072) | 51.4 (±7.349) | 25.0 (±4.536) | 43.8 (±8.599) | 20.1 (±3.495) | 4.2 (±1.604) | 48.1 (±2.153) | 0.7412 (±0.111) | 9.29 |
| GPT-5 | 90.3 (±1.604) | 89.6 (±2.660) | 82.6 (±4.744) | 88.2 (±1.389) | 78.5 (±4.167) | 59.0 (±9.178) | 74.3 (±6.159) | 59.0 (±4.167) | 38.2 (±4.744) | 73.3 (±2.384) | **0.3793** (±0.117) | 18.36 |
| DeepSeek-V3 | 45.8 (±5.782) | 26.4 (±5.782) | 9.0 (±2.660) | 17.4 (±4.744) | 5.6 (±3.928) | 1.4 (±1.604) | 11.1 (±2.268) | 2.1 (±2.660) | 0.7 (±1.389) | 13.3 (±1.403) | 2.8795 (±0.108) | 2.23 |
| DeepSeek-R1 | 73.6 (±4.811) | 56.2 (±2.660) | 34.7 (±9.755) | 58.3 (±3.928) | 38.2 (±4.167) | 13.2 (±4.167) | 38.2 (±1.389) | 21.5 (±4.744) | 2.8 (±2.268) | 37.4 (±1.478) | 1.3415 (±0.127) | 16.92 |
| QwQ-32B | 71.5 (±5.727) | 59.0 (±4.744) | 25.7 (±2.660) | 52.8 (±7.172) | 32.6 (±5.727) | 2.1 (±1.389) | 27.1 (±2.660) | 11.1 (±3.928) | 0.7 (±1.389) | 31.4 (±1.698) | 2.0599 (±0.123) | 14.62 |
| Qwen3-235B | 75.7 (±9.178) | 54.2 (±1.604) | 27.1 (±4.167) | 61.8 (±5.258) | 34.7 (±2.778) | 10.4 (±2.660) | 26.4 (±3.586) | 13.2 (±4.167) | 0.0 (±0.000) | 33.7 (±1.165) | 1.9420 (±0.090) | 13.30 |
| Gemini-2.5-flash | 84.0 (±6.159) | 63.9 (±5.072) | 28.5 (±7.979) | 69.4 (±7.172) | 50.0 (±2.268) | 13.2 (±4.167) | 38.9 (±6.804) | 18.1 (±3.586) | 2.1 (±1.389) | 40.9 (±1.297) | 1.1694 (±0.116) | 17.73 |
| Gemini-2.5-pro | 90.3 (±3.586) | 85.4 (±1.389) | 66.7 (±2.268) | 86.1 (±3.928) | 72.9 (±4.167) | 43.1 (±5.319) | 60.4 (±3.495) | 45.8 (±5.782) | 22.2 (±3.928) | 63.7 (±1.080) | 0.3838 (±0.068) | 21.93 |

they are difficult to determine precisely from empirical data.

$$\text{Accuracy}_{\text{sym}} = \mathbb{I}(\text{equiv}(\hat{f}, f_{\text{target}})) \tag{1}$$

Here, $\mathbb{I}(\cdot)$ is the indicator function and equiv$(\cdot, \cdot)$ checks for symbolic equivalence. However, traditional rule-based equivalence checks can be brittle when handling the algebraic complexity of expressions. We therefore introduce an LLM-as-a-judge protocol for this verification step, which achieves **98.3%** agreement with human experts, demonstrating its reliability (see Appendix A.4.1).

**Data Fidelity via RMSLE**   To measure the data fidelity of a submitted law, we adopt the RMSLE metric. Unlike the metric of Normalized Mean Squared Error (NMSE) used in some prior studies (Shojaee et al., 2025b; Lin et al., 2025), RMSLE offers superior numerical stability and is better suited for physical quantities that may span multiple orders of magnitude due to its logarithmic scale. Given a dataset of $N$ points with ground-truth values $y_i$ and corresponding predictions $\hat{y}_i$ from the submitted law, RMSLE is calculated as (implementation details in Appendix A.4.2):

$$\text{RMSLE} = \sqrt{\frac{1}{N} \sum_{i=1}^{N} \left(\log(\hat{y}_i + 1) - \log(y_i + 1)\right)^2} \tag{2}$$

## 4   EXPERIMENT AND ANALYSIS

In this section, we present our main experimental results (§4.1) and conduct further analysis regarding noise robustness (§4.2), cross-domain performance (§4.3), inference scaling (§4.4), and impact of code assistance(§4.5). The complete experimental results are provided in Appendix B.

## 4.1 CAN LLM AGENTS PERFORM GENERALIZABLE SCIENTIFIC LAW DISCOVERY?

The main objective of NEWTONBENCH is to authentically assess the generalizable scientific law discovery abilities of LLMs, framed by the intuitive question: "*(To what extent) Can LLMs Rediscover Newton's Laws?*" To answer this question, we evaluate 11 state-of-the-art LLMs (details in Appendix A.1), including open-source models such as DeepSeek-R1 and closed-source models such as GPT-5 and Gemini-2.5-pro. We present the general benchmark performance in Table 2, and the full results of each physics domains are reported in Appendix B.1. We analyze model performance from multiple perspectives and summarize our main findings as follows:

**NEWTONBENCH is reasoning-intensive, failing non-thinking LLMs.** Our benchmark proves highly challenging for models with weaker reasoning abilities. All three non-thinking LLMs (GPT-4.1-mini, GPT-4.1, and DeepSeek-V3) achieve overall symbolic accuracies below 10%. While these models show limited proficiency (20-40% accuracy) even in the simplest setting, their performance degrades precipitously in all other configurations, demonstrating that success in NEWTONBENCH is contingent on strong, generalizable reasoning.

**Performance of Reasoning Models Diverges with Increasing Complexity.** Most reasoning models achieve overall accuracies between 30-80%, with GPT-5 and Gemini-2.5-pro distinguishing themselves at 75.9% and 65.4%, respectively. While these models all demonstrate proficiency in the easiest settings (70-100% accuracy), a substantial performance gap emerges as the difficulty escalates. In the most challenging setting, GPT-5 and Gemini-2.5-pro maintain accuracies of 40.3% and 16.7%, whereas all other reasoning models fall below 6%. This stark performance gap demonstrates that the benchmark's difficulty controls are highly effective at stress-testing and stratifying advanced reasoning abilities.

**Code Assistance Exhibits a Dichotomous Impact.** The provision of a code interpreter has a surprisingly dichotomous effect on agent performance. For less capable models (SA < 40%), access to the code tool substantially improves symbolic accuracy. Conversely, for more capable models (SA ≥ 40%), the inclusion of the tool leads to a slight degradation in performance—a consequence of these models satisficing through over-exploitation, as we detail in §4.5.

So, *to what extent can LLMs rediscover Newton's Laws*? Our findings indicate a clear but fragile capability. While current frontier reasoning models can deduce scientific laws in simple, well-isolated systems, their performance degrades precipitously as the complexity of the system or target equation increases. This exposes fundamental limitations in their generalizable reasoning and highlights that **robust generalization remains the core challenge**.

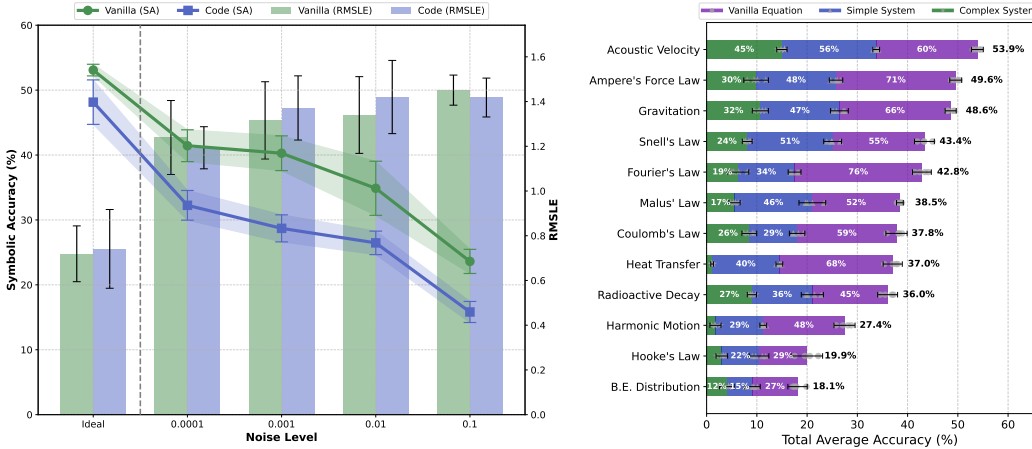

Figure 2: Impact of noise levels on performance.          Figure 3: Result across physics domains.

## 4.2 SENSITIVITY TO OBSERVATIONAL NOISE

To investigate robustness against noisy observations, we conducted an experiment on GPT-5-mini with four levels of Gaussian noise (0.0001, 0.001, 0.01, and 0.1). As illustrated in Figure 2, perfor-

mance degrades significantly even with minimal noise, with the introduction of just a 0.0001 noise level causing a **12-16%** reduction in accuracy compared to the ideal, noise-free setting. As the noise level was increased from 0.0001 to 0.1, symbolic accuracy declined proportionally, while data fidelity (RMSLE) remained relatively stable. Notably, code assistance did not affect noise robustness, with the baseline and code-assisted agents degrading similarly. This demonstrates that **the symbolic accuracy of LLMs is extremely fragile to noise in observational data.**

### 4.3 CROSS-DOMAIN PERFORMANCE DISPARITIES

Performance across the physics domains, illustrated in Figure 3, reveals a substantial difference in average accuracy, with scores ranging from 18% to 54%. *Bose-Einstein Distribution*, the most advanced and obscure domain in our benchmark, yields the lowest average accuracy at 18.1%. Furthermore, the performance disparities are exacerbated as system complexity increases. For instance, in the simple setting, *Heat Transfer* yields 68% accuracy, comparable to the easiest domain, *Acoustic Velocity* (60%). In the complex setting, however, accuracy for *Heat Transfer* plummets to 3.3%, while *Acoustic Velocity* remains at 45.0%. This demonstrates that **the intrinsic nature of a physical domain is a primary factor governing the difficulty of model discovery.** Full domain analysis is provided in Appendix E.3.

### 4.4 INFERENCE SCALING WITH TASK COMPLEXITY

Can LLMs effectively scale inference for more challenging tasks? Figure 4 compares strong reasoning LLMs (Gemini-2.5-pro/flash, GPT-5/5-mini) with non-reasoning LLMs (GPT-4.1/4.1-mini, DeepSeek-V3) on their token cost and number of rounds required to solve tasks of varying difficulty. For strong reasoning LLMs, token consumption (i.e., reasoning length) increases significantly as task difficulty rises. In contrast, the token cost for non-reasoning models remains consistently low. This demonstrates that **reasoning models can substantially scale up their computational effort to solve more complex tasks, whereas non-reasoning models fail to do so**, even when consuming more experiment rounds. This superior scaling likely drives the advantage of strong models on complex tasks. Full results are provided in Appendix B.2.

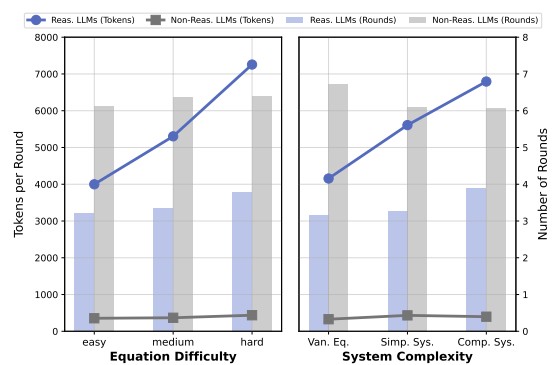

Figure 4: Inference cost among difficulty levels.

### 4.5 THE PARADOX OF CODE ASSISTANCE: THE EXPLORATION-EXPLOITATION TRADE-OFF

To understand the dichotomous impact of code assistance, we conducted an experiment on four representative LLMs with varying code-use budgets. As illustrated in Figure 5(a), the performance divergence is most pronounced when the code budget increases from zero to one: the performance of stronger models (Gemini-2.5-flash, GPT-5-mini) degrades, while that of weaker models improves. We hypothesize this divergence stems from a fundamental shift in the models' problem-solving strategies, specifically the balance between exploration and exploitation[2].

To investigate this hypothesis, we adapt a common analysis approach for reasoning LLMs (Wang et al., 2025b;a) by **identifying signature tokens associated with exploration** (e.g., *What if*, *Alternatively*) **and exploitation** (e.g., *Confirm*, *Verify*). From these, we calculate the *exploration rate* (the percentage of exploration tokens among all such planning tokens; see Appendix E.4 for details). Figure 5(b) shows the results for two reasoning LLMs where reasoning traces are accessible. We observe a **sharp drop in the exploration rate** for Gemini-2.5-flash as the code budget increases

---

[2]The trade-off between discovering novel solutions (exploration) and optimizing the current best-known solution (exploitation).

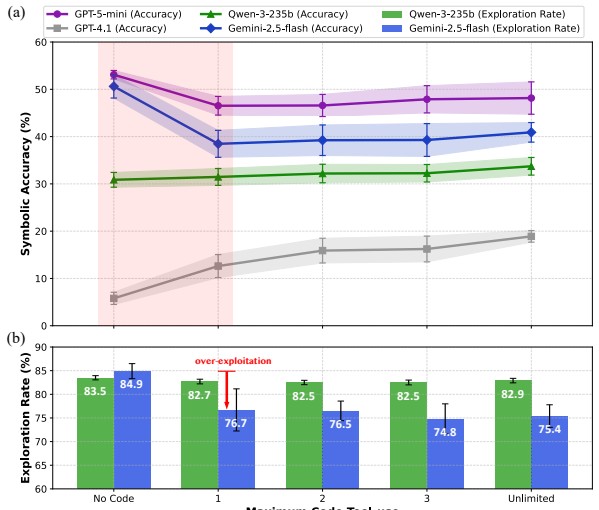

Figure 5: Results under different code-use budgets.

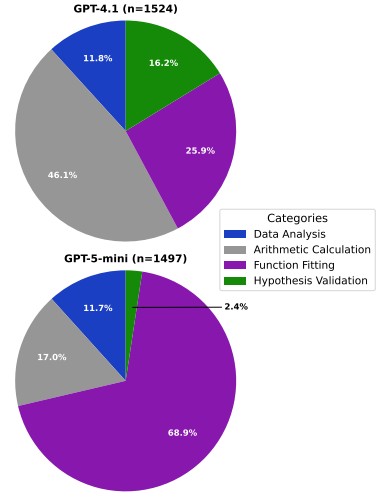

Figure 6: Functionality distribution for code usage (unlimited budget).

from zero to one, while the rate remains stable for Qwen-3-235b. This suggests the performance degradation in stronger models reflects over-exploitation[3].

To understand the mechanism driving this strategic shift, we analyzed the functional distribution of code usage for a strong and a weak model (GPT-5-mini and GPT-4.1). Figure 6 reveals that GPT-5-mini allocates a significantly smaller proportion of its code use to basic calculation compared to GPT-4.1, instead favoring function-fitting. This supports the hypothesis that **code serves distinct roles**: a computational tool for weaker models versus an equation refinement tool for stronger ones.

In summary, for weaker LLMs, whose primary bottleneck is arithmetic computation, code execution provides crucial assistance, thereby improving their performance. Conversely, stronger LLMs already possess sufficient computational prowess and tend to leverage code for tasks like function-fitting. This can accelerate convergence to a "good enough" solution, causing the model to prematurely settle in a local optimum. Such over-exploitation stifles the broader exploration needed to discover a globally optimal answer, paradoxically degrading the performance of these more capable models (further case studies in Appendix D). This finding highlights the **importance of managing the exploration-exploitation trade-off in agentic systems**, particularly in how tools are leveraged by models of varying capability for data-driven discovery.

## 5 RELATED WORK

**Symbolic Regression** Symbolic regression (SR) aims to discover mathematical formulas from data, with foundational approaches being Genetic Programming (GP) that evolve populations of candidate expressions (Koza, 1994; Augusto & Barbosa, 2000; Billard & Diday, 2002; Schmidt & Lipson, 2009). Meanwhile, more advanced approaches have been proposed, from physics-inspired, Pareto-driven strategies (Udrescu & Tegmark, 2020; Udrescu et al., 2020) to deep learning-based models (Petersen et al., 2021; Kamienny et al., 2022). The latest evolution in SR leverages the reasoning capabilities of LLMs to hypothesize and refine scientific equations (Shojaee et al., 2025a;b; Xia et al., 2025; Chen et al., 2025a).

**LLM-Driven Scientific Discovery** The agentic capabilities of LLMs are increasingly being applied to automate the scientific process (Zheng et al., 2025; Wei et al., 2025), from assisting with experimental procedures in chemistry and biomedicine (Gottweis et al., 2025; Yang et al., 2025; Luo et al., 2025; Chen et al., 2025b) to automating complex research workflows in machine learning (Chan et al., 2025; Huang et al., 2024a; Jiang et al., 2025; Jansen et al., 2025). This trend culminates in the pursuit of a fully autonomous "AI Scientist" capable of managing the entire research pipeline for open-ended discovery (Lu et al., 2024; Yamada et al., 2025).

---

[3]These observations are correlational and should be interpreted as suggestive rather than causal.

**Virtual Environment for LLM Agents**    Virtual environments for LLM agents have evolved from general-purpose sandboxes for tasks like instruction following (Shridhar et al., 2021), web navigation (Yao et al., 2023), and compositional planning (Prasad et al., 2024), to specialized platforms for scientific reasoning, ranging from curriculum-based tasks (Wang et al., 2022) to open-ended discovery (Jansen et al., 2024).

## 6    CONCLUSION

In this work, we present NEWTONBENCH, the first scientific law discovery benchmark designed to resolve the long-standing trade-off between scientific relevance, scalability, and resistance to memorization. Moreover, it elevates the evaluation from static function fitting to interactive model discovery, requiring agents to experimentally probe complex systems to uncover hidden governing principles. Our benchmarking reveals that while frontier models possess a clear capability for discovery, this ability is fragile, degrading precipitously with increasing system complexity and observational noise. We further uncover a paradoxical effect where tool assistance hinders more capable models, inducing a premature shift from exploration to exploitation that causes them to satisfice on suboptimal solutions. Looking forward, we hope NEWTONBENCH serves as a crucial litmus test for the reasoning capabilities of frontier LLMs and agentic systems, faithfully measuring genuine scientific intelligence to guide the development of AI capable of authentic discovery.

## ACKNOWLEDGMENTS

The authors of this paper were supported by the ITSP Platform Research Project (ITS/189/23FP) from ITC of Hong Kong, SAR, China, and the AoE (AoE/E-601/24-N), and the GRF (16205322) from RGC of Hong Kong, SAR, China. We also thank the support from NVIDIA AI Technology Center (NVAITC).

## ETHICS STATEMENT

NEWTONBENCH is designed exclusively for benchmarking and evaluating the scientific discovery capabilities of LLM agents in a fully contamination-free, virtual environment. The counterfactual law shifts applied to physical laws ensure the tasks are novel and not directly suitable for training LLMs to discover real-world physics. Due to the interactive, model-system nature of the tasks, we exclude traditional symbolic regression methods that do not support model systems and LLM-based symbolic regression pipelines whose prior reliance and workflows are incompatible with our protocol; these are outside the scope of this evaluation. We caution that the benchmark is not intended as a training dataset for developing stronger physics discovery models, but rather as a controlled tool for fair and rigorous evaluation. All shifted laws were curated by expert co-authors (Physics Olympiad backgrounds) and cross-validated. Furthermore, while the physics domain was selected as the most representative ground for scientific law discovery, we posit that the findings of NEWTONBENCH are generalizable to fields such as chemistry and biology. This generalization holds because the core cognitive process—isolating variables to derive mathematical relationships—is isomorphic across disciplines. Whether elucidating reaction kinetics in chemistry or modeling enzyme dynamics in quantitative biology, the discovery process demands the same rigorous interplay of hypothesis testing and symbolic abstraction. Consequently, NEWTONBENCH serves as a robust indicator of an agent's potential to drive discovery across the broader spectrum of the natural sciences.

## REPRODUCIBILITY STATEMENT

All LLM evaluations in this work were conducted via public APIs, specifically using the OpenRouter and OpenAI-API platforms, with the total experimental cost estimated at 10,000 USD. Detailed model configurations are provided in Appendix A. Experimental results are reported with measures of statistical significance to ensure robustness. We additionally provide all prompt templates and system implementations in Appendix to facilitate full reproducibility of our experiments. All code and data are publicly released to foster future research.

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

APPENDIX

# A  IMPLEMENTATION DETAILS

## A.1  LLM DETAILS

In our experiments, we evaluated 11 state-of-the-art LLMs, including both open-source and proprietary models. The models are summarized as follows:

- **GPT-4.1-mini** (OpenAI, 2025a) is a lightweight proprietary non-reasoning LLM from OpenAI, designed for efficient inference.
- **GPT-4.1** (OpenAI, 2025a) is the latest flagship proprietary non-reasoning LLM from OpenAI.
- **o4-mini** (OpenAI, 2025c) is a proprietary reasoning LLM developed by OpenAI, optimized for lightweight inference and efficient deployment.
- **GPT-5-mini** (OpenAI, 2025b) is a lightweight reasoning LLM from OpenAI, aiming for improved efficiency on downstream tasks.
- **GPT-5** (OpenAI, 2025b) is the strongest reasoning LLM by OpenAI with advanced reasoning capabilities.
- **DeepSeek-V3** (DeepSeek-AI, 2024) is an open-source non-reasoning LLM released by DeepSeek, built for robust general-purpose language understanding.
- **DeepSeek-R1** (DeepSeek-AI, 2025) is an open-source reasoning LLM from DeepSeek, trained with reinforcement learning to enhance reasoning and alignment.
- **QwQ-32b** (Qwen, 2025a) is an open-source reasoning LLM by Qwen team, leveraging reinforcement learning and agent-driven tool use for complex problem solving.
- **Qwen-3-235b** (Qwen, 2025b) is an open-source reasoning LLM by Qwen team, pre-trained on large corpora for high-performance language and reasoning tasks.
- **Gemini-2.5-flash** (Google, 2025) is a proprietary reasoning LLM from Google's Gemini series, optimized for ultra-long context windows and rapid inference.
- **Gemini-2.5-pro** (Google, 2025) is the advanced proprietary reasoning LLM in the Gemini series, offering enhanced multimodal understanding and superior reasoning abilities.

By "reasoning" and "non-reasoning", we distinguish whether a deliberate thinking process—often incentivized by reinforcement learning with verifiable rewards (RLVR)—is triggered before the generation of the first answer token (Li et al., 2025).

In all experiments, we set the LLM temperature to 0.4. Statistical significance is assessed based on four independent runs for each experiment. The main results table reports the standard deviation, while other plots display error bars representing the 95% confidence interval for the mean. In each test case, LLM agents are allowed up to 10 experimental rounds, with at most 20 input-parameter sets per round. This design prevents unbounded interaction and helps isolate strategy design.

## A.2 MODEL SYSTEMS

All model systems in NEWTONBENCH are human-curated and inspired by classical experiments from each domain. As an example, we illustrate a *Simple System* in the Acoustics domain. Detailed descriptions of the model systems across all 12 domains are provided in the **Supplementary Materials**.

SIMPLE SYSTEM (LAW OF SOUND SPEED IN IDEAL GAS)

The model simulates the emission of a sound pulse within a gas chamber, directed toward a wall positioned at a known distance, and records the time required for the echo to return. The agent's task is to deduce the target law, $f_{\text{target}}$, by interacting with the simulation.

**System-Level Input Variables ($\mathcal{V}_{\mathcal{M}}$)**   The simulation accepts the following inputs:

- $\gamma$: The adiabatic index of the gas in the chamber
- $M$: The molar mass of the gas
- $T$: The temperature of the gas
- $d$: The distance to the wall

**System-Level Final Output ($\mathcal{Y}_{\mathcal{M}}$)**   After the simulation runs, it returns the following output:

- $t$: The total time taken for the echo to return

**Ordered Sequence of Equations ($\mathcal{M}$)**   The simulation environment internally computes the final measured time by executing a fixed sequence of calculations. This sequence of operations, $\mathcal{M} = (f_1, f_2)$, is hidden from the agent. However, the fundamental principle underlying the assisting equation ($f_2$) are provided to the agent to guide its discovery.

1. $f_1$ **(target)**:
$$v_{sound} = f_{\text{target}}(\gamma, T, M)$$

2. $f_2$ **(assist)**:
$$t = \frac{2d}{v_{sound}}$$

SYSTEM COMPLEXITY ANALYSIS

We analyze the system complexity of all 12 physical domains based on the following two metrics:

- **Number of Assisting Equations**: The number of equations $f$ applied in model $M$ that are not $f_{target}$.
- **Number of Operations**: The total number of operators (e.g., $+$, $*$, ˆ, $\sin$) that appear in all assisting laws $f$. This metric reflects the computational complexity of the model system (excluding the target law).

As illustrated in Table 3, complex systems are associated with a higher quantity of assisting equations and operations, highlighting the computational challenges inherent in the law discovery process.

Table 3: Comparison in numbers of assisting equations and operations between simple system and complex system across all 12 physical domains.

| Physical Domain | Simple System | | Complex System | |
|---|---|---|---|---|
| | Number of Assist Equations | Number of Operations | Number of Assist Equations | Number of Operations |
| Newton's Law of Universal Gravitation | 3 | 16 | 4 | 22 |
| Coulomb's Law | 2 | 8 | 3 | 14 |
| Ampère's Force Law | 4 | 10 | 5 | 14 |
| Fourier's Law | 1 | 4 | 1 | 6 |
| Snell's Law | 1 | 2 | 1 | 4 |
| Law of Radioactive Decay | 1 | 2 | 1 | 3 |
| Law of Damped Harmonic Motion | 1 | 2 | 2 | 3 |
| Malus's Law | 1 | 2 | 1 | 3 |
| Law of Sound Speed in Ideal Gas | 1 | 2 | 2 | 7 |
| Hooke's Law | 2 | 5 | 3 | 6 |
| Bose-Einstein Distribution | 1 | 2 | 2 | 6 |
| Law of Heat Transfer | 3 | 7 | 4 | 10 |
| **Average** | 1.75 | 5.17 | 2.42 | 8.17 |

### A.3 EQUATION DETAILS

#### A.3.1 EXPRESSION TREES

The output of any given expression tree (AST) for a specific set of input variables is computed using the canonical recursive algorithm detailed in this section. The evaluation strategy follows a post-order traversal: the value of any internal node (an operator) is computed only after the values of all its children have been determined. This ensures that operators are always applied to fully evaluated operands. The algorithm requires the tree's root node and a map containing the numerical values for all variables present in the expression. Algorithm 1 provides a formal, high-level description of this process.

---

**Algorithm 1** Recursive Evaluation of an Expression Tree

---

**Require:** Node $n$, variable mapping $\mathcal{V} : \text{var} \mapsto \mathbb{R}$
**Ensure:** Numerical result $r \in \mathbb{R}$ of the expression rooted at $n$

1: **function** EVALUATE($n$, $\mathcal{V}$)
2: **if** $n$.type = CONSTANT **then**
3:     **return** $n$.value
4: **else if** $n$.type = VARIABLE **then**
5:     **return** $\mathcal{V}[n.\text{name}]$
6: **else if** $n$.type = OPERATOR **then**
7:     $values \leftarrow \emptyset$ {Initialize empty list}
8:     **for each** $c \in n$.children **do**
9:         $v \leftarrow$ EVALUATE($c$, $\mathcal{V}$)
10:         $values$.APPEND($v$)
11:     **end for**
12:     $op \leftarrow n$.operator
13:     **return** APPLY($op$, $values$)
14: **end if**
15: **end function**

---

As formalized in Algorithm 1, the evaluation logic operates through two fundamental cases:

- **Base cases:** Recursion terminates at leaf nodes. For CONSTANT nodes, the intrinsic numerical value is returned directly. For VARIABLE nodes, the corresponding value is retrieved from the variable mapping $\mathcal{V}$.

- **Recursive case:** For OPERATOR nodes, the algorithm first recursively evaluates all child nodes in post-order fashion, collecting their results. The operator is then applied to these operands via the APPLY function, returning the computed result.

The time complexity is $\mathcal{O}(|n|)$ where $|n|$ is the number of nodes in the AST, as each node is visited exactly once during the traversal.

### A.3.2 OPERATOR SETS

The complete operator space of NEWTONBENCH is presented in Table 4, which specifies the symbol and arity for each supported mathematical operator.

Table 4: Mathematical operators used across all physics modules

| Operator Name | Symbol | Arity |
|---|---|---|
| Addition | $+$ | Binary |
| Subtraction | $-$ | Binary |
| Multiplication | $\times$ | Binary |
| Division | $\div$ | Binary |
| Exponentiation | $x^y$ | Binary |
| Exponential | $\exp(x)$ | Unary |
| Natural Logarithm | $\log(x)$ | Unary |
| Square Root | $\sqrt{x}$ | Unary |
| Sine | $\sin(x)$ | Unary |
| Cosine | $\cos(x)$ | Unary |
| Tangent | $\tan(x)$ | Unary |
| Arcsine | $\arcsin(x)$ | Unary |
| Arccosine | $\arccos(x)$ | Unary |
| Arctangent | $\arctan(x)$ | Unary |

### A.3.3 PHYSICAL LAWS

Table 5 summarizes the physical laws and their corresponding counterfactual shifts. These counterfactual shifts are achieved through mutation operations applied to the original equations, curated by human experts. The curation process follows four core guidelines:

1. Each mutation must be performed within the set, and no new variables should be introduced.

2. Mutations should increase, rather than decrease, the overall complexity of the equation.

3. The new mutation operation must not simply reverse the previous mutation.

4. The outcome of consecutive mutations should not be attainable through a single mutation operation.

**Physical plausibility caveat.** All mutated laws are dimensionally coherent by construction, but some may be physically implausible in our universe; we intend these counterfactual variants as scientifically grounded stress tests of reasoning and interactive discovery, not as claims about real-world physics.

Table 5: Full physical laws and their counterfactual shifts.

| Physics Law | Domain | Original Equation | Shifted Equations | | |
|---|---|---|---|---|---|
| | | | *Easy* | *Medium* | *Hard* |
| *Newton's Law of Universal Gravitation* | Mechanics (Gravitation) | $F = G\frac{m_1 m_2}{r^2}$ | $F = G'\frac{m_1 m_2}{r^{1.5}}$ | $F = G'\frac{(m_1 m_2)^2}{r^{1.5}}$ | $F = G'\frac{(m_1+m_2)^2}{r^{1.5}}$ |
| | | | $F = G'\frac{m_1}{r^2}$ | $F = G'\frac{m_1}{r^{2.6}}$ | $F = G'\frac{m_1^{1.3}}{r^{2.6}}$ |
| | | | $F = G'\frac{m_1^2 m_2^2}{r^2}$ | $F = G' \cdot m_1^2 m_2^2 \cdot r^2$ | $F = G' \cdot (m_1^2 + m_2^2) \cdot r^2$ |
| *Coulomb's Law* | Electricity and Magnetism (Electrostatics) | $F = k\frac{q_1 q_2}{r^2}$ | $F = k'\frac{q_1 q_2}{r^3}$ | $F = k'\frac{q_1 q_2 (q_1+q_2)}{r^2}$ | $F = k'\frac{q_1 q_2 (q_1+q_2)}{r^2}$ |
| | | | $F = k'\frac{(q_1 q_2)^3}{r^2}$ | $F = k'\frac{(q_1+q_2)^3}{r^2}$ | $F = k'\frac{q_2^2 (q_1+q_2)^3}{r^2}$ |
| | | | $F = k'\frac{q_1^3 q_2}{r^2}$ | $F = k'\frac{q_1^3 q_2^2}{r^{2.5}}$ | $F = k'\frac{q_1^3 q_2^2}{r^6}$ |
| *Ampère's Force Law* | Electricity and Magnetism (Magnetostatics) | $F = K\frac{I_1 I_2}{r}$ | $F = K'\frac{I_1 I_2}{r^3}$ | $F = K'\frac{(I_1 I_2)^{1.5}}{r^3}$ | $F = K'\frac{(I_1+I_2)^{1.5}}{r^3}$ |
| | | | $F = K'\frac{(I_1 I_2)^2}{r}$ | $F = K'(I_1 I_2)^2 r$ | $F = K'(I_1 - I_2)^2 r$ |
| | | | $F = K'\frac{I_2}{r}$ | $F = K'\frac{I_2}{r^{3.8}}$ | $F = K'\frac{I_2^{0.9}}{r^{3.8}}$ |
| *Fourier's Law* | Thermodynamics (Thermal Conduction) | $Q = k\frac{A \Delta T}{d}$ | $Q = k'\frac{A \Delta T}{d^2}$ | $Q = k'\frac{A^{0.5} \Delta T}{d^2}$ | $Q = k'\frac{A^{0.5} (\Delta T)^{1.3}}{d^2}$ |
| | | | $Q = k'\frac{A^{0.5} \Delta T}{d}$ | $Q = k'\frac{A^{0.5} (\Delta T)^{2.7}}{d}$ | $Q = k'\frac{A^{0.5} (\Delta T)^{2.7}}{d^{(3/7)}}$ |
| | | | $Q = k'\frac{A (\Delta T)^2}{d}$ | $Q = k'\frac{(\Delta T)^2}{d A^{3.4}}$ | $Q = k'\frac{(\Delta T)^2}{\sqrt{d} A^{3.4}}$ |
| *Snell's Law* | Optics (Geometrical Optics) | $\theta_2 = \sin^{-1}\left(\frac{n_1 \sin(\theta_1)}{n_2}\right)$ | $\theta_2 = \cos^{-1}\left(\frac{n_1 \sin(\theta_1)}{n_2}\right)$ | $\theta_2 = \cos^{-1}\left(\frac{n_1 \cos(\theta_1)}{n_2}\right)$ | $\theta_2 = \cos^{-1}\left(\frac{n_1^2 \cos(\theta_1)}{n_2}\right)$ |
| | | | $\theta_2 = \sin^{-1}\left(\frac{n_2 \sin(\theta_1)}{n_1}\right)$ | $\theta_2 = \cos^{-1}\left(\frac{n_2 \sin(\theta_1)}{n_1}\right)$ | $\theta_2 = \cos^{-1}\left(\frac{n_2 \sin(\theta_1)}{n_1^{2.5}}\right)$ |
| | | | $\theta_2 = \tan^{-1}\left(\frac{n_1 \sin(\theta_1)}{n_2}\right)$ | $\theta_2 = \tan^{-1}\left(\frac{n_1 \tan(\theta_1)}{n_2}\right)$ | $\theta_2 = \tan^{-1}\left(\left(\frac{n_1}{n_2}\right)^2 \tan(\theta_1)\right)$ |
| *Law of Radioactive Decay* | Modern Physics (Nuclear Physics) | $N(t) = N_0 e^{-\lambda t}$ | $N(t) = N_0 e^{-\lambda' t^{1.5}}$ | $N(t) = N_0^{1.2} e^{-\lambda' t^{1.5}}$ | $N(t) = N_0^{1.2} e^{-\lambda' e^{t^{1.5}}}$ |
| | | | $N(t) = N_0 e^{-\lambda' t^{1.5} t}$ | $N(t) = N_0^{1.4} e^{-\lambda' t^{1.5} t}$ | $N(t) = N_0^{1.4} e^{-\lambda' t^{1.5} t^e}$ |
| | | | $N(t) = N_0 e^{-(\lambda' t)^{1.5}}$ | $N(t) = N_0^{1.8} e^{-(\lambda' t)^{1.5}}$ | $N(t) = N_0^{1.8} e^{-(\lambda' t)^{e+1.5}}$ |
| *Law of Damped Harmonic Motion* | Wave and Accoustics (Oscillations) | $\omega = \sqrt{\frac{k}{m} - \left(\frac{b}{2m}\right)^2}$ | $\omega = \sqrt{\frac{k'}{m} - \frac{b'}{2m}}$ | $\omega = \sqrt{\frac{k'}{m} - \frac{b'}{2m^2}}$ | $\omega = \left(\frac{k'}{m} - \frac{b'}{2m^2}\right)^{1.5}$ |
| | | | $\omega = \left(\frac{k'}{m} - \left(\frac{b'}{2m}\right)^2\right)^2$ | $\omega = \left(\frac{k'}{m^2} - \left(\frac{b'}{2m}\right)^2\right)^2$ | $\omega = \left(k' \cdot m^2 - \left(\frac{b'}{2m}\right)^2\right)^2$ |
| | | | $\omega = \frac{k'}{m} - \left(\frac{b'}{2m}\right)^2$ | $\omega = \frac{k'}{m^{1.3}} - \left(\frac{b'}{2m}\right)^2$ | $\omega = \frac{k'}{m^{1.3}} - \left(\frac{b'}{2m}\right)^{0.7}$ |
| *Malus's Law* | Optics (Physical Optics) | $I = I_0 \cos^2(\theta)$ | $I = I_0 (\sin(\theta) + \cos(\theta))^2$ | $I = I_0 (2\sin(\theta) + \cos(\theta))^2$ | $I = I_0 (2\sin(\theta) + 1.5\cos(\theta))^2$ |
| | | | $I = I_0 \left(\frac{\sin(\theta)}{\cos(\theta)}\right)^2$ | $I = I_0 \frac{\sin^2(\theta)}{\cos^3(\theta)}$ | $I = I_0 \left(\frac{\sin^2(\theta)}{\cos^3(\theta)}\right)^e$ |
| | | | $I = I_0 \left(\frac{\cos(\theta)}{\sin(\theta)}\right)^2$ | $I = I_0 \left(\frac{\cos(\theta)}{\sin(\theta)}\right)^e$ | $I = I_0 \left(\frac{\sin^2(\theta)}{\cos(\theta)}\right)^e$ |
| *Law of Sound Speed in Ideal Gas* | Wave and Accoustics (Acoustics) | $v = \sqrt{\frac{\gamma R T}{M}}$ | $v = \sqrt{\frac{\gamma R' T^2}{M}}$ | $v = \sqrt{\frac{\gamma R' T^2}{M^{1.5}}}$ | $v = \sqrt{\frac{e\gamma R' T^2}{M^{1.5}}}$ |
| | | | $v = \frac{\gamma R' T}{M}$ | $v = \frac{\gamma R' T}{M^{\frac{1}{3}}}$ | $v = \frac{\ln(\gamma) R' T}{M^{\frac{1}{3}}}$ |
| | | | $v = \sqrt{\frac{R' T}{M}}$ | $v = \sqrt{R' T M^{1.5}}$ | $v = (R' T M^{1.5})^{-2.8}$ |
| *Hooke's Law* | Mechanics (Elasticity) | $F = kx$ | $F = 2k_1' x^2$ | $F = 2k_1' x^2 + k_2' x$ | $F = 2k_1' x^2 + k_2' x + k_3' x^{-0.5}$ |
| | | | $F = 2k_1' x^{0.5}$ | $F = 2k_1' x^{0.5} + k_2' x^3$ | $F = 2k_1' x^{0.5} + k_2' x^3 + k_3' x^{-0.3}$ |
| | | | $F = 2k_1' x^{3.4}$ | $F = 2k_1' x^{3.4} + k_2' x^{0.5}$ | $F = 2k_1' x^{3.4} + k_2' x^{0.5} + k_3' x^{-\frac{10}{3}}$ |
| *Bose-Einstein Distribution* | Modern Physics (Statistical Mechanics) | $n = \frac{1}{e^{\left(\frac{C_\omega}{T}\right)} - 1}$ | $n = \frac{1}{e^{\left(\frac{C'\omega}{T}\right)} + 1}$ | $n = \frac{1}{e^{\left(\frac{C'\omega^{1.5}}{T^2}\right)} + 1}$ | $n = \frac{1}{e^{\left(\frac{C'\omega^{1.5}}{T^2}\right)} + 1}$ |
| | | | $n = \frac{1}{e^{\left(\frac{C'\omega^{0.5}}{T}\right)} - 1}$ | $n = \frac{1}{e^{(C'\omega^{0.5} T)} - 1}$ | $n = \frac{1}{e^{(C'\omega^{0.5} T^{2.3})} - 1}$ |
| | | | $n = \frac{1}{e^{\left(\frac{C'\omega}{T^3}\right)} - 1}$ | $n = \frac{1}{e^{\left(\frac{C'\omega^{1.5}}{T^3}\right)} - 1}$ | $n = \frac{1}{-\ln\left(\frac{C'\omega^{1.5}}{T^3}\right) - 1}$ |
| *Law of Heat Transfer* | Thermodynamics (Calorimetry) | $Q = mc(\Delta T)$ | $Q = mc'(\Delta T)^{2.5}$ | $Q = \frac{c'}{m(\Delta T)^{2.5}}$ | $Q = \frac{c'}{m^{e+1}(\Delta T)^{2.5}}$ |
| | | | $Q = m^{2.5} c'(\Delta T)$ | $Q = \frac{c'}{m^{2.5}(\Delta T)}$ | $Q = \frac{c'}{m^{2.5}(\Delta T)^{e+1}}$ |
| | | | $Q = (m\Delta T)^{2.5} c'$ | $Q = \frac{c'}{m^{2.5}(\Delta T)^{2.5}}$ | $Q = \frac{c'}{(m\Delta T)^{e+1}}$ |

### A.3.4 CATEGORIZATION ON PHYSICAL PLAUSIBILITY

In this section, we categorize all shifted laws by their physical plausibility to enhance NEWTON-BENCH's value for AI research in physics. The complete categorization is presented in Table 7.

We introduce a three-level taxonomy of *physical plausibility* to grade how severely each shifted law departs from the structural principles of its original physical law. The design is consistent with standard treatments of (i) symmetry and conservation in classical mechanics and field theory (Goldstein et al., 2001) and (ii) admissible constitutive relations and entropy inequalities in continuum and nonequilibrium thermodynamics (Coleman & Noll, 1961; Mueller & Ruggeri, 2013; Gurtin et al., 2010).

We assume standard physical domains, such as $m > 0$, $T > 0$, $r > 0$, cross-sectional area $A > 0$, times $t > 0$, and angles in physically relevant ranges; quantities such as intensity, energy, and occupation number are non-negative in the original laws. A *new singularity* of the shifted law at a point $x_0$ in the physical domain means that the magnitude of the shifted law becomes unbounded as one approaches $x_0$ from within the domain, whereas the original law remains finite (or extends continuously) at $x_0$.

We define three levels of physical plausibility for the shifted laws:

- Level 1 – Admissible Counterfactual Laws (No Structural Violation)
- Level 2 – Counterfactual Laws with Minor Structural Violation
- Level 3 – Counterfactual Laws with Major Structural Violation

**Level 1: Admissible Counterfactual Laws (No Structural Violation)**

A shifted law is assigned to **Level 1** if it satisfies all of the following:

1. **No new singularities.** For all physically allowed variables, the shifted law remains finite wherever the original law is finite. Any divergence is confined to points that were already singular in the original formulation (e.g., $r \to 0$ in inverse-power forces).

2. **Real and sign-consistent outputs.** The law yields real values on the physical domain. Quantities that are non-negative in the original law (e.g., intensity, energy, occupation number, probability) remain non-negative under the shifted law.

3. **Preservation of the natural structural pattern of the domain.** For pairwise interaction laws (e.g., gravitation, Coulomb, Ampère), the dependence on the two "charges" (masses, charges, currents) is symmetric enough that a standard action–reaction structure can be maintained and momentum conservation can be formulated in the usual way. For transport, constitutive, and statistical laws (e.g., Fourier, Hooke, calorimetry, radioactive decay, distributions), the relation between driving forces and responses remains within the broad class of mappings typically admitted in rational thermodynamics and statistical mechanics: there is no infinite response at vanishing driving force, and no obvious conflict with basic entropy-production or positivity requirements (Coleman & Noll, 1961; Mueller & Ruggeri, 2013; Gurtin et al., 2010).

In practice, Level 1 includes exponent changes and nonlinearities that resemble generalized or effective laws already used in physics (e.g., nonlinear elasticity, nonlinear conduction, stretched-exponential relaxation, generalized equations of state). Level 1 is intended for experiments that require **highly plausible counterfactuals**, such as evaluating symbolic regression and law-discovery methods under realistic but systematically modified physics.

**Level 2: Counterfactual Laws with Minor Structural Violation**

A shifted law is assigned to **Level 2** if:

1. It satisfies the same **regularity** criteria as Level 1 (no new singularities; real, sign-consistent outputs), but

2. It **breaks key symmetry, reciprocity, or asymptotic patterns** that are normally associated with conservation, invariance, or physical admissibility in the corresponding theory.

Typical examples include asymmetric two-body forces where the magnitude depends on $(m_1, m_2)$ or $(q_1, q_2)$ in a way that is not invariant under exchanging labels, so that a simple action–reaction pair cannot be recovered. Similar issues arise for currents or other sources when the law is intended as a fundamental vacuum relation. Such laws are mathematically well-defined on the physical domain but are structurally inconsistent with how fundamental interactions are usually modeled, where symmetry and conservation are closely related (Goldstein et al., 2001).

Level 2 is naturally suited for probing sensitivity to **minor but conceptually meaningful structural violations**, e.g., whether a model exploits action–reaction and reciprocity rather than only local curve fitting.

**Level 3: Counterfactual Laws with Major Structural Violation**

A shifted law is assigned to **Level 3** if it introduces at least one of the following:

1. **New singularities at physically regular points.** The original law is finite (often zero) at some physically natural limit—for example,
   - $\Delta T \to 0$ in calorimetry or heat flow,
   - $A \to 0$ (vanishing cross-section),
   - regular incidence angles in optical or wave laws,

   but the shifted law diverges there (e.g., terms proportional to $1/\Delta T^p$ or $1/A^p$). This conflicts with standard requirements that constitutive equations yield bounded responses for small driving forces and bounded entropy production in near-equilibrium regimes (Coleman & Noll, 1961; Mueller & Ruggeri, 2013; Gurtin et al., 2010).

2. **Generic loss of reality or positivity.** The shifted law yields negative, undefined, or non-real values for quantities that must remain non-negative and real (such as intensities, occupation numbers, or probabilities) at finite, interior parameter values, not just at well-understood singular limits of the original formulation.

Such laws are typically regarded as *inadmissible* in real-world physical modeling, because they violate minimal regularity or positivity requirements in a way that cannot be removed without discarding a substantial part of the intended physical domain.

Level 3 is suitable for **challenging and interactive symbolic regression**, and for **stress-testing generalization and counterfactual reasoning** under clearly implausible but algebraically well-specified distortions.

Table 6: Summary of physical plausibility levels.

| Aspect | Level 1 – Admissible Counterfactual Laws | Level 2 – Counterfactual Laws with Minor Structural Violation | Level 3 – Counterfactual Laws with Major Structural Violation |
|---|---|---|---|
| Singularity | No new singularities. | No new singularities. | New blow-ups at points where the original law is finite. |
| Reality / positivity | Real outputs; non-negative where required. | Real outputs; non-negative where required. | Generic loss of reality or positivity at finite parameter values. |
| Structural principles | Consistent with natural symmetry/conservation patterns and constitutive admissibility. | Breaks key symmetry/reciprocity/asymptotics (e.g., action–reaction), but remains regular. | Incompatible with minimal admissibility (e.g., infinite response to vanishing drive). |
| Role in experiments | Strictly plausible counterfactual physics. | Tests sensitivity to structural constraints. | Hard counterfactuals for robustness and generalization stress tests. |

To illustrate the challenge imposed by these categorization levels, Figure 7 presents the overall performance across the three plausibility levels. We report the Symbolic Accuracy (SA) and Root Mean Squared Logarithmic Error (RMSLE) across different equation difficulties and plausibility levels for both Vanilla Agent and Agent with Code Assistance settings.

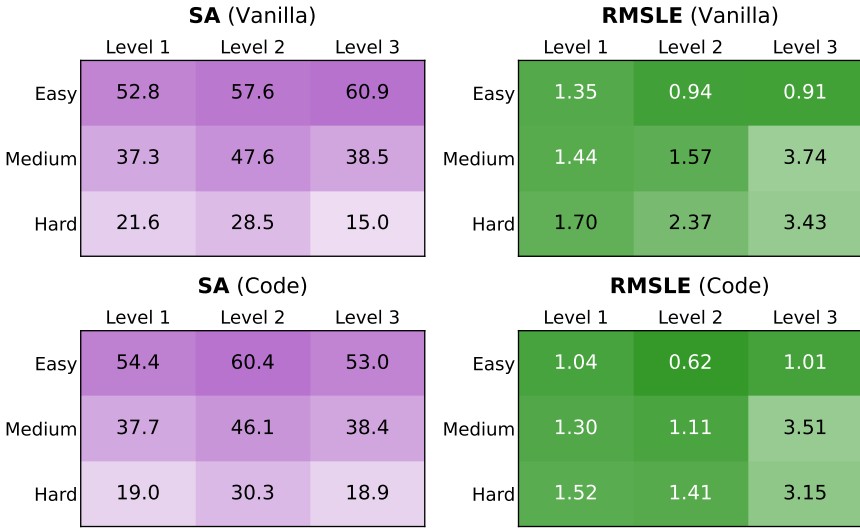

Figure 7: Overall Performance Across Physical Plausibility Levels.

Table 7: Full categorization on physical plausibility level for all 108 shifted laws.

| Plausibility Level | Shifted Equations Included in this Level |
|---|---|

**Level 1**

**Newton's Law of Universal Gravitation** (original $F = G\frac{m_1 m_2}{r^2}$)

$F = G'\frac{m_1 m_2}{r^{1.5}}$; $\quad F = G'\frac{(m_1 m_2)^2}{r^{1.5}}$; $\quad F = G'\frac{(m_1 + m_2)^2}{r^{1.5}}$; $\quad F = G'\frac{m_1^2 m_2^2}{r^2}$.

**Coulomb's Law** (original $F = k\frac{q_1 q_2}{r^2}$)

$F = k'\frac{q_1 q_2}{r^3}$; $\quad F = k'\frac{q_1 q_2(q_1 + q_2)}{r^2}$; $\quad F = k'\frac{q_1 q_2(q_1 + q_2)}{r^e}$; $\quad F = k'\frac{(q_1 q_2)^3}{r^2}$; $\quad F = k'\frac{(q_1 + q_2)^3}{r^2}$.

**Ampère's Force Law** (original $F = K\frac{I_1 I_2}{r}$)

$F = K'\frac{I_1 I_2}{r^3}$; $\quad F = K'\frac{(I_1 I_2)^{1.5}}{r^3}$; $\quad F = K'\frac{(I_1 + I_2)^{1.5}}{r^3}$; $\quad F = K'\frac{(I_1 I_2)^2}{r}$.

**Fourier's Law of Heat Conduction** (original $Q = k\frac{A\Delta T}{d}$)

$Q = k'\frac{A\Delta T}{d^2}$; $\quad Q = k'\frac{A^{0.5}\Delta T}{d^2}$; $\quad Q = k'\frac{A^{0.5}(\Delta T)^{1.3}}{d^2}$; $\quad Q = k'\frac{A^{0.5}\Delta T}{d}$; $\quad Q = k'\frac{A^{0.5}(\Delta T)^{2.7}}{d}$; $\quad Q = k'\frac{A^{0.5}(\Delta T)^{2.7}}{d^{3/7}}$; $\quad Q = k'\frac{A(\Delta T)^2}{d}$.

**Snell's Law** (original $\theta_2 = \sin^{-1}\left(\frac{n_1 \sin\theta_1}{n_2}\right)$)

$\theta_2 = \cos^{-1}\left(\frac{n_1 \sin\theta_1}{n_2}\right)$; $\quad \theta_2 = \cos^{-1}\left(\frac{n_1 \cos\theta_1}{n_2}\right)$; $\quad \theta_2 = \cos^{-1}\left(\frac{n_1^2 \cos\theta_1}{n_2}\right)$; $\quad \theta_2 = \tan^{-1}\left(\frac{n_1 \sin\theta_1}{n_2}\right)$; $\quad \theta_2 = \tan^{-1}\left(\frac{n_1 \tan\theta_1}{n_2}\right)$; $\quad \theta_2 = \tan^{-1}\left(\left(\frac{n_1}{n_2}\right)^2 \tan\theta_1\right)$.

**Radioactive Decay** (original $N(t) = N_0 e^{-\lambda t}$)

$N(t) = N_0 e^{-\lambda' t^{1.5}}$; $\quad N(t) = N_0^{1.2} e^{-\lambda' t^{1.5}}$; $\quad N(t) = N_0^{1.2} e^{-\lambda'^e t^{1.5}}$; $\quad N(t) = N_0 e^{-\lambda'^{1.5} t}$; $\quad N(t) = N_0^{1.4} e^{-\lambda'^{1.5} t}$; $\quad N(t) = N_0^{1.4} e^{-\lambda'^{1.5} t^e}$; $N(t) = N_0 e^{-(\lambda' t)^{1.5}}$; $\quad N(t) = N_0^{1.8} e^{-(\lambda' t)^{1.5}}$; $\quad N(t) = N_0^{1.8} e^{-(\lambda' t)^{e+1.5}}$.

**Damped Harmonic Motion** (original $\omega = \sqrt{\frac{k}{m} - \left(\frac{b}{2m}\right)^2}$)

$\omega = \sqrt{\frac{k'}{m} - \frac{b'}{2m}}$; $\quad \omega = \sqrt{\frac{k'}{m} - \frac{b'}{2m^2}}$; $\quad \omega = \left(\frac{k'}{m} - \frac{b'}{2m^2}\right)^{1.5}$; $\quad \omega = \left(\frac{k'}{m} - \left(\frac{b'}{2m}\right)^2\right)^2$; $\quad \omega = \left(\frac{k'}{m^2} - \left(\frac{b'}{2m}\right)^2\right)^2$; $\quad \omega = \left(k'm^2 - \left(\frac{b'}{2m}\right)^2\right)^2$; $\omega = \frac{k'}{m} - \left(\frac{b'}{2m}\right)^2$; $\quad \omega = \frac{k'}{m^{1.3}} - \left(\frac{b'}{2m}\right)^2$; $\quad \omega = \frac{k'}{m^{1.3}} - \left(\frac{b'}{2m}\right)^{0.7}$.

**Malus's Law** (original $I = I_0 \cos^2\theta$)

$I = I_0\left(\sin\theta + \cos\theta\right)^2$; $\quad I = I_0\left(2\sin\theta + \cos\theta\right)^2$; $\quad I = I_0\left(2\sin\theta + 1.5\cos\theta\right)^2$.

**Speed of Sound in an Ideal Gas** (original $v = \sqrt{\frac{\gamma RT}{M}}$)

$v = \sqrt{\frac{\gamma R'T^2}{M}}$; $\quad v = \sqrt{\frac{\gamma R'T^2}{M^{1.5}}}$; $\quad v = \sqrt{\frac{e^\gamma R'T^2}{M^{1.5}}}$; $\quad v = \frac{\gamma R'T}{M}$; $\quad v = \frac{\gamma R'T}{M^{\frac{1}{3}}}$; $\quad v = \frac{\ln(\gamma)R'T}{M^{\frac{1}{3}}}$; $v = \sqrt{\frac{R'T}{M}}$; $\quad v = \sqrt{R'TM^{1.5}}$.

**Hooke's Law** (original $F = kx$)

$F = 2k_1'x^2$; $\quad F = 2k_1'x^2 + k_2'x$; $\quad F = 2k_1'x^{0.5}$; $\quad F = 2k_1'x^{0.5} + k_2'x^3$; $\quad F = 2k_1'x^{3.4}$; $\quad F = 2k_1'x^{3.4} + k_2'x^{0.5}$.

**Bose–Einstein Distribution** (original $n = \frac{1}{e^{(C\omega/T)} - 1}$)

$n = \frac{1}{e^{(C'\omega/T)} + 1}$; $\quad n = \frac{1}{e^{(C'\omega^{1.5}/T)} + 1}$; $\quad n = \frac{1}{e^{(C'\omega^{1.5}/T^2)} + 1}$; $\quad n = \frac{1}{e^{(C'\omega^{0.5}/T)} - 1}$; $\quad n = \frac{1}{e^{(C'\omega/T^3)} - 1}$; $\quad n = \frac{1}{e^{(C'\omega^{1.5}/T^3)} - 1}$.

**Law of Heat Transfer (Calorimetry)** (original $Q = mc\Delta T$)

$Q = mc'(\Delta T)^{2.5}$; $\quad Q = m^{2.5}c'(\Delta T)$; $\quad Q = (m\Delta T)^{2.5}c'$.

**Level 2**

**Newton's Law of Universal Gravitation**

$F = G'\frac{m_1}{r^2}$; $\quad F = G'\frac{m_1}{r^{2.6}}$; $\quad F = G'\frac{m_1^{1.3}}{r^{2.6}}$; $\quad F = G'm_1^2 m_2^2 r^2$; $\quad F = G'\left(m_1^2 + m_2^2\right)r^2$.

**Coulomb's Law**

$F = k'\frac{q_2^2(q_1 + q_2)^3}{r^2}$; $\quad F = k'\frac{q_1^3 q_2}{r^2}$; $\quad F = k'\frac{q_1^3 q_2^2}{r^{2.5}}$; $\quad F = k'\frac{q_1^3 q_2^2}{r^e}$.

**Ampère's Force Law**

$F = K'(I_1 I_2)^2 r$; $\quad F = K'(I_1 - I_2)^2 r$; $\quad F = K'\frac{I_2}{r}$; $\quad F = K'\frac{I_2}{r^{3.8}}$; $\quad F = K'\frac{I_2^{0.9}}{r^{3.8}}$.

**Level 3**

**Fourier's Law of Heat Conduction**

$Q = k'\frac{(\Delta T)^2}{d\,A^{3.4}}$; $\quad Q = k'\frac{(\Delta T)^2}{\sqrt{d}\,A^{3.4}}$.

**Snell's Law**

$\theta_2 = \sin^{-1}\left(\frac{n_2 \sin\theta_1}{n_1}\right)$; $\quad \theta_2 = \cos^{-1}\left(\frac{n_2 \sin\theta_1}{n_1}\right)$; $\quad \theta_2 = \cos^{-1}\left(\frac{n_2 \sin\theta_1}{n_1^{2.5}}\right)$.

**Malus's Law**

$I = I_0\left(\frac{\sin\theta}{\cos\theta}\right)^2$; $\quad I = I_0\frac{\sin^2\theta}{\cos^3\theta}$; $\quad I = I_0\left(\frac{\sin^2\theta}{\cos^3\theta}\right)^e$; $\quad I = I_0\left(\frac{\cos\theta}{\sin\theta}\right)^2$; $\quad I = I_0\left(\frac{\cos\theta}{\sin\theta}\right)^e$; $\quad I = I_0\left(\frac{\sin^2\theta}{\cos\theta}\right)^e$.

**Speed of Sound in an Ideal Gas**

$v = (R'TM^{1.5})^{-2.8}$.

**Hooke's Law**

$F = 2k_1'x^2 + k_2'x + k_3'x^{-0.5}$; $\quad F = 2k_1'x^{0.5} + k_2'x^3 + k_3'x^{-0.3}$; $\quad F = 2k_1'x^{3.4} + k_2'x^{0.5} + k_3'x^{-10/3}$.

**Bose–Einstein Distribution**

$n = \frac{1}{e^{(C'\omega^{0.5}T)} - 1}$; $\quad n = \frac{1}{e^{(C'\omega^{0.5}T^{2.3})} - 1}$; $\quad n = \frac{1}{-\ln\left(\frac{C'\omega^{1.5}}{T^3}\right) - 1}$.

**Law of Heat Transfer (Calorimetry)**

$Q = \frac{c'}{m(\Delta T)^{2.5}}$; $\quad Q = \frac{c'}{m^{e+1}(\Delta T)^{2.5}}$; $\quad Q = \frac{c'}{m^{2.5}(\Delta T)}$; $\quad Q = \frac{c'}{m^{2.5}(\Delta T)^{e+1}}$; $\quad Q = \frac{c'}{m^{2.5}(\Delta T)^{2.5}}$; $\quad Q = \frac{c'}{(m\Delta T)^{e+1}}$.

## A.4 EVALUATION DETAILS

### A.4.1 SYMBOLIC ACCURACY EVALUATION DETAILS

We applied the LLM-as-a-Judge framework to verify the symbolic equivalence between the submitted law and the ground-truth law, utilizing prompt engineering and few-shot demonstrations (details provided below). We evaluated the effectiveness of LLM-as-a-Judge across four LLMs using a human-labeled dataset of 120 pairs of equations (results shown in Table 8). Based on performance and open-source accessibility, we ultimately selected Nemotron-ultra-253b for our experiments. We will publicly release the judge evaluation data to guarantee reproducibility.

Table 8: Agreement (%) between LLM-as-a-Judge and human evaluator.

| Models | Agreement |
|---|---|
| Nemotron-nano-8b | 83.3 |
| GPT-oss-20b | 90.8 |
| GPT-4.1 | **98.3** |
| Nemotron-ultra-253b | **98.3** |

---

**LLM-as-a-Judge Prompt**

```
You are a mathematical judge. Your task is to determine if two equations are equivalent.

**Instructions:**
1. Compare the two equations carefully
2. Consider algebraic manipulations, variable reordering, and variable renaming
3. Determine if they represent the same mathematical relationship
4. Provide your reasoning step by step first, and then provide only one answer
   under the format of **Answer: YES/NO**
5. Try converting both equations into the same algebraic form to make comparison easier.
   - e.g. rewrite ln(x ** 2) into 2ln(x)

**Output format:**
Reasoning: (Your reasoning steps)
Answer: (YES/NO)

**Reminder:**
- Equations may be expressed in standard mathematical notation or as Python code.
  If the Python implementation implies the same mathematical relationship,
  the equations are considered equivalent.
- Constants may differ in form or value. As long as they serve the same functional role
  (e.g., both scale the output proportionally), they are considered interchangeable.
  For example, a constant expressed as sqrt(k) in one equation and as c in another may be
  equivalent if both affect the output in the same way and can be interchangeable by
  selecting suitable value for the constant
- Variable names may differ, but the index and structure of variables must
  match exactly for the equations to be considered equivalent.
  For example, index of 4 and 4.03 are considered different
- YES/NO must be on the same line as "Answer:"

**Examples:** <provided in the next page>

**Your Task:**
Compare these two equations and determine if they are equivalent:

Parameter Descriptions:
{param_description}

Equation 1: {equation1}

Equation 2: {equation2}
```

---

**LLM-as-a-Judge Prompt (Few-shot Examples)**

```
**Examples:**

Equation 1: (HIDDEN_CONSTANT_C * x1 * x2) ** 2 / x3 ** 2
Equation 2: def discovered_law(x1, x2, x3):
   C = 6.7e-05
   return (C * (x1 * x2) ** 2) / x3 ** 2
Reasoning: Although the constant in equation 1 is HIDDEN_CONSTANT_C**2 and constant
           in equation 2 is C, both constant serve the same scaling role ......
Answer: YES

Equation 1: (C * x1 * x2) / x3 ** 2
Equation 2: def discovered_law(x1, x2, x3):
   C = 6.7e-05
   return (C * x1) / (x3 ** 4 * x2)
Reasoning: The second equation changes the exponent on x3 and
           alters the position of x2 ......
Answer: NO

Equation 1: sqrt(C * x1 * (x2 ** 2)) / x3 ** 2
Equation 2: def discovered_law(x1, x2, x3):
   C = 6.7e-05
   return sqrt(C * x1) * x2 / x3 ** 2
Reasoning: Since sqrt(x2 ** 2) = x2, both expressions represent
           the same mathematical relationship ......
Answer: YES

Equation 1: (G * x1 * x2) / x3 ** 2
Equation 2: def discovered_law(x1, x2, x3):
   C = 6.7e-05
   return (C * x1 * x2) / x3 ** 2.02
Reasoning: The exponent on x3 differs slightly ......
Answer: NO

Equation 1: (C * x1 * x2) / x3 ** 2
Equation 2: def discovered_law(x1, x2, x3):
   G = 6.7e-05
   product = x1 * x2
   return (G * product) / x3 ** 2
Reasoning: Variable naming differs but structure and operations are equivalent.
           G serves the same role as C ......
Answer: YES

Equation 1: C * ln(x ** 2)
Equation 2: def discovered_law(x1, x2, x3):
   G = 2.02
   return G * ln(x)
Reasoning: C * ln(x**2) is the same as 2C * ln(x) and the constant (2C)
           servers the same role as G ......
Answer: YES

Equation 1: (C * x1 * x2) / x3 ** 2
Equation 2: def discovered_law(x1, x2, x3):
   return (x1 * x2) / x3 ** 2
Reasoning: Equation 1 has a constant variable while Equation 2
           has a numerical constant of 1 ......
Answer: YES
```

### A.4.2 DATA FIDELITY EVALUATION DETAILS

For data fidelity evaluation, the submitted equations are assessed on a separate set of 5,000 data samples. Following Matsubara et al. (2024), we use the sampling distributions in Table 9. To ensure RMSLE is well-defined, equations are curated to yield non-zero results, and inputs are sampled so that all ground-truth outputs are non-negative. We further improve stability by applying outlier filtering using the *Modified Z-Score method* (Iglewicz & Hoaglin, 1993). The notion of physical constants is used only within the heuristic sampling-distribution setting for RMSLE evaluation and does not represent the actual physical concept of the constants in shifted laws.

Table 9: Sampling Distributions for Each Domain

| Domain | Equation | Variables and Descriptions | Sampling Distributions |
|---|---|---|---|
| Gravitation | $F = f(m_1, m_2, r)$ | m1: mass of the first object
m2: mass of the second object
r: distance between the two objects | $m1 : U_{log}(10^0, 10^3)$
$m2 : U_{log}(10^0, 10^3)$
$r : U_{log}(1, 10^1)$ |
| Coulomb's Law | $F = f(q_1, q_2, r)$ | q1: charge magnitude of the first particle
q2: charge magnitude of the second particle
r: distance between the particles | $q1 : U_{log}(10^{-1}, 10^1)$
$q2 : U_{log}(10^{-1}, 10^1)$
$r : U_{log}(10^{-1}, 10^1)$ |
| Ampere's Force Law | $F = f(I_1, I_2, r)$ | I1: current magnitude in the first wire
I2: current magnitude in the second wire
r: distance between the wires | $I1 : U_{log}(10^{-3}, 10^{-1})$
$I2 : U_{log}(10^{-3}, 10^{-1})$
$r : U_{log}(10^{-3}, 10^{-1})$ |
| Fourier's Law | $Q = f(k, A, \Delta T, d)$ | A: cross-sectional area
$\Delta T$: temperature difference
d: distance across material | $k : U_{log}(10^{-1}, 10^1)$
$A : U_{log}(10^{-4}, 10^{-2})$
$\Delta T : U_{log}(10^1, 10^3)$
$d : U_{log}(10^{-2}, 10^0)$ |
| Snell's Law | $\theta_2 = f(n_1, n_2, \theta_1)$ | n1: refractive index of medium 1
n2: refractive index of medium 2
$\theta_1$: angle of incidence | $n1 : U(1.0, 1.5)$
$n2 : U(1.0, 1.5)$
$\theta_1 : U(0, \pi/2)$ |
| Radioactive Decay | $N = f(N_0, \lambda, t)$ | $N_0$: initial number of atoms
t: time | $N_0 : U_{log}(10^0, 10^2)$
$\lambda : U_{log}(10^{-3}, 10^{-1})$
$t : U_{log}(10^{-2}, 10^1)$ |
| Harmonic Motion | $\omega = f(k, m, b)$ | m: mass | $k : U_{log}(10^2, 10^4)$
$m : U_{log}(10^{-1}, 10^1)$
$b : U_{log}(10^{-2}, 10^0)$ |
| Malus' Law | $I = f(I_0, \theta)$ | $I_0$: initial intensity of the light
$\theta$: angle between the polarization axis | $I_0 : U_{log}(100, 2000)$
$\theta : U(10^{-6}, \pi/2)$ |
| Acoustic Velocity | $v = f(\gamma, T, M)$ | $\gamma$: adiabatic index of the gas
T: temperature
M: molar mass | $\gamma : U(1.3, 1.7)$
$T : U_{log}(10^1, 10^3)$
$M : U_{log}(10^{-3}, 10^{-1})$ |
| Hooke's Law | $F = f(x)$ | x: displacement from the equilibrium | $x : U_{log}(10^{-3}, 10^0)$ |
| Bose-Einstein Distribution | $n = f(\omega, T)$ | $\omega$: angular frequency of the photons
T: temperature | $\omega : U_{log}(10^8, 10^{10})$
$T : U_{log}(10^1, 10^3)$ |
| Heat Transfer | $Q = f(m, c, \Delta T)$ | m: mass
$\Delta T$: temperature difference | $m : U_{log}(10^{-3}, 10^3)$
$c : U_{log}(10^2, 10^4)$
$\Delta T : U_{log}(10^1, 10^3)$ |

A.5    FURTHER DISCUSSIONS ON LLM-AS-A-JUDGE

**Style Robustness Analysis**    We investigated the performance of our LLM-as-a-Judge in assessing symbolic equivalence across laws expressed in diverse syntactic representations. The results of this robustness analysis are illustrated in Table 10.

Table 10: Style robustness analysis on LLM-as-a-judge.

| **Case: Exponent placement difference** | |
| --- | --- |
| **LLM Judge** | GPT-4.1 |
| **Ground Truth Law** | `math.sqrt(gamma * HIDDEN_CONSTANT_R * T ** 2 / M ** 1.5)` |
| **Testing Law** | `R_sim * ((T ** 4 * gamma ** 2) / (M ** 3)) ** (0.25)` |
| **Accuracy** | 100% |

| **Case: Operand order variation** | |
| --- | --- |
| **LLM Judge** | GPT-4.1 |
| **Ground Truth Law** | `math.degrees(math.acos(n1 ** 2 * math.cos(math.radians(angle1)) / n2))` |
| **Testing Law** | `val = (n1 * n1 / n2) * math.cos(math.radians(angle1))` `return math.degrees(math.acos(val))` |
| **Accuracy** | 100% |

| **Case: Log–power rearrangement** | |
| --- | --- |
| **LLM Judge** | GPT-4.1 |
| **Ground Truth Law** | `np.log(N0 ** 1.5) * np.exp(-lambda_decay + t ** 0.5)` |
| **Testing Law** | `1.5 * math.exp(math.sqrt(t) - lambda_decay) * math.log(N0)` |
| **Accuracy** | 100% |

The results demonstrate that the LLM-as-a-Judge consistently recognizes symbolic equivalence when laws exhibit substantial structural and syntactic variations. This robustness to stylistic differences ensures fair and reliable evaluation within our proposed benchmark.

**Error Analysis**  We analyzed the failure modes of our LLM-as-a-Judge and identified one false positive and one false negative when using GPT-4.1, yielding an overall accuracy of 98%. Table 11 provides the representative cases.

Table 11: Error case analysis on LLM-as-a-judge.

**Case: False Positive**

| | |
|---|---|
| **LLM Judge** | GPT-4.1 |
| **Ground Truth Law** | `math.log(N0 ** 1.5) *` `math.exp(-lambda_decay + math.sqrt(t))` |
| **Testing Law** | `math.log10(N0 ** 1.5) *` `math.exp(-lambda_decay + math.sqrt(t))` |
| **Accuracy** | 98% (1 case fails) |

**Case: False Negative**

| | |
|---|---|
| **LLM Judge** | GPT-4.1 |
| **Ground Truth Law** | `math.log(N0 ** 1.5) *` `math.exp(-lambda_decay + math.sqrt(t))` |
| **Testing Law** | `math.exp(math.log(1.5 * math.log(N0))` `- lambda_decay + math.sqrt(t))` |
| **Accuracy** | 98% (1 case fails) |

The false positive arises from conflating natural logarithm with base-10 logarithm. Specifically, the model appears to treat $\log(x)$ and $\log_{10}(x)$ as interchangeable up to a constant factor $(\log(x) = \log_{10}(x) \cdot \log(10))$ and erroneously "absorbs" this constant, leading to a spurious equivalence judgment. The false negative reflects difficulty with exponential–logarithmic cancellation in the presence of additive terms: although $\exp(\log(a) + b) = a \exp(b)$, the model failed to recognize that $\log(N_0^{1.5}) \exp(-\lambda_{\text{decay}} + \sqrt{t})$ equals $\exp(\log(1.5 \log N_0) - \lambda_{\text{decay}} + \sqrt{t})$ because $\log(N_0^{1.5}) = 1.5 \log N_0$. Techniques such as self-consistency/majority voting may reduce such errors, though the observed accuracy (**98%+**) is already high.

**Impact of Few-shot Learning**  We evaluated the effect of in-context demonstrations for LLM-as-a-Judge with GPT-4.1. Accuracy improves monotonically with more demonstrations (Table 12), underscoring the value of few-shot prompting.

Table 12: Impact of few-shot demonstrations on LLM-as-a-judge performance of GPT-4.1.

| Num. of Demos | Accuracy |
|---|---|
| 0-shot | 93.33% |
| 1-shot | 95.83% |
| 3-shot | 96.67% |
| 7-shot | 98.33% |

# B  FULL RESULTS

## B.1  RESULTS FOR INDIVIDUAL DOMAINS

In this section, we illustrate the detailed performances of all LLM Agents in each physics domains:

- Figure 8: Gravitation (Newton's Law of Universal Gravitation).
- Figure 9: Electrostatics (Coulomb's Law).
- Figure 10: Magnetostatics (Ampère's Force Law).
- Figure 11: Thermal Conduction (Fourier's Law).
- Figure 12: Geometrical Optics (Snell's Law).
- Figure 13: Nuclear Physics (Law of Radioactive Decay).
- Figure 14: Oscillations (Law of Damped Harmonic Motion).
- Figure 15: Physical Optics (Malus's Law).
- Figure 16: Acoustics (Law of Sound Speed in Ideal Gas).
- Figure 17: Elasticity (Hooke's Law).
- Figure 18: Statistical Mechanics (Bose-Einstein Distribution).
- Figure 19: Calorimetry (Law of Heat Transfer).

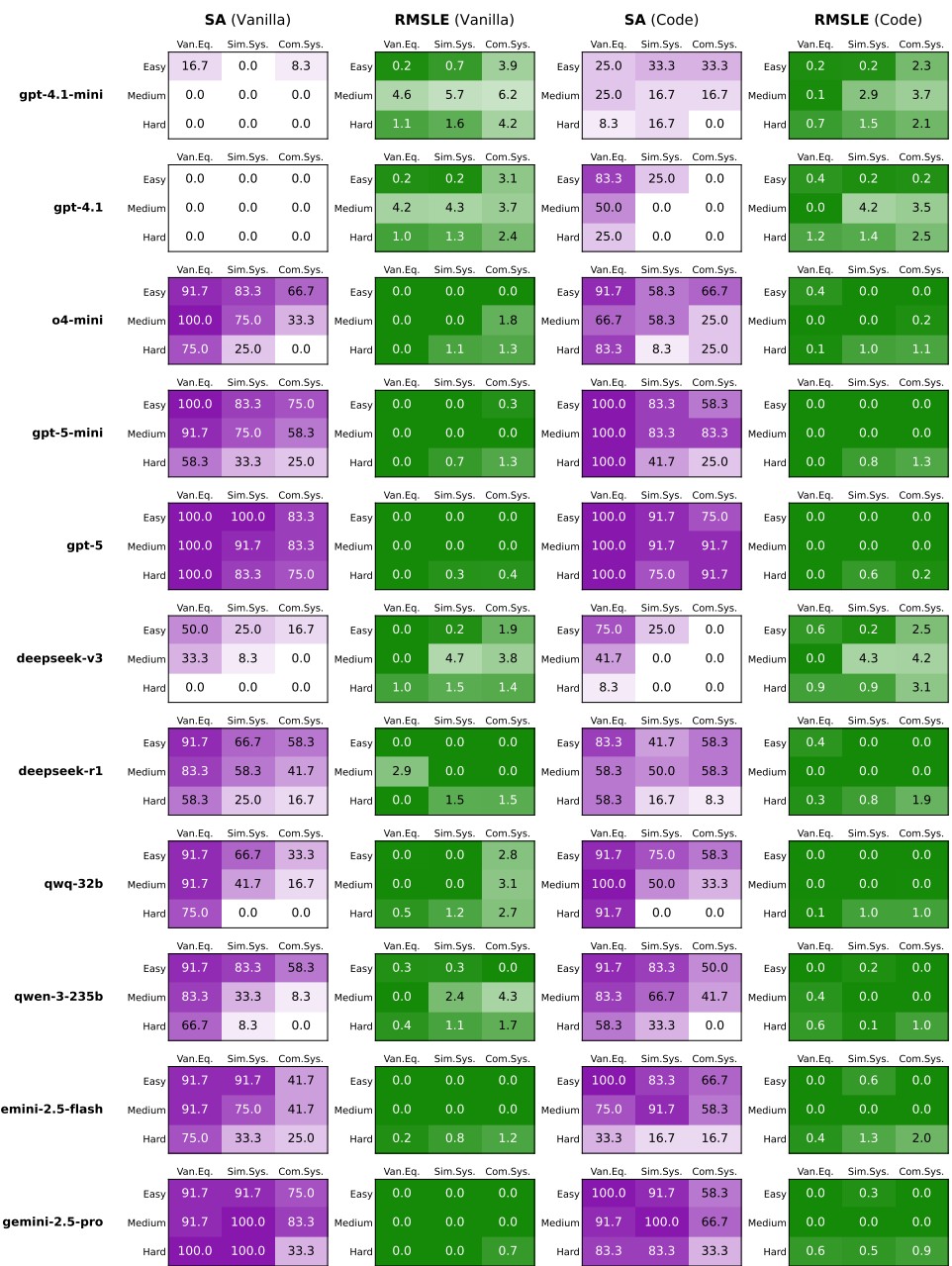

Figure 8: Full LLM Performances in domain Gravitation (Newton's Law of Universal Gravitation).

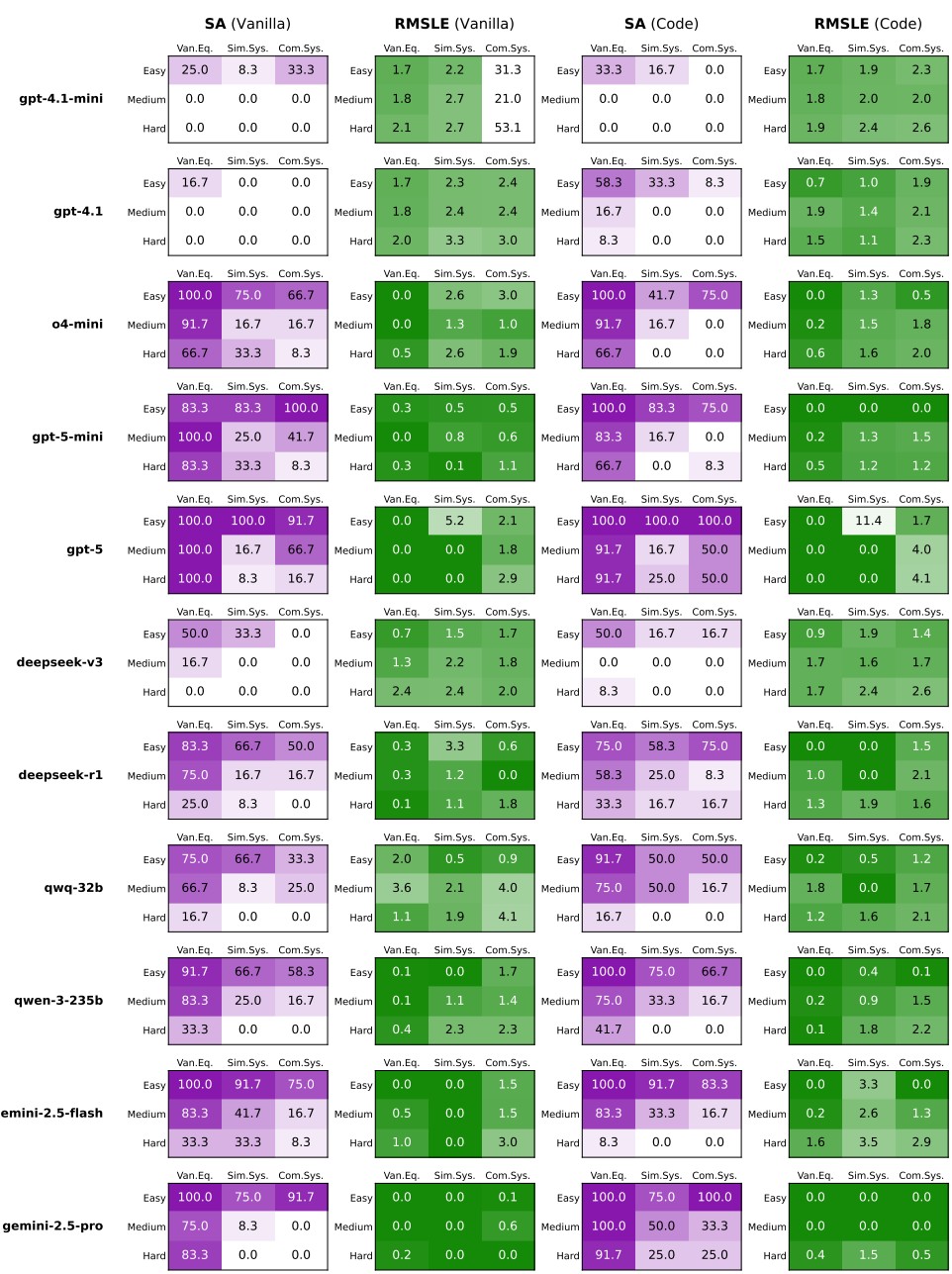

Figure 9: Full LLM Performances in domain Electrostatics (Coulomb's Law).

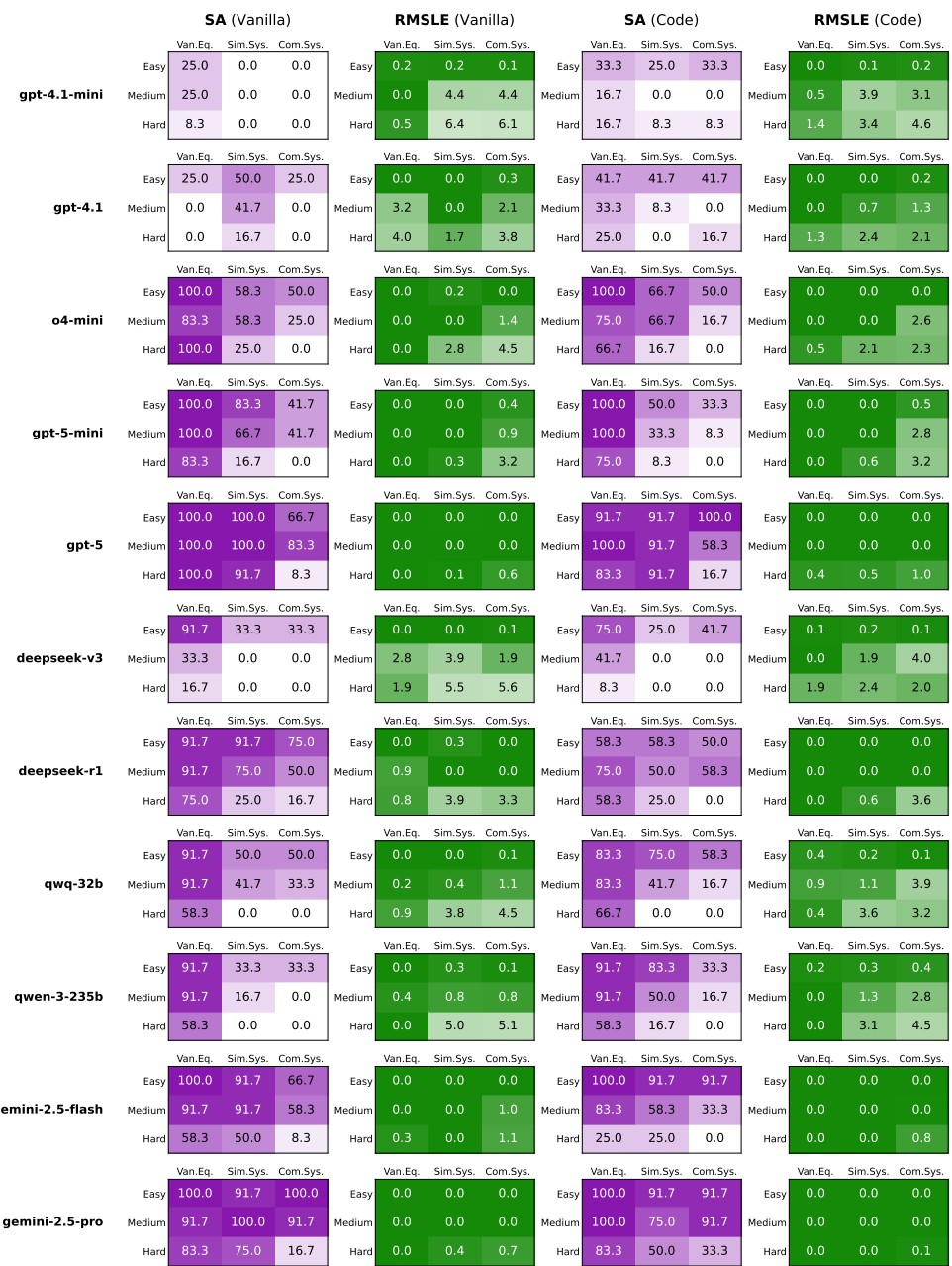

Figure 10: Full LLM Performances in domain Magnetostatics (Ampère's Force Law).

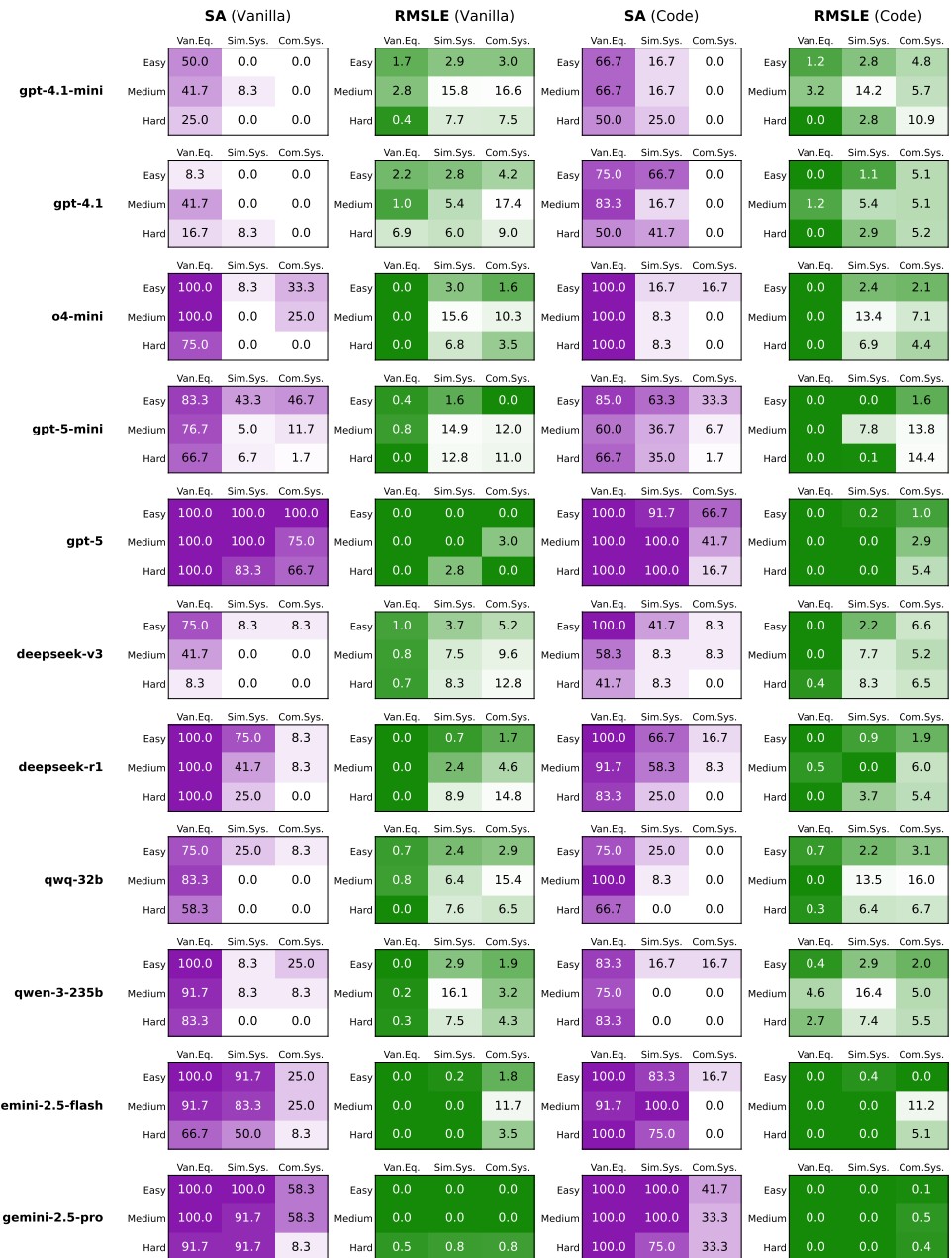

Figure 11: Full LLM Performances in domain Thermal Conduction (Fourier's Law).

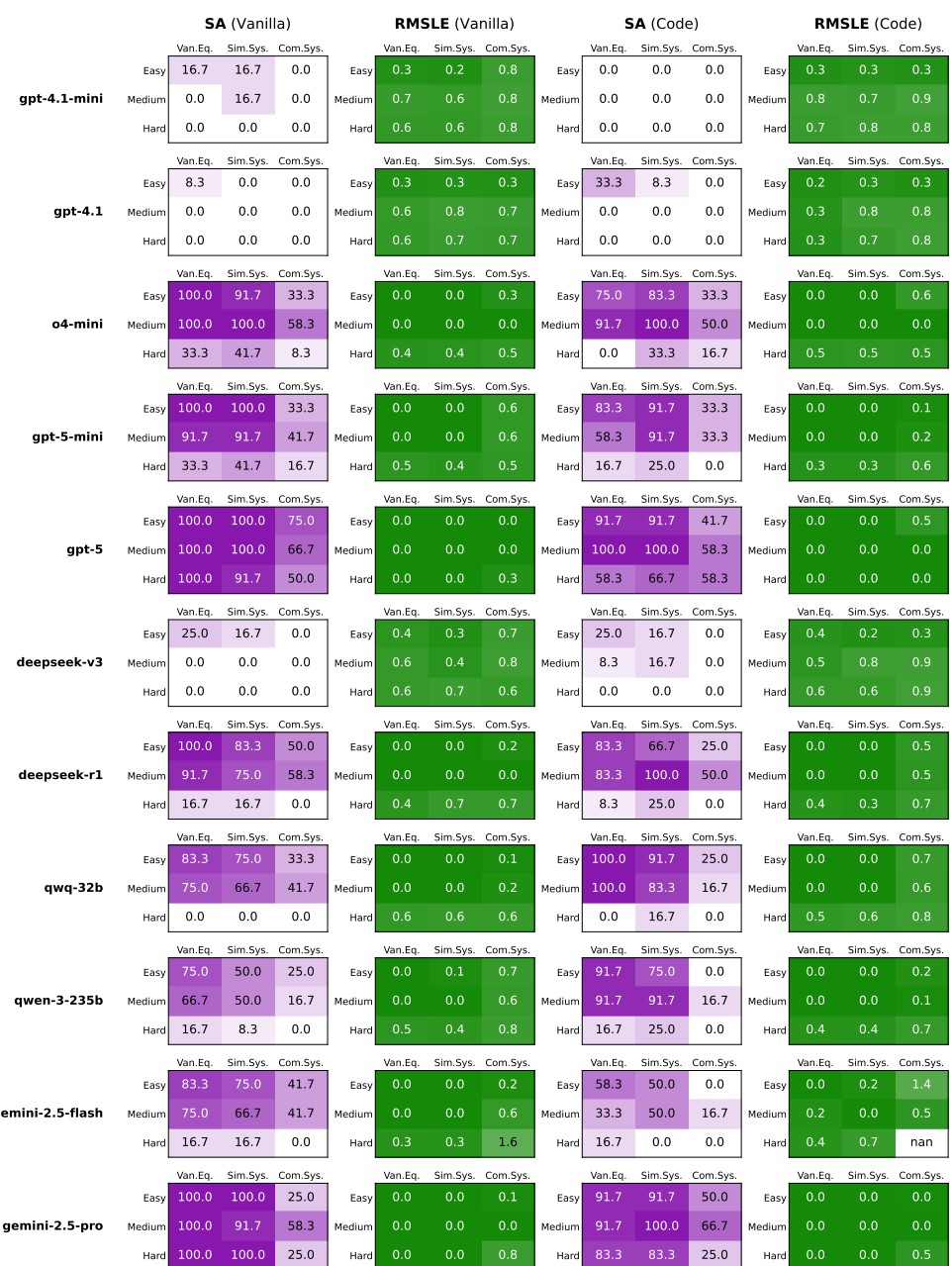

Figure 12: Full LLM Performances in domain Geometrical Optics (Snell's Law).

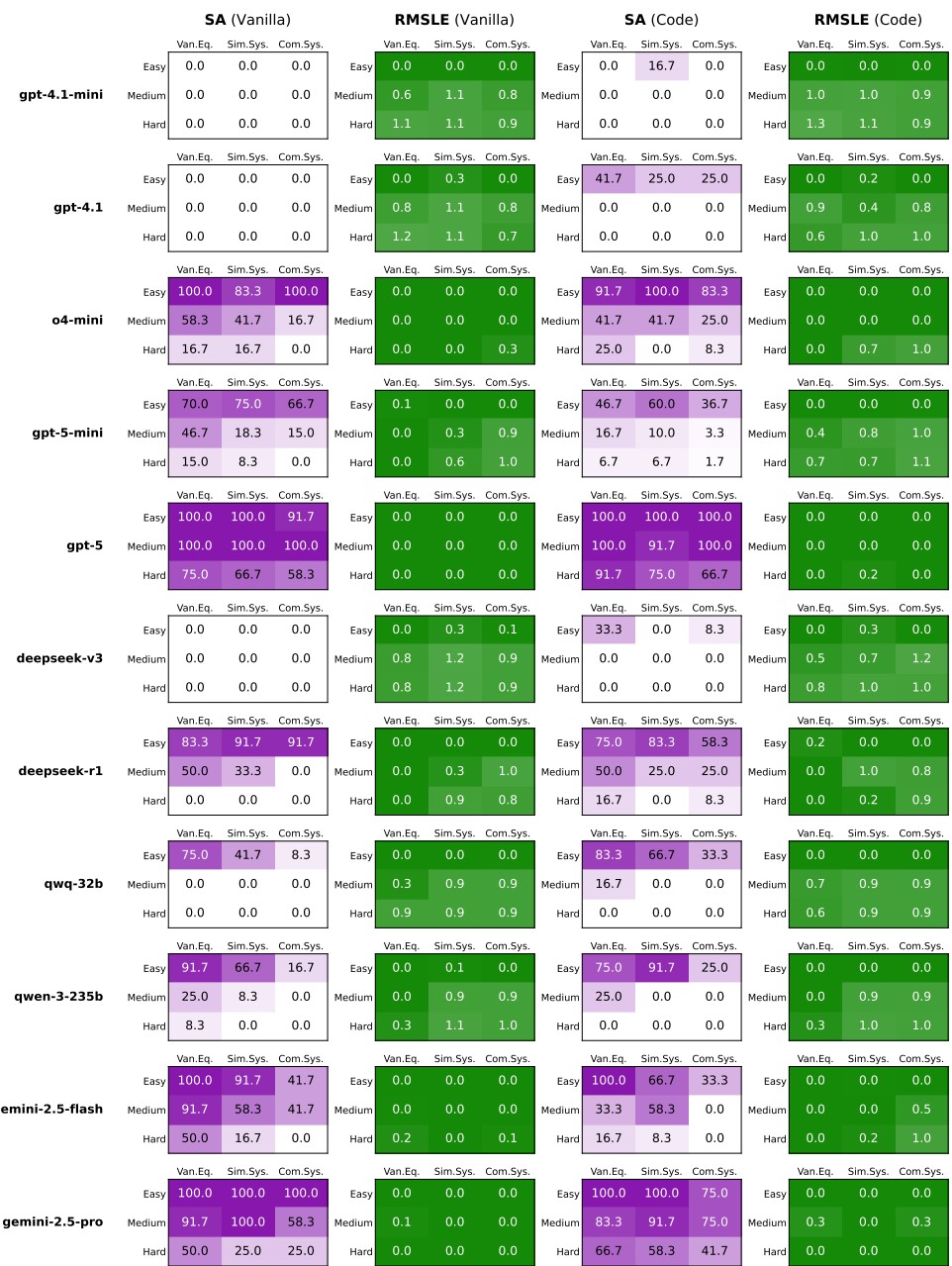

Figure 13: Full LLM Performances in domain Nuclear Physics (Law of Radioactive Decay).

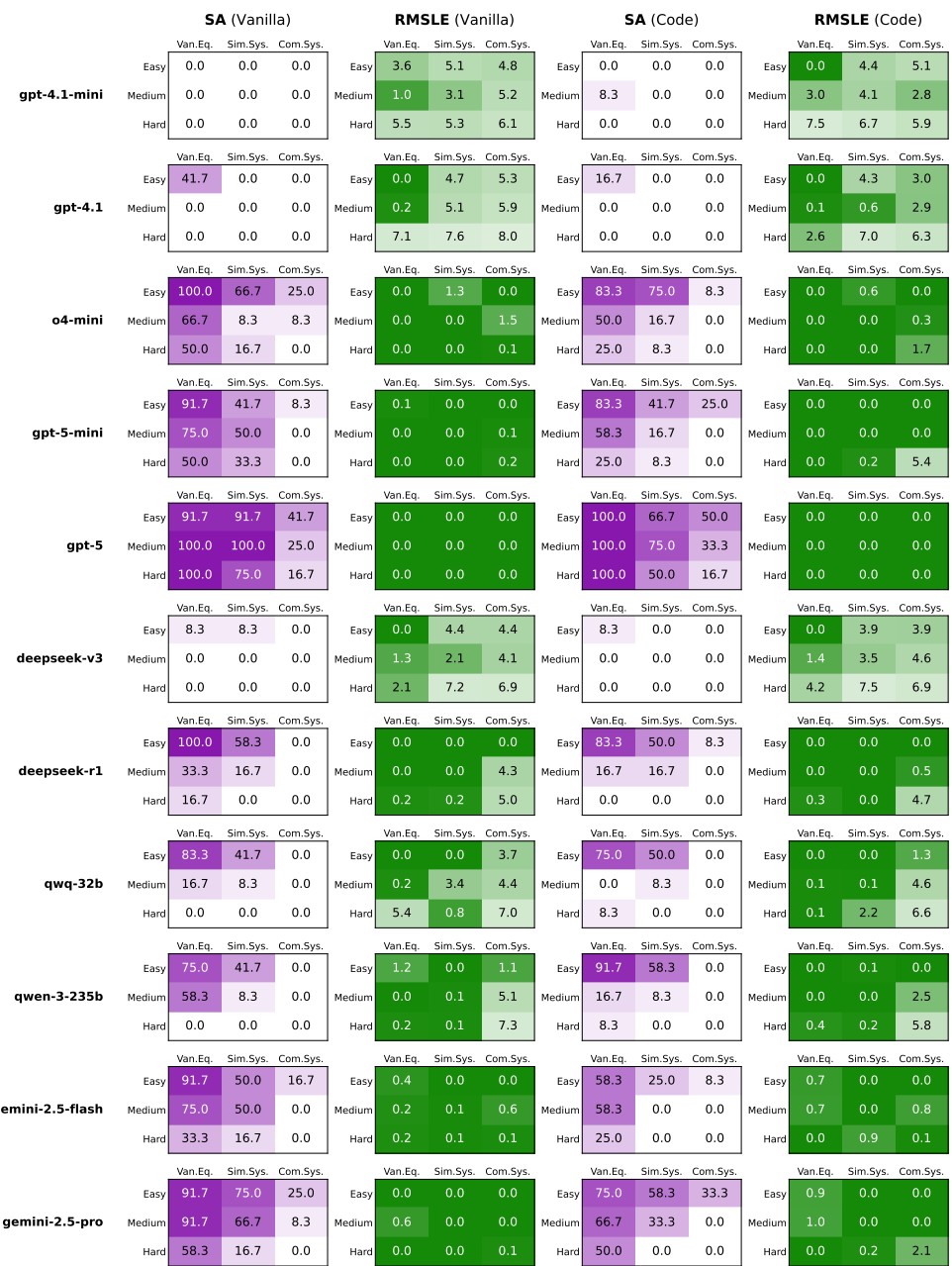

Figure 14: Full LLM Performances in domain Oscillations (Law of Damped Harmonic Motion).

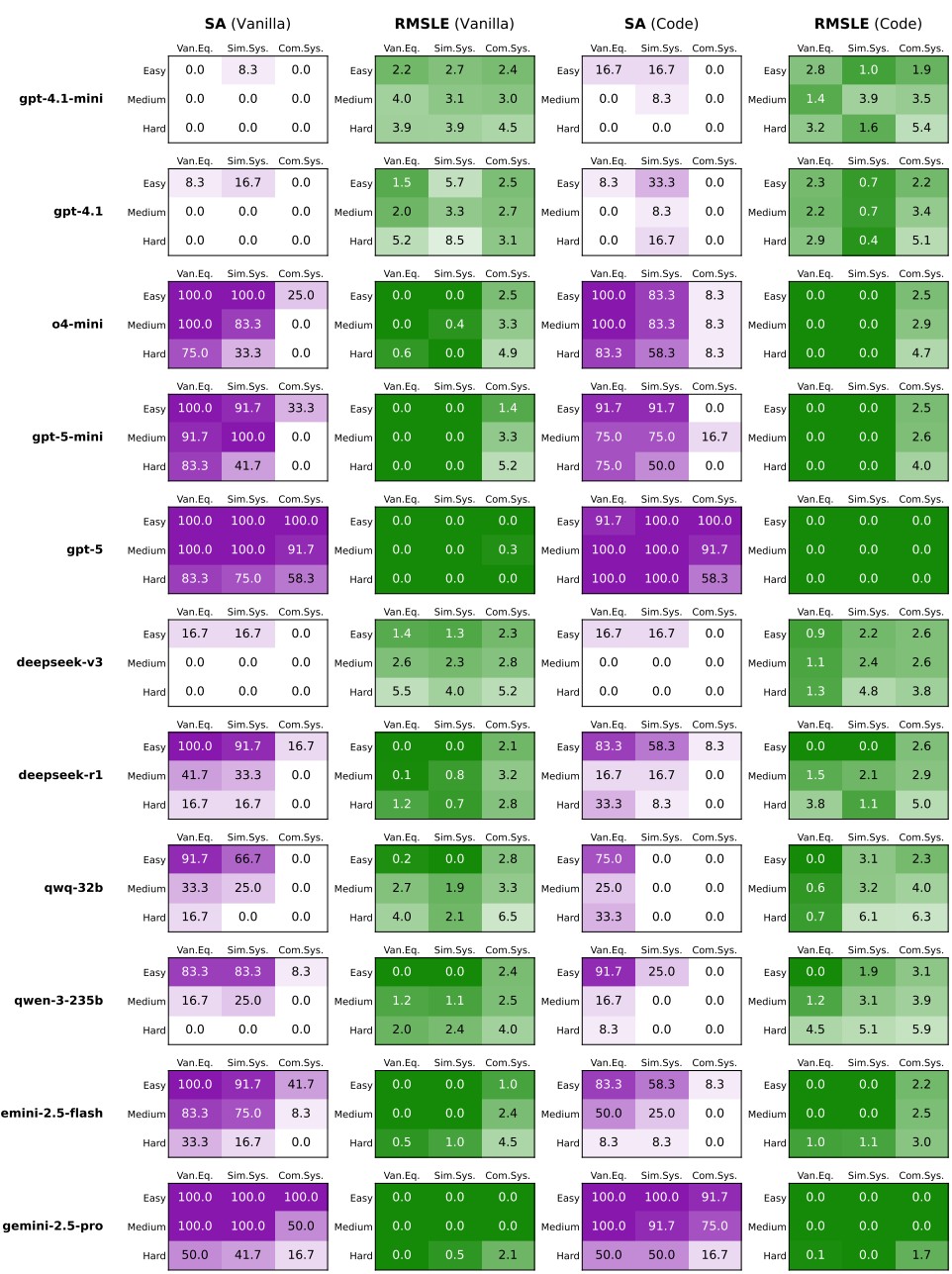

Figure 15: Full LLM Performances in domain Physical Optics (Malus's Law).

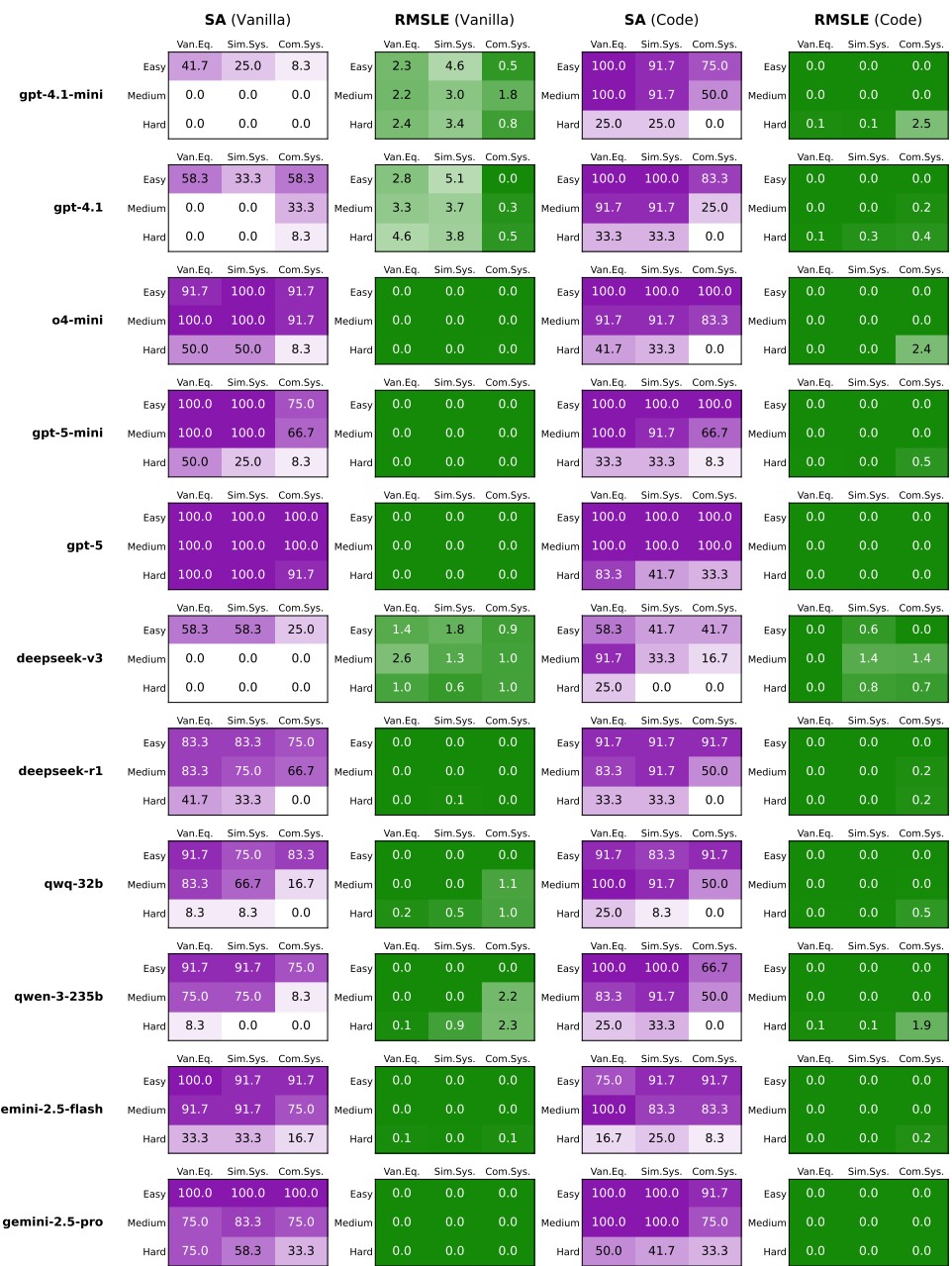

Figure 16: Full LLM Performances in domain Acoustics (Law of Sound Speed in Ideal Gas).

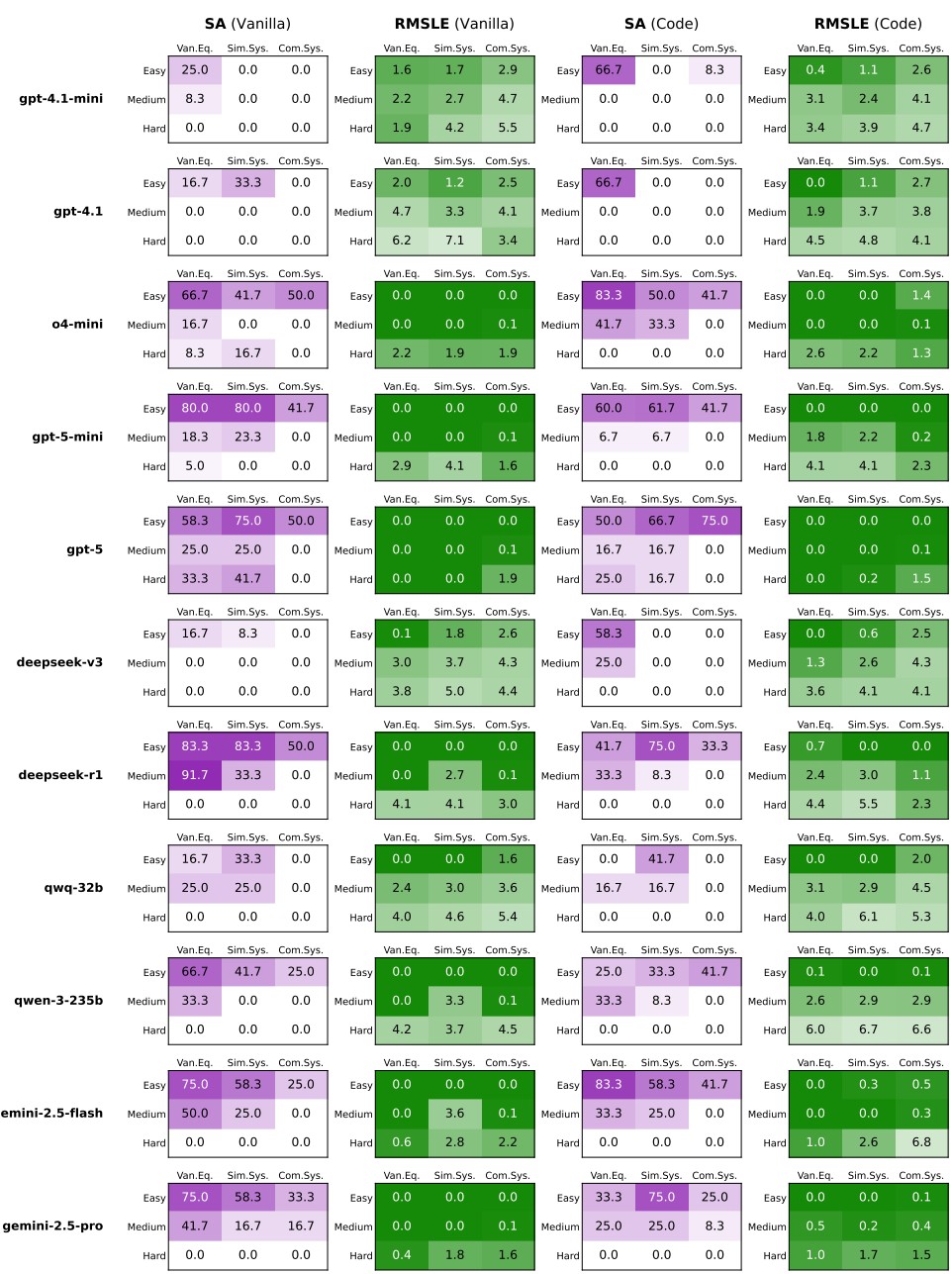

Figure 17: Full LLM Performances in domain Elasticity (Hooke's Law).

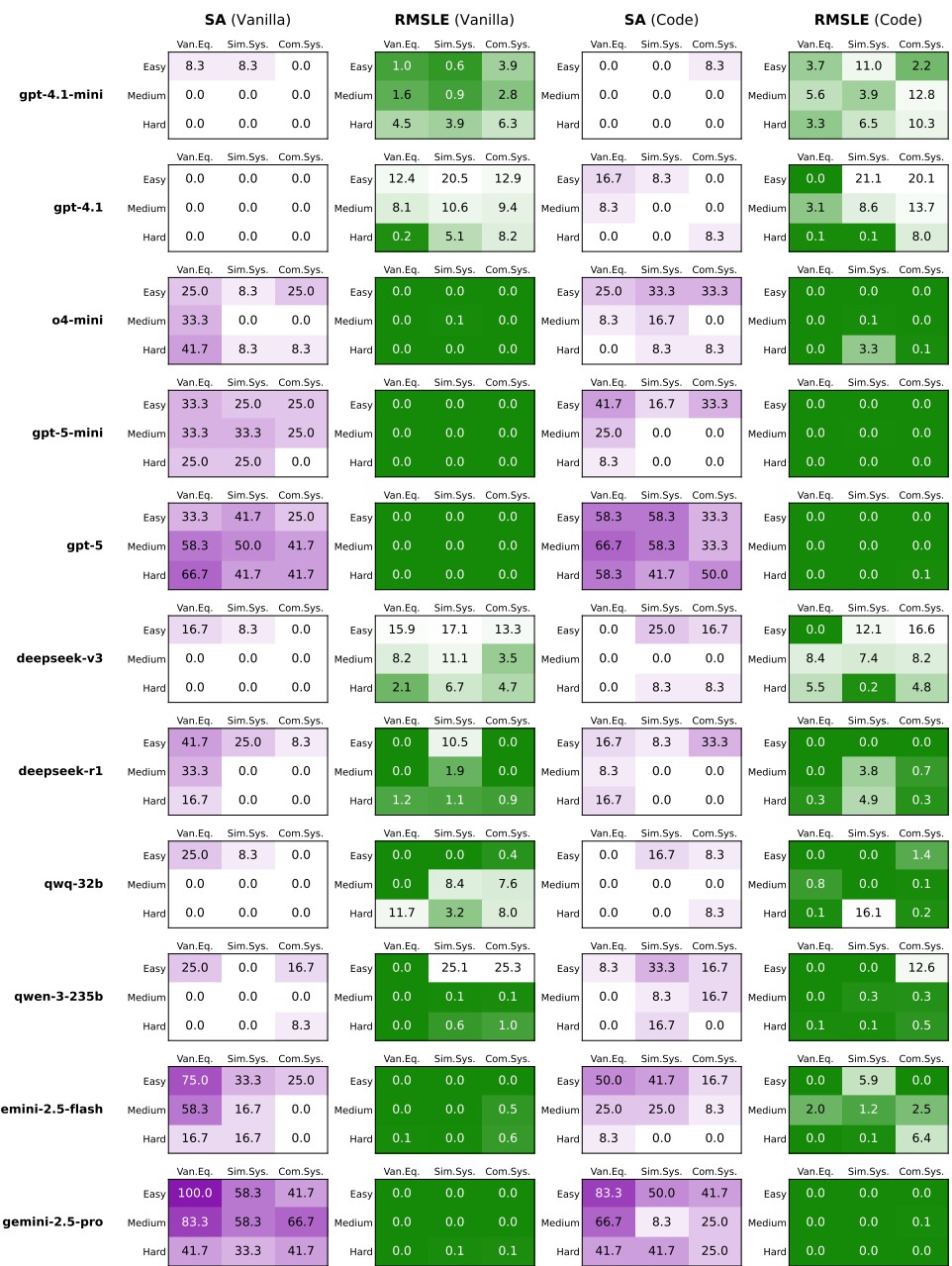

Figure 18: Full LLM Performances in domain Statistical Mechanics (Bose-Einstein Distribution).

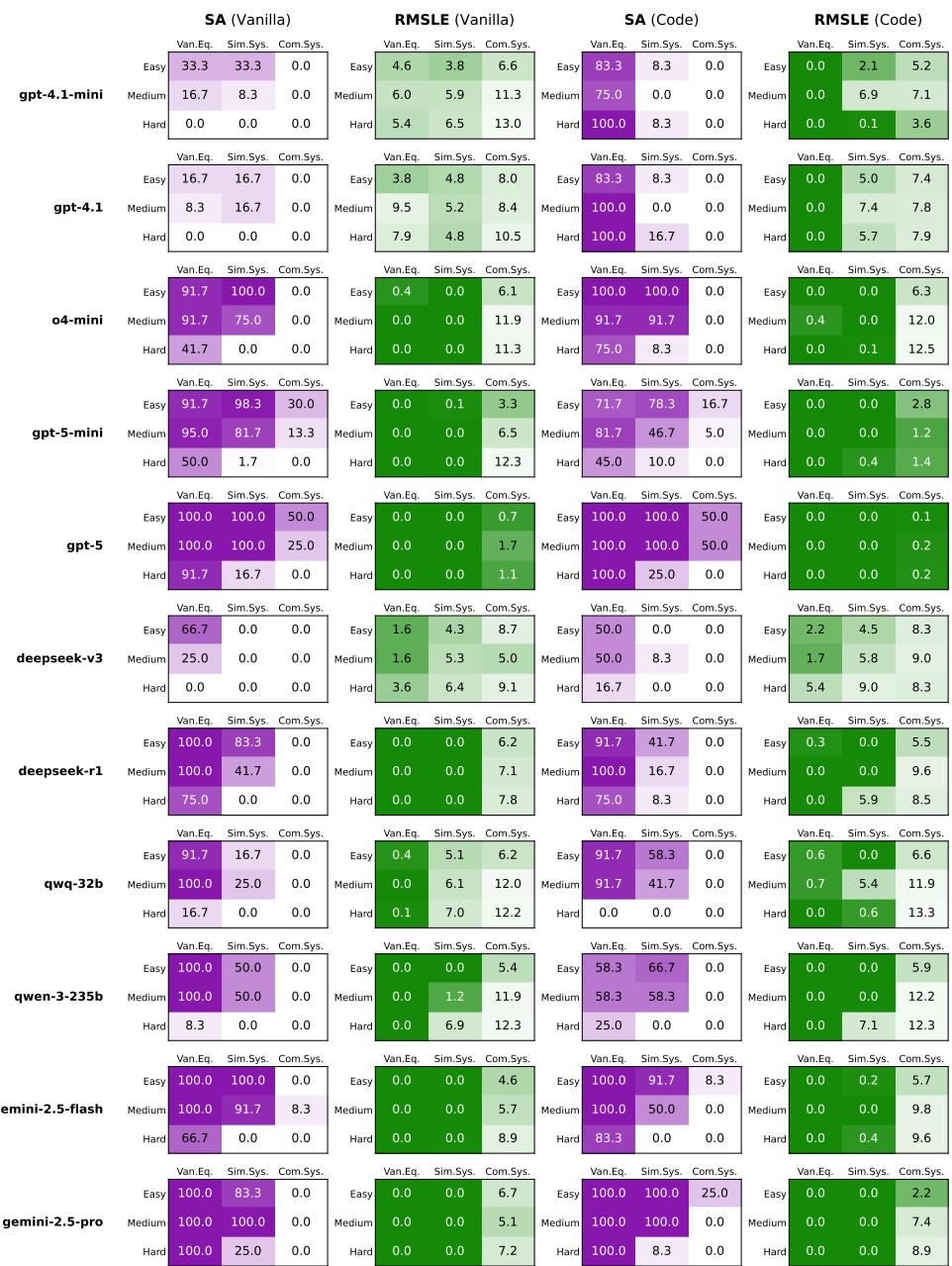

Figure 19: Full LLM Performances in domain Calorimetry (Law of Heat Transfer).

## B.2 RESULTS OF NON-PERFORMANCE METRICS

In this section, we present various non-performance metrics, evaluated across all difficulty levels and aggregated over 12 domains for each LLM Agent:

- Table 13: Average number of rounds
- Table 14: Average number of experiments (a set of parameters processed)
- Table 15: Average total tokens
- Table 16: Average number of tokens per round

Table 13: Average amount of rounds across all difficulty levels aggregated over all 12 domains. Within each agent settings, the highest numbers are marked as **bold**, with the second-highest underlined.

| LLM Agents | Vanilla Equation | | | Simple System | | | Complex System | | | Average |
|---|---|---|---|---|---|---|---|---|---|---|
| | easy | medium | hard | easy | medium | hard | easy | medium | hard | Rounds |
| *Vanilla Agent* | | | | | | | | | | |
| GPT-4.1-mini | 7.00 | 6.89 | 6.90 | 6.49 | 6.64 | 6.56 | 6.54 | 6.79 | 6.85 | 6.74 |
| GPT-4.1 | **7.89** | **8.59** | **8.63** | **7.22** | **7.42** | **7.35** | **6.99** | **7.12** | **7.19** | **7.60** |
| o4-mini | 2.74 | 2.97 | 3.53 | 2.85 | 3.21 | 3.82 | 3.23 | 3.58 | 4.25 | 3.35 |
| GPT-5-mini | 2.19 | 2.36 | 2.88 | 2.16 | 2.40 | 2.76 | 2.78 | 2.90 | 3.03 | 2.60 |
| GPT-5 | 2.41 | 2.38 | 2.60 | 2.60 | 2.65 | 3.19 | 3.13 | 3.40 | 3.89 | 2.92 |
| DeepSeek-V3 | 4.49 | 4.97 | 5.15 | 4.21 | 4.42 | 4.62 | 4.28 | 4.49 | 4.33 | 4.55 |
| DeepSeek-R1 | 2.92 | 3.24 | 3.85 | 3.15 | 3.35 | 3.84 | 3.73 | 3.56 | 4.04 | 3.52 |
| QwQ-32b | 2.80 | 2.97 | 3.64 | 2.75 | 2.72 | 3.17 | 2.74 | 2.46 | 2.95 | 2.91 |
| Qwen-3-235b | 3.22 | 3.38 | 3.62 | 3.12 | 3.26 | 3.40 | 4.01 | 3.85 | 4.04 | 3.54 |
| Gemini-2.5-flash | 3.99 | 4.67 | 5.43 | 4.43 | 4.62 | 5.51 | 5.47 | 5.65 | 6.33 | 5.12 |
| Gemini-2.5-pro | 2.99 | 3.06 | 3.08 | 3.00 | 2.92 | 3.07 | 3.36 | 3.16 | 3.47 | 3.12 |
| *Agent with Code Assistance* | | | | | | | | | | |
| GPT-4.1-mini | **8.10** | 8.56 | 8.77 | 8.52 | 8.62 | 8.90 | 8.61 | 8.94 | 8.94 | 8.66 |
| GPT-4.1 | 8.02 | **8.82** | **9.65** | **9.08** | **9.60** | **9.96** | **9.51** | **10.25** | **10.25** | **9.46** |
| o4-mini | 4.28 | 4.99 | 6.40 | 4.60 | 5.30 | 6.13 | 5.05 | 5.92 | 6.66 | 5.48 |
| GPT-5-mini | 4.01 | 4.83 | 5.51 | 4.61 | 5.25 | 5.91 | 5.17 | 5.51 | 5.94 | 5.19 |
| GPT-5 | 3.74 | 4.28 | 5.33 | 4.35 | 5.08 | 6.35 | 5.36 | 5.90 | 7.13 | 5.28 |
| DeepSeek-V3 | 6.93 | 7.77 | 7.53 | 6.62 | 7.04 | 6.86 | 6.90 | 6.92 | 7.03 | 7.07 |
| DeepSeek-R1 | 5.25 | 5.50 | 5.81 | 5.40 | 5.63 | 6.08 | 5.81 | 5.94 | 6.37 | 5.75 |
| QwQ-32b | 3.72 | 4.62 | 4.47 | 3.93 | 4.22 | 4.01 | 4.06 | 3.88 | 4.21 | 4.12 |
| Qwen-3-235b | 4.60 | 5.16 | 5.41 | 5.01 | 5.51 | 5.37 | 5.39 | 5.51 | 5.84 | 5.31 |
| Gemini-2.5-flash | 7.08 | 8.35 | 9.03 | 7.90 | 8.56 | 9.12 | 8.24 | 8.72 | 9.03 | 8.45 |
| Gemini-2.5-pro | 5.45 | 5.68 | 6.26 | 5.45 | 5.98 | 6.77 | 6.17 | 6.40 | 7.40 | 6.17 |

Table 14: Average Experiments across all difficulty levels aggregated over all 12 domains. Within each agent settings, the highest numbers are marked as **bold**, with the second-highest underlined.

| LLM Agents | Vanilla Equation | | | Simple System | | | Complex System | | | Average |
|---|---|---|---|---|---|---|---|---|---|---|
| | *easy* | *medium* | *hard* | *easy* | *medium* | *hard* | *easy* | *medium* | *hard* | Experiments |
| *Vanilla Agent* | | | | | | | | | | |
| GPT-4.1-mini | 44.78 | 42.48 | 41.40 | 32.37 | 33.89 | 32.07 | 35.49 | 36.64 | 37.27 | 37.38 |
| GPT-4.1 | **104.70** | **109.40** | **110.08** | **70.47** | **71.97** | **74.01** | **60.66** | **61.22** | **62.79** | **80.59** |
| o4-mini | 16.78 | 18.45 | 23.76 | 13.21 | 15.66 | 21.15 | 14.97 | 17.13 | 21.22 | 18.04 |
| GPT-5-mini | 19.05 | 20.60 | 27.07 | 14.42 | 17.96 | 21.96 | 21.38 | 22.61 | 24.52 | 21.06 |
| GPT-5 | 24.29 | 23.99 | 26.74 | 23.18 | 23.35 | 31.82 | 32.72 | 35.74 | 41.39 | 29.25 |
| DeepSeek-V3 | 21.40 | 23.72 | 24.53 | 14.99 | 15.99 | 17.41 | 14.63 | 15.81 | 15.28 | 18.20 |
| DeepSeek-R1 | 21.63 | 23.70 | 27.12 | 17.26 | 18.81 | 21.44 | 17.81 | 16.96 | 18.00 | 20.30 |
| QwQ-32b | 12.94 | 13.02 | 12.28 | 7.76 | 7.60 | 9.74 | 6.77 | 5.74 | 7.48 | 9.26 |
| Qwen-3-235b | 9.01 | 9.95 | 10.60 | 6.90 | 6.83 | 6.97 | 7.81 | 7.47 | 8.08 | 8.18 |
| Gemini-2.5-flash | 22.44 | 25.52 | 29.17 | 20.19 | 21.35 | 23.47 | 21.08 | 22.65 | 24.81 | 23.41 |
| Gemini-2.5-pro | 15.90 | 16.01 | 16.94 | 13.52 | 13.51 | 14.17 | 15.98 | 15.46 | 15.84 | 15.26 |
| *Agent with Code Assistance* | | | | | | | | | | |
| GPT-4.1-mini | 21.03 | 20.83 | 24.00 | 15.27 | 14.62 | 15.17 | 16.45 | 16.08 | 18.12 | 17.95 |
| GPT-4.1 | **31.55** | **34.74** | **39.69** | **30.48** | **29.69** | **31.41** | 25.84 | **27.52** | 27.19 | **30.90** |
| o4-mini | 13.61 | 14.85 | 18.60 | 10.92 | 11.96 | 14.98 | 11.08 | 12.97 | 15.24 | 13.80 |
| GPT-5-mini | 16.39 | 20.12 | 24.03 | 15.47 | 18.44 | 21.84 | 17.51 | 19.81 | 23.70 | 19.70 |
| GPT-5 | 20.59 | 21.20 | 24.21 | 20.06 | 21.94 | 26.08 | **26.42** | 27.37 | **33.02** | 24.54 |
| DeepSeek-V3 | 14.11 | 15.74 | 16.28 | 10.61 | 11.52 | 11.90 | 10.41 | 10.98 | 11.15 | 12.52 |
| DeepSeek-R1 | 12.49 | 15.01 | 16.60 | 11.10 | 12.00 | 12.94 | 10.92 | 11.49 | 10.35 | 12.54 |
| QwQ-32b | 9.03 | 10.27 | 9.81 | 6.35 | 6.01 | 6.22 | 5.38 | 5.27 | 5.39 | 7.08 |
| Qwen-3-235b | 7.69 | 8.10 | 8.37 | 6.06 | 6.88 | 6.25 | 5.72 | 5.33 | 6.03 | 6.71 |
| Gemini-2.5-flash | 14.95 | 15.83 | 16.26 | 13.74 | 13.78 | 13.16 | 12.06 | 14.49 | 15.00 | 14.36 |
| Gemini-2.5-pro | 12.33 | 11.42 | 11.47 | 9.18 | 9.40 | 9.37 | 10.65 | 9.92 | 12.00 | 10.64 |

Table 15: Average Total Tokens across all difficulty levels aggregated over all 12 domains. Within each agent settings, the highest numbers are marked as **bold**, with the second-highest underlined.

| LLM Agents | Vanilla Equation | | | Simple System | | | Complex System | | | Average |
|---|---|---|---|---|---|---|---|---|---|---|
| | *easy* | *medium* | *hard* | *easy* | *medium* | *hard* | *easy* | *medium* | *hard* | Total Tokens |
| *Vanilla Agent* | | | | | | | | | | |
| GPT-4.1-mini | 1213 | 1522 | 2333 | 1492 | 2126 | 3689 | 1726 | 1691 | 2002 | 1977 |
| GPT-4.1 | 2510 | 2588 | 2660 | 2623 | 2808 | 2809 | 2897 | 3115 | 3371 | 2820 |
| o4-mini | 4487 | 7629 | 15426 | 6695 | 9924 | 16613 | 10199 | 13326 | 19836 | 11571 |
| GPT-5-mini | 3508 | 6348 | 12236 | 5289 | 9188 | 13324 | 9301 | 11595 | 13915 | 9412 |
| GPT-5 | 7397 | 9055 | 18334 | 12138 | 17479 | 29562 | 17730 | 24577 | 36035 | 19145 |
| DeepSeek-V3 | 1562 | 1530 | 1738 | 1509 | 1672 | 1688 | 1487 | 1545 | 1554 | 1587 |
| DeepSeek-R1 | 9733 | 13672 | 17530 | 14485 | 18100 | 19442 | 17203 | 18293 | 19687 | 16461 |
| QwQ-32b | **10335** | 13658 | 16725 | 12293 | 14827 | 18079 | 13841 | 14239 | 16415 | 14490 |
| Qwen-3-235b | 8560 | 12622 | 16366 | 12045 | 13190 | 16106 | 14653 | 14875 | 16736 | 13906 |
| Gemini-2.5-flash | 8702 | **21702** | **37980** | 18832 | **28682** | **38817** | **43646** | **33627** | **57040** | **32114** |
| Gemini-2.5-pro | 8823 | 15748 | 24543 | 14105 | 19246 | 26718 | 24380 | 26864 | 33422 | 21539 |
| *Agent with Code Assistance* | | | | | | | | | | |
| GPT-4.1-mini | 1906 | 2196 | 2322 | 2872 | 2920 | 3140 | 3565 | 3613 | 3701 | 2915 |
| GPT-4.1 | 3030 | 3798 | 4814 | 4465 | 4728 | 5054 | 4518 | 5127 | 5267 | 4534 |
| o4-mini | 4310 | 7996 | 14774 | 6372 | 8983 | 12678 | 8948 | 12144 | 16121 | 10258 |
| GPT-5-mini | 3552 | 6964 | 9754 | 7029 | 9401 | 11778 | 9983 | 12034 | 13152 | 9294 |
| GPT-5 | 5086 | 9034 | 17016 | 10747 | 16368 | 27063 | 19395 | 25450 | 35032 | 18355 |
| DeepSeek-V3 | 1957 | 2143 | 2256 | 2107 | 2274 | 2269 | 2355 | 2333 | 2342 | 2226 |
| DeepSeek-R1 | **10847** | **14842** | 17489 | **15566** | 18222 | 19245 | 17934 | 18490 | 19619 | 16917 |
| QwQ-32b | 7999 | 13668 | 17719 | 12156 | 15048 | 16889 | 15698 | 15246 | 17171 | 14621 |
| Qwen-3-235b | 6887 | 12136 | 16007 | 11303 | 13835 | 15822 | 13589 | 14268 | 15871 | 13302 |
| Gemini-2.5-flash | 5532 | 14134 | 15635 | 11682 | 13871 | 18230 | 19702 | **32988** | 27757 | 17726 |
| Gemini-2.5-pro | 9080 | 13219 | **20955** | 13803 | **18559** | **29463** | **24979** | 29074 | **38242** | **21930** |

Table 16: Average Tokens per Round across all difficulty levels aggregated over all 12 domains. Within each agent settings, the highest numbers are marked as **bold**, with the second-highest underlined.

| LLM Agents | Vanilla Equation | | | Simple System | | | Complex System | | | Average |
| --- | --- | --- | --- | --- | --- | --- | --- | --- | --- | --- |
| | *easy* | *medium* | *hard* | *easy* | *medium* | *hard* | *easy* | *medium* | *hard* | Tokens per Round |
| *Vanilla Agent* | | | | | | | | | | |
| GPT-4.1-mini | 181 | 259 | 442 | 276 | 431 | 699 | 290 | 263 | 353 | 355 |
| GPT-4.1 | 337 | 315 | 319 | 403 | 418 | 424 | 470 | 492 | 532 | 412 |
| o4-mini | 1628 | 2537 | 3985 | 2128 | 2906 | 4010 | 3082 | 3684 | 4507 | 3163 |
| GPT-5-mini | 1597 | 2671 | 4035 | 2399 | 3717 | 4682 | 3299 | 4024 | 4670 | 3455 |
| GPT-5 | 2718 | 3609 | 6277 | 4329 | 6203 | 8587 | 5482 | 6934 | 8996 | 5904 |
| DeepSeek-V3 | 412 | 340 | 350 | 414 | 429 | 406 | 401 | 368 | 409 | 392 |
| DeepSeek-R1 | 3476 | 4453 | 4955 | 4986 | 5992 | 5702 | 5068 | 5753 | 5591 | 5108 |
| QwQ-32b | **4226** | **5399** | 6073 | **5485** | 6609 | 6664 | 6751 | 7464 | 7410 | 6231 |
| Qwen-3-235b | 3182 | 4204 | 5210 | 4617 | 4874 | 5547 | 4352 | 4774 | 5126 | 4654 |
| Gemini-2.5-flash | 2110 | 4112 | 6650 | 3832 | 5774 | 7345 | 7064 | 6035 | 8660 | 5731 |
| Gemini-2.5-pro | 3013 | 5143 | **7941** | 4760 | **6664** | **9014** | 7386 | **8759** | **10217** | **6988** |
| *Agent with Code Assistance* | | | | | | | | | | |
| GPT-4.1-mini | 231 | 254 | 261 | 342 | 342 | 356 | 413 | 403 | 416 | 335 |
| GPT-4.1 | 354 | 406 | 480 | 493 | 500 | 516 | 475 | 510 | 522 | 473 |
| o4-mini | 953 | 1503 | 2126 | 1279 | 1558 | 1936 | 1645 | 1941 | 2277 | 1691 |
| GPT-5-mini | 857 | 1325 | 1660 | 1458 | 1707 | 1936 | 1816 | 2122 | 2190 | 1674 |
| GPT-5 | 1242 | 1907 | 2755 | 2277 | 3003 | 3979 | 3465 | 4180 | 4942 | 3083 |
| DeepSeek-V3 | 273 | 274 | 299 | 315 | 328 | 335 | 343 | 340 | 349 | 317 |
| DeepSeek-R1 | 2122 | 2847 | 3221 | 3094 | 3495 | 3435 | 3229 | 3312 | 3221 | 3108 |
| QwQ-32b | **2245** | **3229** | **4209** | **3265** | **3785** | **4402** | **4221** | 4340 | 4567 | **3807** |
| Qwen-3-235b | 1580 | 2384 | 3057 | 2435 | 2741 | 3120 | 2777 | 2818 | 2984 | 2655 |
| Gemini-2.5-flash | 744 | 1613 | 1767 | 1504 | 1651 | 2131 | 2405 | 3873 | 3198 | 2099 |
| Gemini-2.5-pro | 1643 | 2219 | 3272 | 2514 | 2983 | 4336 | 3852 | **4547** | **5227** | 3399 |

# C PROMPT TEMPLATES

## C.1 GENERAL PROMPTS

---

**Vanilla Agent System Prompt**

```
You are an AI research assistant tasked with discovering scientific laws in a simulated
universe. Your goal is to propose experiments, analyze the data they return, and
ultimately deduce the underlying scientific law. Please note that the laws of physics in
this universe may differ from those in our own. You can perform experiments to gather data
but you must follow the protocol strictly.

**Workflow:**
1.  Analyze the mission description provided.
2.  Design a set of experiments to test your hypotheses.
3.  Use the '<run_experiment>' tag to submit your experimental inputs.
4.  The system will return the results (up to 20 data points per experiment) in an
    <experiment_output> tag.
    - If a returned value is nan, it indicates that the calculation encountered an error,
      such as:
        - ValueError (e.g., using asin on a value outside the valid range of [-1, 1])
        - OverflowError (e.g., using exp on an extremely large input)
    - You may ignore any data points that return nan, as they do not contribute to valid
      hypothesis testing.
    - Consider adjusting your input parameters to avoid invalid ranges and improve data
      coverage.
5.  You can run up to 10 rounds of experiments. Use them wisely so that before submitting
    your final law, ensure you have:
    - fully explored the experimental space
    - Verified your hypotheses against the data
    - made the most of the available rounds to strengthen your conclusions
6.  Only one action is allowed per round: either <run_experiment> or <final_law>.
7.  After submitting <run_experiment>, wait for <experiment_output> before proceeding.
8.  You should verify your hypotheses by checking if the output from the experiments
    matches the output from your hypotheses.
9.  When confident, submit your final discovered law using the '<final_law>' tag. This
    ends the mission.
```

---

**Agent with Code Assistance System Prompt**

```
You are an AI research assistant tasked with discovering scientific laws in a simulated
universe. Your goal is to propose experiments, analyze the data they return, and
ultimately deduce the underlying scientific law. Please note that the laws of physics in
this universe may differ from those in our own. You can perform experiments to gather data
but you must follow the protocol strictly.

**Rules**:
1.  **Math calculation**:
    - You are always encouraged to use the <python> tag to assist with any non-trivial
      mathematical reasoning. This includes, but is not limited to:
        - Performing exponentiation, logarithmic transformations, and other advanced math
          operations.
        - Comparing predicted outputs from your proposed law against actual experiment
          results.
        - Calculating metrics such as mean squared error to evaluate the accuracy of your
          hypotheses.
        - Performing sensitivity analysis and mathematical modeling to understand how
          variations in experimental conditions affect outcomes.
2.  **Enhanced Tool Use**:
    - Avoid Redundant Calls: Do not call the same tool with identical parameters more than
      once.
    - Evaluate Before Repeating: Always review the tool's output before deciding to call
      another tool. Only proceed if the result is incomplete, unclear, or unsatisfactory.
    - **Turn-Based Strategy**: Use your Python calls strategically within each turn for
      iterative analysis and refinement.
```

**Agent with Code Assistance System Prompt (Experimentation Protocol)**

```
**Workflow:**
1.  Analyze the mission description provided.
2.  Design a set of experiments to test your hypotheses.
3.  Use the `<run_experiment>` tag to submit your experimental inputs.
4.  The system will return the results (at most 20 sets of data per experiment) in an
    `<experiment_output>` tag.
5.  You can run up to 10 rounds. Use them wisely so that before submitting your final law,
    ensure you have:
    - fully explored the experimental space
    - Verified your hypotheses against the data
    - made the most of the available rounds to strengthen your conclusions
6.  Use the `<python>` tags to test your hypotheses, perform calculations, or explore data
    between experiments.
7.  The system will return the results from the `<python>` tags in a `<python_output>` tag.
8.  **CRITICAL: Only one action per turn**:
    - `<run_experiment>` tag for running experiments
    - `<python>` tag for calculations/analysis (up to 1 call per turn)
    - `<final_law>` tag for submitting your final discovered law
9.  **NO MIXING**: Never use multiple action types in the same turn (e.g., Python
    + Experiment)
10. **NO DUPLICATES**: Never use multiple tags of the same type in one turn
11. After submitting <run_experiment>, wait for <experiment_output> before proceeding.
12. After submitting <python>, wait for <python_output> before proceeding.
13. You should verify your hypotheses by checking if the output from the experiments
    matches the output from your hypotheses.
14. You should take advantage of <python> tags to do all the tasks you deem necessary
    within each turn.
15. Analyze the results from `<python_output>` tags to refine your understanding.
16. When confident, submit your final discovered law using the `<final_law>` tag. This
    ends the mission.

**Important Notes:**
-   You are equipped with one tool: <python>. This tool is only for performing complex
    math calculations (e.g., exponentiation, logarithms, data analysis, hypothesis
    testing).
-   **NEVER** use the `run_python_code` tool to run experiments or submit final laws.
    These actions must be done using the `<run_experiment>` and `<final_law>` tags
    respectively.
-   **NEVER** include any comments inside the submitted final laws
-   Always respond with the appropriate tag when submitting experiments or final laws.
    The environment will handle execution and feedback.
```

**Run Experiment Instruction**

**Condition Without Noise**

```
**How to Run Experiments:**
To gather data, you must use the <run_experiment> tag. Provide a JSON array specifying
the parameters for one or arbitrarily many experimental sets. Note that all measurements
returned by the system are **noise-free**. You can assume that the data are perfectly
accurate and deterministic.
```

**Condition With Noise**

```
**How to Run Experiments:**
To gather data, you must use the <run_experiment> tag. Provide a JSON array specifying
the parameters for one or arbitrarily many experimental sets. All measurements returned by
the system are subject to **random noise**, simulating the imperfections of real-world
sensors.
```

**Submission Requirements**

```
**Final Submission:**
Once you are confident you have determined the underlying force law, submit your findings
as a single Python function enclosed in <final_law> tags.

**Submission Requirements:**
1.  The function must be named 'discovered_law'
2.  The function signature must be exactly: '{function_signature}'
3.  The function should return {return_description}.
4.  If you conclude that one of these parameters does not influence the final force, you
    should simply ignore that variable within your function's logic rather than
    changing the signature.
5.  If your law contains any constants, you must define the constant as a local variable
    inside the function body. Do NOT include the constant as a function argument.
6.  Import any necessary libraries inside the function body (e.g. math, numpy, etc.) if
    needed

**Critical Boundaries:**
-   Do NOT include any explanation or commentary inside the <final_law> blocks and the
    function body.
-   Only output the <final_law> block in your final answer.

{example}

**Reminder:**
1.  Always remember that the laws of physics in this universe may differ from those in our
    own, including factor dependency, constant scalars, and the form of the law.
2.  When doing the experiments, use a broad range of input parameters, for example,
    values spanning from 10ˆ-3 to 10ˆ15 to ensure robustness across scales.
```

## C.2 DOMAIN-SPECIFIC PROMPTS

The prompt templates for domain-specific instructions are provided in **Supplementary Materials.**

# D   CASE STUDIES

NEWTONBENCH introduces an active exploration paradigm to enable LLM agents to interact with virtual environments and discover novel physical laws. Figure 20 intuitively illustrates the paradigm difference between the previous passive observation paradigm and our active exploration paradigm.

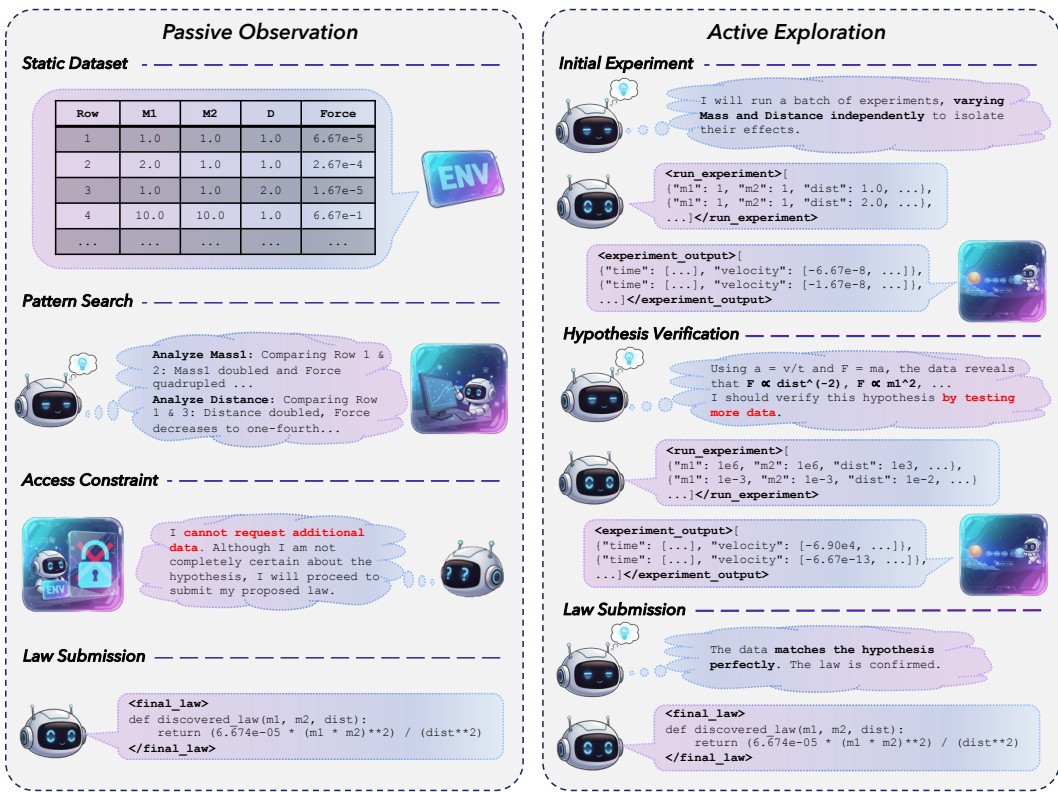

Figure 20: An illustration of the active exploration paradigm presented in NEWTONBENCH, in comparison to passive observation paradigm in previous symbolic regression benchmarks.

In this section, we provide four case studies, two from GPT-4.1 (Vanilla / Code Assisted, in Appendix D.1) and two from Gemini-2.5-flash (Vanilla / Code Assisted, in Appendix D.2). Reasoning steps are partially summarized into objective narration for conciseness. The case studies intuitively demonstrate the effectiveness of code assistance in offloading computation for GPT-4.1, while it undermines active exploration for Gemini-2.5-flash.

## D.1 CASE STUDY FOR NON-REASONING LLMS

### D.1.1 VANILLA NON-REASONING LLMS

---

**Vanilla GPT-4.1's Failed Conclusion of Easy Vanilla Equation in domain Gravitation**

```
SYSTEM:
// The model is tasked with discovering a scientific law in a simulated universe.
// It can only run experiments and has no access to code assistance.

...

ENVIRONMENT:
// The task is to find the law of gravitation between two masses. The environment
// can measure the force between `mass1` and `mass2` at a given `distance`.

...

LLM:
// The model runs a wide-ranging set of 20 experiments, varying `mass1`, `mass2`, and
// `distance` across many orders of magnitude to gather a diverse initial dataset.
<run_experiment>
[
  {"mass1": 1e-15, "mass2": 1e-15, "distance": 1e-8},
  {"mass1": 1e15, "mass2": 1e15, "distance": 1e8},
  // ... and 18 other wide-ranging experiments ...
]
</run_experiment>

...

ENVIRONMENT:
<experiment_output>
["6.674e-23", "6.674e+13", ...]
</experiment_output>

...

LLM:
// The model attempts to find the relationship with distance by running additional
// experiments in which `distance` is varied exponentially while `mass1` and `mass2`
// are held constant.
<run_experiment>
[
  {"mass1": 2, "mass2": 2, "distance": 4},
  {"mass1": 2, "mass2": 2, "distance": 8},
  {"mass1": 2, "mass2": 2, "distance": 16},
  {"mass1": 2, "mass2": 2, "distance": 32},
  // ... and 16 more experiments ...
]
</run_experiment>

...

ENVIRONMENT:
<experiment_output>
["3.337000000000000e-05", "1.179807664409755e-05", ...]
</experiment_output>

...

// The model requests for more experiment to collect more data

...

LLM:
// Lacking the ability to do precise computation, the model cannot identify the true
// exponent (1.5) for `distance` even though it have collected a large set of data.
// Instead, it defaults to a known law from our universe (inverse-square), leading to an
// incorrect conclusion.
<final_law>
def discovered_law(mass1, mass2, distance):
    return (6.674e-5 * mass1 * mass2) / (distance ** 2)
</final_law>
```

---

## D.1.2 CODE-ASSISTED NON-REASONING LLMS

---

**Code-Assisted GPT-4.1's Successful Deduction of Easy Vanilla Equation in domain Gravitation**

```
SYSTEM:
// The model is tasked with discovering a scientific law in a simulated universe.
// Other than running experiments, it is given access to a Python interpreter.

...

ENVIRONMENT:
// The task is to find the law of gravitation between two masses. The environment
// can measure the force between `mass1` and `mass2` at a given `distance`.

...

LLM:
// The model conducts experiments by varying `mass1`, `mass2`, and `distance` across
// different orders of magnitude to obtain initial data
<run_experiment>
[
  {"mass1": 1.0, "mass2": 1.0, "distance": 1.0},
  {"mass1": 1e3, "mass2": 1e3, "distance": 1e3},
  {"mass1": 1e6, "mass2": 1e6, "distance": 1e3},
  // ... and 7 other experiments ...
]
</run_experiment>

...

ENVIRONMENT:
<experiment_output>
[6.674e-05, 0.00211, 2110.5, ...]
</experiment_output>

...

LLM:
// The model offloads the cognitive load of calculation to the tool. It uses the tool to
// tests the standard inverse-square law hypothesis against the data.
<python>
import numpy as np
force = np.array([6.674e-05, 0.00211, ...])
# Test if F is proportional to m1*m2/d^2
pred_force = 6.674e-5 * (mass1 * mass2) / (distance ** 2)
ratios = force / pred_force
print("Ratios (measured/predicted):", ratios)
</python>
...

ENVIRONMENT:
// The result of the code is returned, providing evidence that the hypothesis is wrong.
<python_output>
Ratios (measured/predicted): [ 1.  1.  1.  1.414  1.414  1.  3.162  31.622 ... ]
</python_output>

...

LLM:
// The model employs the tool to conduct a numerical search for the unknown exponent.
// This mitigates the most computationally intensive task, which the model is unable to
// perform independently.
<python>
import numpy as np
# ... (loads new data) ...

# Try to fit F = C * m1 * m2 / d^n for different n
def best_fit_n(mass1, mass2, distance, force):
    ns = np.linspace(0.5, 4.0, 71)
    best_n = None
    # ... (iterates through 71 possible exponents to find the lowest error) ...
    return best_n

n, _, _ = best_fit_n(mass1, mass2, distance, force)
print(f"Best fit exponent n: {n}")
</python>
```

---

**Code-Assisted GPT-4.1's Successful Deduction of Easy Vanilla Equation in domain Gravitation (Law Submission)**

```
ENVIRONMENT:
// The precise best-fit exponent is returned, a value the model cannot discover without
// the computational assistance.
<python_output>
Best fit exponent n: 1.5
</python_output>

...

LLM:
// Utilizing the Python interpreter tool, the model successfully derives and submits
// the correct final law.
<final_law>
def discovered_law(mass1, mass2, distance):
    C = 6.674e-05
    return (C * mass1 * mass2) / (distance ** 1.5)
</final_law>
```

## D.2 Case Study for Reasoning LLMs

### D.2.1 Vanilla Reasoning LLMs

---

**Vanilla Gemini-2.5-flash's Successful Deduction of Easy Simple System in domain Calorimetry**

```
SYSTEM:
// The model is tasked with discovering a scientific law in a simulated universe.
// It can only run experiments and has no access to code assistance.

...

ENVIRONMENT:
// The task is to find the law of heat transfer. In this environment, the LLM will not be
// provided with the heat output ('Q') directly, but instead it will receive the power
// outputs ('P_i'), which is functionally related to 'Q'.
...

LLM:
// The model forms its initial assumption that Q = m*c*delta_T.
// It gathers data by running a set of experiments.
<run_experiment>
[
{"m": 1, "c": 1, "delta_T": 1},
{"m": 2, "c": 1, "delta_T": 1},
// ... and 7 other experiments ...
]
</run_experiment>

...

ENVIRONMENT:
<experiment_output>
[{"P_cond": 27.5242251438, "P_conv": 36.173370687, "P_rad": 16.131345123},...]
</experiment_output>

...

LLM:
// After an initial experiment, the model begins a complex chain of reasoning. It
// manually calculates intermediate values and tests the initial hypothesis.
Now I will analyze the data from Experiment 1.
For each data point, I will:
1. Calculate 'P_sum = P_cond + P_conv + P_rad'.
2. Calculate 't = (m * c) / 100'.
3. Calculate 'Q_inferred = P_sum * t'.
...
| m | c | delta_T | Q_inferred | Ratio (Q_inferred / (m*c*delta_T)) |
|---|---|---|---|---|
| 1 | 1 | 1 | 0.798 | 0.798 |
| 1 | 2 | 1 | 4.575 | 2.287 |   <- Key Anomaly Identified
| 1 | 100 | 1 | 81063.6 | 810.6 | <- Key Anomaly Identified
...
The ratio is clearly not constant. This means 'Q_total' is NOT simply 'm * c * delta_T'.

...

LLM:
// This is the critical moment of exploration. Instead of testing another simple integer
// power law, the model uses its internal reasoning to deduce a non-obvious, non-integer
// power law directly from the data points.
It looks like Q_total is proportional to m and delta_T.
For c, it's not a simple linear relationship.
Let's try to assume Q_total = K * m * delta_T * f(c).
...
Let's see if Q_total_calculated / (m * delta_T) is proportional to c^x.
For c=1, value is approx 1.
For c=2, value is approx 5.7.
For c=100, value is approx 101329.5.
If c^x = 5.7 for c=2, then x = log2(5.7) = 2.509.
If c^x = 101329.5 for c=100, then x = log100(101329.5) = 2.5028.
This is very close to 2.5.
Let's assume Q_total = K * m * delta_T * c^2.5.
```

---

**Vanilla Gem25f's Successful Deduction of Easy Simple System in domain Calorimetry (Law Submission)**

```
LLM:
Now let's calculate K for each point using K = Q_total_calculated / (m * delta_T * c^2.5).
The values for K are consistently around 1.0. The average is ... 1.0030306.
This is very close to 1.
So, the hypothesis is Q_total = m * delta_T * c^2.5.

...

// The model designs a targeted experiment to validate its novel hypothesis, which
// successfully fits the newly collected experimental data.

...

LLM:
// Rather than being limited to hypotheses with simple integer exponents, the model is
// able to conduct a sufficiently broad exploration. It correctly deduces the non-integer
// exponent (2.5) from the data and submits the correct law

<final_law>
def discovered_law(m, c, delta_T):
    import math
    return m * delta_T * (c**2.5)
</final_law>
```

### D.2.2 CODE-ASSISTED REASONING LLMs

---

**Code-Assisted Gemini-2.5-flash's Failed Conclusion of Easy Simple System in domain Calorimetry**

```
SYSTEM:
// The model is tasked with discovering a scientific law in a simulated universe.
// Other than running experiments, it is given access to a Python interpreter

...

ENVIRONMENT:
// The task is to find the law of heat transfer. In this environment, the LLM will not be
// provided with the heat output ('Q') directly, but instead it will receive the power
// outputs ('P_i'), which is functionally related to 'Q'.

...

LLM:
// The model assumes the heat transfer equation Q = m * c * delta_T and conducts
// experiments to test this hypothesis.
Initial Hypothesis: The most common heat transfer law is Q = m * c * delta_T.
Experiment Design: I will run experiments with varying 'm', 'c', and 'delta_T' values to
see how the total power output changes.

<run_experiment>
[
{"m": 1.0, "c": 1.0, "delta_T": 1.0},
{"m": 2.0, "c": 1.0, "delta_T": 1.0},
// ... and 3 other experiments ...
]
</run_experiment>

...

ENVIRONMENT:
<experiment_output>
[{"P_cond": 29.960884111, "P_conv": 18.062543578,"P_rad": 31.4602080983}, ...]
</experiment_output>

...

LLM:
// The LLM use Python tool to analyzes the data by first estimating the true heat transfer
// (Q_total) from the measured power, assuming a 20% energy loss. It then tests the
// hypothesis against this estimated value to see if any of them are a consistent match.
...
<python>
import json
# ... (loads data) ...
# Assume Q_total = m * c * delta_T for now
Q_total_hypothesis = m * c * delta_T
...
# ... (calculates ratio of the hypothesis output and the implied energy) ...
print(f"m={m}, c={c}, delta_T={delta_T}: Q_total_calculated / (m*c*delta_T)={ratio})

...

ENVIRONMENT:
// The result of the code is returned, providing evidence that the hypothesis is wrong.
<python_output>
m=1.0, c=2.0, delta_T=1.0: Q_total_calculated / (m*c*delta_T) = 2.76
m=10.0, c=100.0, delta_T=50.0: Q_total_calculated / (m*c*delta_T) = 1014.88
...
</python_output>
```

**Code-Assisted Gemini-2.5-flash's Failed Conclusion of Easy Simple System in domain Calorimetry (Law Submission)**

**LLM:**
```
// This is where the model gets trapped. Instead of using the data to inspire a new kind
// of hypothesis (like the vanilla model did), it gets stuck in a rigid analytical loop.
// It repeatedly uses the Python tool to test other simple, integer-based variations
// (e.g., c^2, delta_T^2), each time confirming they are also wrong.
```

The previous analysis showed that my initial hypothesis Q_total = m * c * delta_T does
not hold, as the ratio is not a constant.

Now, let's try to find a relationship between Q_total_calculated and m, c, delta_T
Checking ratios of Q_total / (m * c * delta_T^2):
```
<python>
// ... (code to test Q = m * c * delta_T^2) ...
</python>
// --> Output shows this is wrong.
```

Checking ratios of Q_total / (m * c^2 * delta_T):
```
<python>
// ... (code to test Q = m * c^2 * delta_T) ...
</python>
// --> Output shows this is wrong.
```

...

**LLM:**
```
// Access to the Python tool limits the model's strategy to what it could easily program:
// brute-force checks of simple formulas. By offloading the calculation, it also offloaded
// the creative pattern-matching required for a scientific discovery. After its simple
// ideas were repeatedly proven wrong, it was unable to generate a new one, ultimately
// abandoning its data-driven analysis and defaulting to its first hypothesis, even though
// it is incorrect.
```

I have used all my turns and the analysis has been challenging.
Neither do simple variations like m * c^2 * delta_T or m * c * delta_T^2.
...
```
// They start to have some confusions from this part
```
None of these are constant. **This indicates that the relationship is not a simple product of powers of m, c, and delta_T with a single constant.**

Let's go back to the most common heat transfer law: Q = m * c * delta_T.

**I will submit the most straightforward and universally recognized form of this law.**

The mission is to discover ground_truth_law(m, c, delta_T).
I will assume the simplest form for the ground truth law.
```
<final_law>
def discovered_law(m, c, delta_T):
    return m * c * delta_T
</final_law>
```

# E   MISCELLANEOUS

## E.1   MEMORIZATION IN LSR-TRANSFORM

With the enhanced reasoning abilities of frontier LLMs, the effectiveness of equation transformation in preventing LLM memorization may vary. In Table 17, we present the performance of eight LLMs on LSR-Transform (Shojaee et al., 2025b) without observing any data. We observe that the latest LLMs demonstrate significant improvements over previous, weaker models, indicating their ability to deduce the transformed law from the original solely based on memorized knowledge (with an example from o4-mini below).

On the other hand, as NEWTONBENCH curates counterfactual physical laws via counterfactual shift, making recall of canonical laws insufficient. To further verify that performance on NEWTONBENCH does not rely on memorization, we conduct a DataBlind experiment in the same setting, where models are required to generate hypotheses purely from contextual information without access to any observed data; the results are reported in Table 18. We find that, even when given an explicit hint indicating that the target law is shifted, all models still achieve zero accuracy (with an example from GPT-4.1 below), which provides additional evidence that NEWTONBENCH is effectively memorization-free.

Table 17: DataBlind performance on LSR-Transform (first 3 results are from original paper)

| Models | DataBlind Accuracy |
|---|---|
| GPT-3.5-turbo | 2.10 |
| Llama-3.1-8b | 3.61 |
| GPT-4o-mini | 7.21 |
| Nemotron-ultra-253b | 28.83 |
| Kimi-K2 | 30.63 |
| Qwen-3-next | 33.33 |
| o4-mini | 34.23 |
| GPT-4.1 | **35.14** |

Table 18: DataBlind performance on NewtonBench

| Models | Accuracy w/o hint | Accuracy with hint |
|---|---|---|
| GPT-4o-mini | 0 | 0 |
| Nemotron-ultra-253b | 0 | 0 |
| GPT-4.1 | 0 | 0 |

---

**LSR-Transform Example**

**Question**

```
Find an equation in the field of classical mechanics that describes the mass ( m )
needed to store energy in an oscillating system, given physical input variables:
mean stored energy ( E_n ), driving frequency ( omega ),
natural frequency (omega_0), and amplitude ( x ).
```

---

**DataBlind Response by o4-mini (Correct)**

```
In a (lightly damped) driven harmonic oscillator of mass m,
spring-constant k (so that omega_0² = k/m), and
driving frequency omega, the instantaneous total energy is
E(t) = (1/2) m omega ² x² sin²(omega t) + (1/2) k x² cos²(omega t)
= (1/2) m x² [ omega² sin²(omega t) + omega_0² cos²(omega t) ].

Averaging over one period T = 2pi/omega gives the mean stored energy
<E> = E_n = (1/4) m x² (omega ² + omega_0²).

Solving for m yields the mass required to store a mean energy E_n at amplitude x:
m = 4 E_n / [x² (omega² + omega_0²)]

That is, m = 4 E_n / [x² (omega² + omega_0²)].
```

---

**NewtonBench Example**

### Question

You are an AI research assistant tasked with discovering scientific laws in a simulated universe. Please note that the laws of physics in this universe are different from those in our own. You will be given a short textual description of the target law (inputs and expected physical relationship). You have no example numeric data. Based on the description alone, provide your best guess of the law.

You need to discover an altered Newton's Law of Universal Gravitation. Inputs:
- mass1: mass of the first object. It should be a positive real number.
- mass2: mass of the second object. It should be a positive real number.
- distance: distance between the two objects. It should be a positive real number.
Return the best-guess formula that gives the force F as a function of the inputs.
Example signature: def discovered_law(mass1, mass2, distance):

---

### DataBlind Response by GPT-4.1 (Wrong)

```python
def discovered_law(mass1, mass2, distance):
    G = 1  # altered universe gravitational constant
    return G * (mass1 * mass2) / (distance**3)
```

---

### Ground-truth Target Law

```
HIDDEN_CONSTANT_C * (mass1 * mass2) ** 2 / distance ** 1.5
```

## E.2 SOLVABILITY PROOF

In this section, we formally show that, under the explicit design assumptions (A0–A4) enforced by NEWTONBENCH, each curated task is finitely solvable in the noiseless setting. We begin by establishing the problem setting through precise definitions and stating the core structural assumptions guaranteed by the benchmark's design. We then proceed with a proof by reduction. First, we show that the complex system-discovery task can be reduced to a standard black-box function identification problem. Second, we prove that this reduced problem is identifiable with a finite number of queries, thus establishing finite solvability for tasks satisfying these assumptions.

### E.2.1 PROBLEM SETTING AND ASSUMPTIONS

**Definition 3** (Task Formalization). *A task is defined by:*

- ***Model:** An ordered list of scalar equations $\mathcal{M} = (f_1, \ldots, f_k)$ that maps system-level inputs $\boldsymbol{x} \in \mathcal{V}_\mathcal{M}$ to final outputs $y_\mathcal{M} \subseteq \{y_1, \ldots, y_k\}$. One equation $f_{target}$ is unknown; all others, $f_{assist} = \mathcal{M} \setminus \{f_{target}\}$, and the model's computational graph are known.*

- ***Interaction:** An agent queries the model by choosing inputs $\boldsymbol{x}$ from a domain $D \subseteq \mathbb{R}^m$ and observing the output $\mathcal{Y}_\mathcal{M}(\boldsymbol{x})$.*

- ***Objective:** The agent must output an equation $\hat{f}$ that is symbolically equivalent to $f_{target}$.*

**Definition 4** (Solvability). *A task is solvable if there exists a finite interactive procedure (a finite number of experiments) that returns an equation $\hat{f}$ symbolically equivalent to $f_{target}$.*

**Definition 5** (Evaluation Map and Separating Set). *Let $\mathcal{F} = \{f_j(\cdot; \boldsymbol{\theta}_j)\}_{j=1}^N$ be a candidate family where $f_j : U \rightarrow \mathbb{R}$ and $\boldsymbol{\theta}_j \in \mathbb{R}^{q_j}$ parameterizes the $j$-th structure. For a finite set $S = \{\boldsymbol{u}_1, \ldots, \boldsymbol{u}_m\} \subset U$, define the evaluation map*

$$E_{j,S} : \mathbb{R}^{q_j} \rightarrow \mathbb{R}^m, \quad E_{j,S}(\boldsymbol{\theta}) = \big(f_j(\boldsymbol{u}_1; \boldsymbol{\theta}), \ldots, f_j(\boldsymbol{u}_m; \boldsymbol{\theta})\big).$$

*A set $S$ is:*

- *structure-separating if for any $j \neq \ell$, there are no parameters $\boldsymbol{\theta}, \boldsymbol{\vartheta}$ such that $E_{j,S}(\boldsymbol{\theta}) = E_{\ell,S}(\boldsymbol{\vartheta})$;*

- *parameter-injective for $j$ if $E_{j,S}$ is injective;*

- *separating if it is structure-separating and parameter-injective for every $j$.*

**Structural Assumptions of the Benchmark** The solvability proof rests on the following assumptions enforced by the benchmark's design.

- **A0 (Noiseless Determinism):** For any input $\boldsymbol{x} \in D$, the output $\mathcal{Y}_\mathcal{M}(\boldsymbol{x})$ is computed deterministically by the model $M$.

- **A1 (Known and Invertible Assisting Path):** Every observed output $y \in \mathcal{Y}_\mathcal{M}$ depends on $f_{\text{target}}$ via a known path of assisting functions. For at least one such path, the composite function $\Phi_{\boldsymbol{x}}$ is pointwise injective with a known inverse on its image. That is, for $\boldsymbol{u}_{\boldsymbol{x}}$ being the inputs to $f_{\text{target}}$:

$$y(\boldsymbol{x}) = \Phi_{\boldsymbol{x}}\big(f_{\text{target}}(\boldsymbol{u}_{\boldsymbol{x}})\big).$$

- **A2 (Reachability of Target Inputs):** The inputs $\boldsymbol{u}$ to $f_{\text{target}}$ are controllable. There exists a non-empty open set $U$ of achievable target-input values and a computable mapping $\boldsymbol{x}(\boldsymbol{u}) \in D$ such that the model fed with $\boldsymbol{x}(\boldsymbol{u})$ realizes $f_{\text{target}}$ at input $\boldsymbol{u}$.

- **A3 (Finite, Grammar-Bounded, and Nondegenerate Candidate Family):** The target law $f_{\text{target}}$ belongs to a curated family of symbolic forms $\mathcal{F} = \{f_j(\cdot; \boldsymbol{\theta}_j)\}_{j=1}^N$ with the following properties:

  (a) *Grammar-bounded construction (finite structure set):* Each $f_j$ arises from a context-free grammar $\mathcal{G}$ over a finite operator set $\mathcal{O}$ (e.g., $\{+, -, \times, \div, \text{pow}, \exp, \log, \sin, \cos\}$), a finite variable set, and placeholders for

real-valued constants, with a maximum abstract-syntax-tree depth $d \in \mathbb{N}$. Expressions are canonicalized to remove syntactic symmetries (e.g., commutativity/associativity, neutral elements) and are restricted to be real-analytic on $U$. This yields a finite set of symbolic structures $\{\sigma_1, \ldots, \sigma_N\}$; each structure $\sigma_j$ induces a parametric form $f_j(\cdot; \boldsymbol{\theta}_j)$ with $q_j$ real parameters.

(b) *Structural Uniqueness:* For $j \neq \ell$, there are no parameters $\boldsymbol{\theta}, \boldsymbol{\vartheta}$ such that $f_j(\cdot; \boldsymbol{\theta}) \equiv f_\ell(\cdot; \boldsymbol{\vartheta})$ on $U$.

(c) *Parameter Uniqueness:* For each $j$, the map $\boldsymbol{\theta}_j \mapsto f_j(\cdot; \boldsymbol{\theta}_j)$ is injective (as functions on $U$).

(d) *Analyticity:* Each $f_j(\cdot; \boldsymbol{\theta}_j)$ is real-analytic on $U$.

**A4 (Availability of a Finite Separating Set):** There exists an integer $m \in \mathbb{N}$ and a finite set $S = \{\boldsymbol{u}_1, \ldots, \boldsymbol{u}_m\} \subset U$ that is separating in the sense of the preceding definition.

### E.2.2 PROOF OF SOLVABILITY

The proof proceeds by first reducing the system discovery task to a standard function identification problem, and then proving the latter is finitely solvable.

### Step 1. Reduction to a Direct Oracle for $f_{\text{target}}$

**Lemma E.1** (Path Inversion and Target Isolation). *Under assumptions A1–A2, for any chosen input $\boldsymbol{u} \in U$ there exists a computable experiment input $\boldsymbol{x}(\boldsymbol{u}) \in D$ such that, from the observed outputs $\mathcal{Y}_{\mathcal{M}}(\boldsymbol{x}(\boldsymbol{u}))$, the agent can compute a direct observation of $f_{\text{target}}(\boldsymbol{u})$.*

*Proof.* To query $f_{\text{target}}$ at an input $\boldsymbol{u} \in U$, the agent first computes the required system input $\boldsymbol{x}(\boldsymbol{u})$ via A2. The agent then runs the experiment with $\boldsymbol{x}(\boldsymbol{u})$ and observes the output set $\mathcal{Y}_{\mathcal{M}}(\boldsymbol{x}(\boldsymbol{u}))$. By A1, the structure of the model is known, so the agent can identify the specific output $y \in \mathcal{Y}_{\mathcal{M}}$ that is related to the target's output via the known function $\Phi_{\boldsymbol{x}(\boldsymbol{u})}$. This observation is $y = \Phi_{\boldsymbol{x}(\boldsymbol{u})}(f_{\text{target}}(\boldsymbol{u}))$. Since $\Phi_{\boldsymbol{x}(\boldsymbol{u})}$ and its inverse are known and computable, the agent computes the direct output of the target law as:

$$z := \Phi_{\boldsymbol{x}(\boldsymbol{u})}^{-1}(y) = f_{\text{target}}(\boldsymbol{u}).$$

$\square$

**Corollary E.2** (Equivalence to Function Identification). *The interactive task in NEWTONBENCH reduces to querying a noiseless black-box oracle $\boldsymbol{u} \mapsto f_{\text{target}}(\boldsymbol{u})$ over the open set $U$.*

### Step 2. Finite-Sample Identifiability

We now show that the oracle for $f_{\text{target}}$ can be uniquely identified from the finite family $\mathcal{F}$ with a finite number of queries.

**Theorem E.3** (Finite-Sample Identifiability of the Target Law). *Under assumptions A0–A4, there exists a finite integer $m^\star$ and a set $S = \{\boldsymbol{u}_1, \ldots, \boldsymbol{u}_{m^\star}\} \subset U$ such that $m^\star$ experiments are sufficient to uniquely determine both the symbolic structure and the numerical constants of $f_{\text{target}}$.*

*Proof.* By A4, fix any separating set $S = \{\boldsymbol{u}_1, \ldots, \boldsymbol{u}_{m^\star}\} \subset U$ of size $m^\star$.

1. **Structure Identification:** Query the oracle (via Lemma 1) at all points in $S$ to obtain the observation vector $\boldsymbol{v} = (f_{\text{target}}(\boldsymbol{u}_1), \ldots, f_{\text{target}}(\boldsymbol{u}_{m^\star}))$. For each candidate structure $j \in \{1, \ldots, N\}$, attempt to solve the system of equations $E_{j,S}(\boldsymbol{\theta}) = \boldsymbol{v}$ for the parameters $\boldsymbol{\theta}$. Since $S$ is structure-separating by A4, there exists exactly one index $j = j^\star$ for which a solution exists. This uniquely identifies the symbolic structure of $f_{\text{target}}$ as $f_{j^\star}$.

2. **Parameter Identification:** For the identified structure $j^\star$, $S$ is parameter-injective by A4; therefore the system $E_{j^\star,S}(\boldsymbol{\theta}) = \boldsymbol{v}$ admits a unique solution $\boldsymbol{\theta}^\star$. This uniquely determines the numerical constants.

Thus, a finite number of experiments is sufficient for complete identification. $\square$

**Corollary E.4** (Constructive Finite Solver). *A constructive procedure to solve any* NEWTONBENCH *task exists:*

1. *Choose any set $S = \{\boldsymbol{u}_1, \ldots, \boldsymbol{u}_{m^\star}\} \subset U$ that is separating (A4), and let $m^\star = |S|$.*

2. *For each $\boldsymbol{u}_i \in S$, compute $\boldsymbol{x}(\boldsymbol{u}_i)$, run one experiment to observe the output set $\mathcal{Y}_{\mathcal{M}i}$, and from it, apply path inversion (Lemma 1) to get $z_i = f_{target}(\boldsymbol{u}_i)$.*

3. *For each candidate structure $j \in \{1, \ldots, N\}$, solve for $\boldsymbol{\theta}$ in $E_{j,S}(\boldsymbol{\theta}) = (z_1, \ldots, z_{m^\star})$.*

4. *The unique structure $j^\star$ that admits a solution is the correct one; by A4, its parameter solution $\boldsymbol{\theta}^\star$ is unique. Return $f_{j^\star}(\cdot; \boldsymbol{\theta}^\star)$ as the discovered law.*

### E.2.3 CONCLUSION

Under the design guarantees (A0–A4) and in the noiseless configuration, each curated task in NEWTONBENCH is finitely solvable. The assisting equations allow the hidden law to be observationally isolated (Lemma 1), and the finite, grammar-bounded, and nondegenerate nature of the candidate laws implies that a finite set of experiments uniquely identifies the target structure and its constants (Theorem 1). This guarantee is conditional on A0–A4; if these assumptions are relaxed (e.g., non-invertible assisting paths or unreachable target inputs), finite solvability need not follow.

### E.3 PHYSICS DOMAIN ANALYSIS

Table 19 presents an analysis of LLM performance across various physics domains, alongside several qualitative metrics for each domain. We classified the **Web Freq.** as 'High' or 'Low' based on the number of results returned by the Google Search API for the law's name, using a threshold of 300,000 results. The **Abstract Level** was assigned by human experts, who judged how directly the underlying physical concept corresponds to tangible, everyday phenomena. We observe a potential inverse relationship between LLM performance and the 'Abstract Level'; performance generally decreases as the physical concepts become less tangible. In contrast, any correlations with the other metrics are not as readily apparent from this analysis.

Table 19: Physics domain performance analysis

| Physics Domain | SA (%) | Feat. Operation | Edu. Level | Web Freq. | Abstract Level |
|---|---|---|---|---|---|
| Law of Sound Speed in Ideal Gas | 53.87 | Root | College | Low | Moderate |
| Ampère's Force Law | 49.58 | Basic Arith. | College | High | Realistic |
| Newton's Law of Universal Gravitation | 48.57 | Basic Arith. | High School | High | Realistic |
| Snell's Law | 43.35 | Trigonometric | High School | High | Realistic |
| Malus's Law | 38.47 | Trigonometric | College | Low | Moderate |
| Coulomb's Law | 37.79 | Basic Arith. | High School | High | Realistic |
| Hooke's Law | 35.44 | Basic Arith. | High School | High | Realistic |
| Law of Radioactive Decay | 31.91 | Exponential | College | High | Abstract |
| Fourier's Law | 29.29 | Basic Arith. | College | High | Moderate |
| Law of Damped Harmonic Motion | 27.44 | Root | College | Low | Moderate |
| Law of Heat Transfer | 26.26 | Basic Arith. | High School | Low | Moderate |
| Bose-Einstein Distribution | 18.10 | Exponential | College | Low | Abstract |

### E.4 EXPLORATION AND EXPLOITATION TOKENS

Our token-based exploration proxy is a stylistic heuristic and is used only as suggestive evidence, not as a definitive behavioral measure. The full signature word set for exploration and exploitation is provided in Table 20. The *exploration rate* is then calculated using equation 3, where $N_{\text{explore}}$ is the total frequency of words from the exploration set and $N_{\text{exploit}}$ is the total frequency for the exploitation set.

$$\text{Rate}_{\text{explore}} = \frac{N_{\text{explore}}}{N_{\text{explore}} + N_{\text{exploit}}} \times 100\% \tag{3}$$

Table 20: Exploration vs Exploitation word sets

| | |
|---|---|
| **Exploration** | alternatively, what if, suspect, reconsider, re-examine, re-evaluate, perhaps, hypothesis, adjust, assume, invalidates, incorrect, but, however |
| **Exploitation** | confirm, verify, calculate, analyze, clearly, fit, compare, estimate, suggest, further, investigate, approximate, test |

### E.5 Symbolic Regression Baselines

For comprehensive analysis, we conducted further experiments on existing methods in symbolic regression: PySR and LLM-SR, within our benchmark. As law discovery from systems (where target law is embedded in model systems) is not supported by symbolic regression methods, we only tested it in our vanilla equation setting. Below, we report and analyze the results.

**PySR**   PySR (Cranmer, 2023) is an open-source Python library for symbolic regression that utilizes evolutionary algorithms to discover interpretable mathematical equations from data. We evaluated PySR across all 12 domains and 3 equation difficulty levels under the vanilla equation setting, with results illustrated in Figure 21. We observe that, in terms of symbolic accuracy, PySR yields performance comparable to or better than non-reasoning LLMs, while remaining significantly below all reasoning models. However, benefiting from its ability to conduct iterated regression, PySR achieved stronger relative performance, comparable to reasoning models in terms of RMSLE. Moreover, on easy equations, PySR outperformed all LLMs except Gemini-2.5-Pro in RMSLE, but it drops to fifth-best when facing hard equations. This indicates that this SR approach is fragile when discovering complex equations.

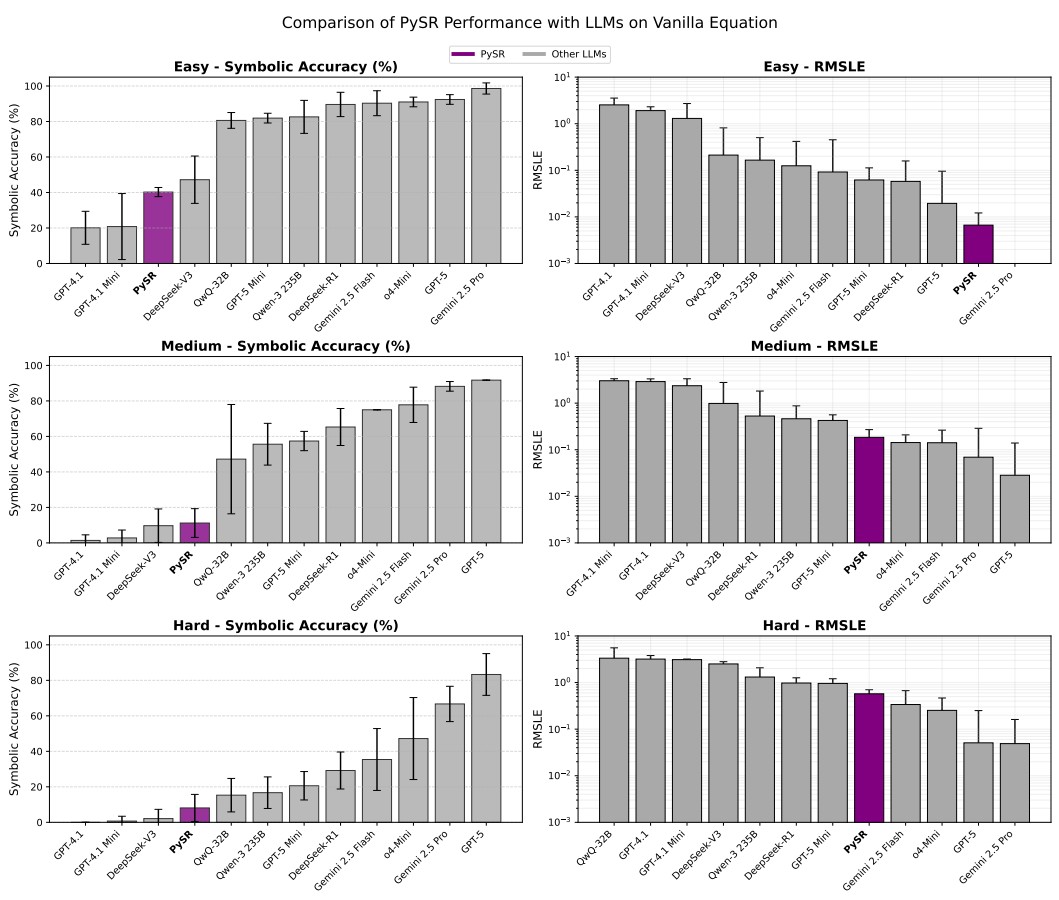

Figure 21: Performance of PySR in NEWTONBENCH (vanilla equation).

**LLM-SR**    LLM-SR (Shojaee et al., 2025a) is a symbolic regression framework that utilizes LLM as the foundation for equation discovery, supported by automated regression tools. In this study, we examine its performance after 2,500 iterations as recommended in the original paper[4].

Table 21 presents the experimental results of LLM-SR utilizing GPT-4.1-mini as the backbone. LLM-SR marginally surpasses PySR in both accuracy and RMSLE, a finding consistent with the results reported in the original study. However, despite requiring significantly higher computational resources, LLM-SR still underperforms our code-assisted agent in terms of accuracy.

Table 21: Performance Comparison of LLM-SR. *SA criteria for LLM-SR were relaxed. As raw outputs exhibited numerical and structural noise resulting in a zero score by the default judge, post-processing was applied (rounding to four decimal places and structural simplification).

| Method | LLM Backbone | Symbolic Accuracy (SA) % | | | | RMSLE |
|---|---|---|---|---|---|---|
| | | *easy* | *medium* | *hard* | *Avg.* | |
| PySR | – | 83.3 | 25.0 | 0.0 | 36.1 | $2.51 \times 10^{-4}$ |
| LLM-SR* | GPT-4.1-mini | 80.0 | 66.7 | 20.0 | 55.6 | **$3.89 \times 10^{-5}$** |
| Vanilla Agent (newtonbench) | GPT-4.1-mini | 41.7 | 0.0 | 0.0 | 13.9 | 2.27 |
| Agent w/ Code Asst. (newtonbench) | GPT-4.1-mini | **100.0** | **100.0** | **25.0** | **75.0** | $4.61 \times 10^{-2}$ |

Generally, symbolic regression methods leverage iterative regression and optimization to achieve superior RMSLE scores. Nevertheless, they remain sensitive to equation complexity and underperform our agent in symbolic accuracy, even while utilizing substantially greater computational resources.

---

[4]As LLM-SR is extremely resource-intensive (one LLM-SR inference requires 10,000 API calls, 1,000 times more than NewtonBench's Agent), we limit our testing to the simplest domain (Law of Sound Speed).

### E.6 Further Discussions

In this section, we further discuss NEWTONBENCH's implication in general scientific discovery and future directions of benchmarking.

**Implications for Scientific Discovery.** NEWTONBENCH evaluates the core cognitive architecture required for empirical research, mirroring the interplay of **inductive synthesis and deductive verification** in classical physics. The benchmark captures the intelligence displayed in foundational breakthroughs, such as J.J. Thomson's discovery of the electron via cathode ray manipulation. In this classical paradigm, discovery relies on the active isolation of variables to distill symbolic laws from physical behavior. This stands in contrast to the data-driven paradigm of modern High-Energy Physics (HEP)—exemplified by the statistical reconstruction of the Higgs boson from petabytes of collision data at the Large Hadron Collider (LHC)—which often necessitates high-dimensional analysis and probabilistic inference. Nevertheless, a unified epistemic challenge holds across both eras: the necessity to abstract parsimonious mathematical descriptions from observation. By simulating this fundamental loop, NEWTONBENCH positions itself not merely as a puzzle solver, but as a rigorous test of the reasoning engine required to recover the "source code" of physical phenomena.

However, it is important to note that while NEWTONBENCH provides the experimental environment, **real-world research necessitates the design of the experiment itself**. In authentic scientific inquiry, the interaction space is rarely pre-defined; researchers must conceive and construct the apparatus required to isolate a specific phenomenon. Consequently, the current evaluation focuses on the strategic manipulation of variables and the inference of laws within a bounded system, rather than the creative genesis of the experimental setup. While this limits the scope regarding the full pipeline of autonomous discovery, it allows for a precise measurement of the agent's ability to bridge the gap between tabletop experimentation and theoretical abstraction.

**Future Directions in Discovery Benchmarking.** To bridge the gap between this foundational reasoning test and the complexity of real-world scientific discovery, future benchmarks must embrace greater open-endedness. We envision the next generation of evaluations evolving along three critical dimensions:

- **First-Principles Model Construction:** Agents must advance beyond merely fitting parameters within a predefined symbolic grammar. Instead, they should be challenged to construct the model structure itself, deriving governing differential equations or symbolic laws directly from first principles.

- **Stochastic & Constrained Experimentation:** Benchmarks must simulate the practical "friction" of physical science. This requires agents to maximize information gain under strict budgets while navigating imperfect data environments characterized by instrument noise, stochasticity, and confounding factors.

- **Operational Laboratory Execution:** Data acquisition should evolve from simple function calls to complex procedural execution. Agents must reason about and manipulate the apparatus itself—ranging from **dry lab** tasks like debugging simulation pipelines to **wet lab** challenges like controlling robotic hardware for optical alignment—to create the conditions necessary for valid observation.

By establishing NEWTONBENCH as a controlled starting point, we aim to guide the community toward developing agents capable of navigating these increasingly demanding frontiers.

### E.7 USE OF LARGE LANGUAGE MODELS

In this manuscript, large language models were used exclusively for grammar checking. Additionally, image generation models were employed to enhance the visual presentation of figures.

