# OpenReview forum: "NewtonBench: Benchmarking Generalizable Scientific Law Discovery in LLM Agents"
_ICLR.cc/2026/Conference — ICLR 2026 Poster_

### Official Review · Reviewer_T5rR · 2025-10-15

**Soundness:** 3
**Presentation:** 2
**Contribution:** 3
**Rating:** 6
**Confidence:** 3

**Summary:**

This paper presents NewtonBench, which is a new, large-scale benchmark designed to assess and drive progress in LLM-based scientific law discovery. The benchmark introduces the “metaphysical shift” method, which use systematically mutating canonical physical laws to generate a suite of 324 interactive tasks spanning 12 domains. Unlike prior benchmarks focused on static function fitting, NewtonBench requires agents to engage in interactive experimentation within virtualized model systems, aiming to uncover hidden governing equations. Results across 11 leading LLMs reveal both the promise and current fragility of language models in scientific reasoning, especially as complexity and noise increase. The paper analyzes emergent behaviors such as the paradoxical impact of code assistance and provides an in-depth characterization of LLMs’ reasoning limitations in Automating Scientific Discovery.

**Strengths:**

1. Interesting task formulation: I think it is a valuable direction to test frontier agents' capability in autonomously make discovery especially in the domain of physics, the authors identified a crucial field worth exploring by the AI4Sci community at large and this direction is critical for future LLM agents for science research.
2. Evaluation and Discussion: The authors carefully evaluated a wide range of models and made a nice analysis of the exploration-exploitation tradeoff from observation to deeper reasoning.
3. Nice details: The work has quite a lot of details in appendix and shows considerable effort from the authors to make this a detailed presentation.

**Weaknesses:**

1. "Re-discover Newton's Laws": I'm a bit skeptical of how this central RQ is presented, because apparently the LMs have picked up what these laws mean in their pre-training corpora, and this is impossible to undo without systematic machine unlearning technique and access to these proprietary corpora.

2. I understand that the authors' reponse to 1. might be that the ”metaphysical shift" solves that by essentially permutating the laws into new forms, which appears different but remain physically valid. I'm also skeptical about this claim "fully prevent memorization" in 2 points:

a. How could we know that the permutated forms are not in training at all? Granted, it might be significantly less exposed than the original law, but is there any actual proof that these variants are unseen or not memorized "at all"? I imagine a validation experiment on this may be via elicitation or some other kind of contamination probing experimentation to offer more robust evidence as to whether models have memorized even some parts of these permutated variants.

b. The current explanation used philosophical references and counterfactual reasoning etc. which honestly are not convincing enough for me, maybe the authors would like to strengthen this part, also I don't feel like it's necessary to say it's "metaphysical shift" because in essence isn't that just data augmentation under governing physical principles? I feel this wording adds unnecessary complexity to understanding this, making it look more philosophical than it needs to be

3. The presentation is not super clear to me as to the 3 tier systems (vanilla-simple-complex), while the 3 tier permutation is more clear to me, my current understanding is that the systems are defined by the number of "assisting equations" towards the f_target which raise a number of questions

a. How do you guarantee that there's only one path towards f_target? often times there are many ways to discover/deduce the same laws of physics via different routes

b. Why is it justified to define level of complexity by this? I feel like the number of deduction may not be the best indicator for complexity, especially when these assisting equations aren't really explained or given sample of in a very clear manner in the current manuscript

As a result I think the permutation 3-level makes more sense to me than the other axis (experimental system etc.) Maybe the authors could improve the presentation of this part as it's very confusing for now.

**Questions:**

See weakness, additionally could the authors clarify and formalize the claim that metaphysical shifts fully prevent memorization? Has any empirical analysis (e.g., recall, overfitting checks) been performed especially with large, closed-source LLMs known for such a method to fully prevent memorization? (mitigate might be a more accurate wording? fully is a really strong wording that requires more evidence to back it up)

Also I feel like one suggestion I have for the authors is that the manuscript would be significantly easier to comprehend if they can offer visualization of a sample eval run, e.g. with a given model what exactly is the input-output of each step, the current discussion section e.g. have the dichotomous impact part which can be merged with the paradox discussion to save some space for this example. As it's highly non-trivial to illustate the key diff betwee "active" and "passive", a nice visualization of how any given agent actually do in terms of the concrete input-output would significantly strengthen my understanding of their approach.

In general, I feel this paper is exploring an interesting+valuable direction but could really benefit a lot from better presentation, therefore I recommend a boardline rating on the positive side with more room for improvement. Overall I appreciate the design of this approach by the authors but this paper could really benefit from better articulation for larger impact.

---

> ### Author Response · Authors · 2025-11-22
>
> Dear Reviewer T5rR,
>
> Thanks for your thoughtful review and acknowledgement in:
> - The task formulation and importance.
> - Comprehensive evaluation and discussion.
> - Detailed presentation in appendix.
>
>
> We hope the following revisions and clarifications addresses your concerns!
>
> **[Revision Summary]** In short, (1) we conduct DataBlind experiment on Newtonbench to further demonstrate that it is **fully** memorization-free. (2) We remove excessive philosophical discussions to improve readability. (3) Include more details regarding system-level complexity setting.
>
> ---
>
> >Q1. "Re-discover Newton's Laws": I'm a bit skeptical of how this central RQ is presented...
>
> **A1:** Yes, the explanation in Q2 is right. Our curation of novel equations enabled evaluation on discovery of physical laws (the term “re-discover” makes our motivation more intuitive). Without our approach, the discovery of such established physical laws cannot be evaluated due to memorization.
>
> ---
> >Q2a: It might be significantly less exposed than the original law, but is there any actual proof that these variants are unseen or not memorized "at all"?
>
> **A2(a):** Thanks for the question. As we perform counterfactual shifts on the equations, we believe it’s sound to say that our benchmark is “memorization-free”, with reasons as follows:
> - In literature of knowledge reasoning benchmarks [1] [2] [3] [4] , counterfactual edits/shifts are an established approach to avoid memorization / contamination in LLMs, and there’s no discussion on “how much contamination is left” or “whether counterfactual edit fully avoid memorization”.
>
> - To further validate the memorization-free feature of NewtonBench, we conduct a DataBlind setting for NewtonBench in **Appendix E.1**. The result is clear:
>
> | Model| LSR-Transform | NewtonBench (no hint) | NewtonBench (hint*) |
> |-|-|-|-|
> |GPT-4o-mini| 7.21| 0.0| 0.0 |
> |Nemotron-Ultra-253B| 28.83| 0.0 | 0.0 |
> |GPT-4.1 | 35.14| 0.0 | 0.0  |
>
> *with prompting hints indicating that the ground‑truth physical laws are shifted.
>
> ---
> >Q2b:  I feel the philosophical wording adds unnecessary complexity to understanding this...
>
> **A2(b):** Thanks for the suggestion. We agree that excessive philosophical discussions (section 3.1, paragraph 2) can potentially overwhelm readers. In our revision, we directly use the term “counterfactual shifts” in our paper for clarity, and removed the excessive discussions.
>
> ---
>
> >Q3: Questions regarding our system-level complexity setting (Vanilla → Simple → Complex).
>
>
> **A3:** Thank you for raising these points. We agree that the system-level axis required clearer exposition. Below we clarify the curation logic and the rationale behind our definition of system-level complexity:
>
> - Our experimental systems are curated by domain experts with explicit paired instructions (simple vs. complex) and sanity checks. Thus, each domain contains two empirically distinguishable system setups, independent of any particular inference path an agent might follow.
>
> - The primary observable difference between these paired systems is that complex systems require more assisting equations and operations to reach the target equation. Empirically:
>
> || Simple Sys.| Complex Sys.|
> |-|-|-|
> |Avg. # of Assisting Equations | 1.75 | 2.42 |
> |Avg. # of Operations| 5.17 | 8.17 |
>
> - We do not require or assume a *unique* path to the target equation. Instead, we use assisting-equation and operation counts as *system properties*—features of the curated setup—rather than properties of the reasoning trajectory an LLM may choose. This avoids over-constraining the problem while still capturing meaningful structural differences between systems.
>
> - We do not use “deduction steps” as a complexity metric because LLM agents follow diverse reasoning strategies, making such steps non-comparable and non-repeatable across models.
>
> - Empirically, Table 2 shows consistent performance degradation from Vanilla → Simple → Complex across nearly all models and equation tiers. This validates that the curated system tiers successfully stratify task difficulty in practice.
>
> - Finally, our system-level tiering is intended as a practical task classification for controlled analysis, not a claim about a universal or theoretical notion of complexity. Since system complexity is not our primary scaling axis, empirical separability and transparent curation are the key goals.
>
>
> We have incorporated the details in our revised manuscript (**Appendix A2**).
>
> >Q4. Visualization for case studies.
>
> **A4:** Thanks for the suggestion. We're working on more intuitive case studies and will later update the manuscript accordingly.
>
> ---
>
> We sincerely appreciate your valuable suggestions and thoughtful feedback. If you have any further questions or require additional clarification, we would be glad to provide it. We kindly ask you to consider adjusting your score if our revisions and clarifications adequately address your concerns.

---

> > ### Comment · Reviewer_T5rR · 2025-11-23
> >
> > Thanks for your response, could you elaborate a bit more on DataBlind experiment as it's not super clear to me what the setting for such experiment entails

---

> > > ### Author Response · Authors · 2025-11-23
> > >
> > > Thanks for following up!
> > >
> > > Datablind is a direct prompting method that requires LLM agents to **generate hypotheses purely from contextual information** (concepts of input/output variables, etc), **without access to actual experimental data**. It was adopted in previous work [1] to assess memorization in LLMs.
> > >
> > > ---
> > > Here's an example prompt used in [1] (LSR-Transform):
> > >
> > > ```Find an equation in the field of classical mechanics that describes the mass ( m ) needed to store energy in an oscillating system, given physical input variables: mean stored energy ( E_n ), driving frequency ( omega ), natural frequency (omega_0), and amplitude ( x ).```
> > >
> > > Under this configuration, the model must infer the governing equation solely from its “memory,” without any possibility of deriving it from provided data. This design ensures that the evaluation isolates memorization and prevents contamination from data-driven inference. However, LSR-Transform fails this contamination test (its DataBlind score remains high), which makes some subsequent works [2] failed to adopt it as a contamination-free evaluation.
> > >
> > > ---
> > >
> > > In our experiment, we adjusted our prompt (hint) to explicitly inform the LLMs that the physical law is shifted, yet all LLMs still achieve zero accuracy, demonstrating the memorization-free nature of NewtonBench.
> > >
> > > We have updated the discussion accordingly in **Appendix E5**.
> > >
> > > [1] LLM-SRBench: A New Benchmark for Scientific Equation Discovery with Large Language Models, ICML 2025.
> > > [2] SR-Scientist: Scientific Equation Discovery With Agentic AI, arxiv 2025.

---

> > > ### Comment · Reviewer_T5rR · 2025-11-23
> > > **Recommend Acceptance**
> > >
> > > Great, I encourage the authors to further polish the presentation to be less "over-intellectualizing/philosophical" and resonate/implicate with the physical science even more (e.g. elaborate more on the implication for scientific discovery in the realm of physics, how it might generalize to other natural science, and what your future versions of more realistic, more demanding discovery benchmarks might entail)
> > >
> > > Nevertheless, I believe this paper warrants attention of the field and should be presented at ICLR.
> > >
> > > I recommend acceptance to the AC

---

> > > > ### Comment · Reviewer_T5rR · 2025-11-23
> > > > **Raised Score to 8**
> > > >
> > > > In accordance with my recommendation for acceptance, I've updated my score to 8, another point I forgot to write above is that mayhaps the authors want to tone down a bit with something adjacent to a scope explanation to more accurately position this work as a beginning point for further scientific discovery related bench as future work.

---

> > > > > ### Author Response · Authors · 2025-11-26
> > > > > **Follow-up by Authors**
> > > > >
> > > > > Dear Reviewer T5rR,
> > > > >
> > > > > Thanks for the acknowledgement and thoughtful feedback. We are happy to address your follow-up concerns with following revisions.
> > > > >
> > > > > ---
> > > > > **[Revision Summary]**
> > > > >
> > > > > To enhance the scientific implication in the realm of physics:
> > > > >
> > > > > - We introduced a fine-grained taxonomy for classification of "physical plausibility levels" for all shifted laws.
> > > > >
> > > > > - We included a new section to discuss the general implication in scientific discovery and future directions of discovery benchmarking.
> > > > >
> > > > > Furthermore, upon your previous comments regarding case visualization:
> > > > >
> > > > > - We included a visualized figure that intuitively compare the passive observation and active exploration paradigm.
> > > > >
> > > > > ---
> > > > > **Revision 1.  Introduced classification on Physical Plausibility for enhanced real-world physical implication.**
> > > > >
> > > > > To cater to future research that requires more realistic physical environments, we provide a fine-grained classification taxonomy for the "physical plausibility" of each shifted laws. We measure "physical plausibility" with the compatibility of the shifted law with real-world physics. A law exhibits lower physical plausibility if it violates more fundamental physical principles in the real world. The three plausibility levels are defined as follows:
> > > > >
> > > > > - **Level 1 (no structural violation)**: Level 1 consists of admissible counterfactuals that modify the functional form while strictly preserving the original law's finiteness, real-valued outputs, and fundamental symmetries like action-reaction.
> > > > > - **Level 2 (minor structural violation)**: Level 2 includes laws that maintain mathematical regularity and finiteness but diverge from standard physical theory by altering essential symmetries or reciprocity relations required for conservation.
> > > > > - **Level 3 (major structural violation)**: Level 3 characterizes laws with significant structural anomalies, such as the introduction of singularities or physically invalid value ranges. These deviations challenge the minimal regularity and positivity constraints typically required in real-world physical modeling. However, because dimensional homogeneity is preserved, these candidates remain eligible for scientific law discovery benchmarking.
> > > > >
> > > > > The plausibility classification serves as a **practical guideline for future applications** of NewtonBench. It allows researchers to select specific subsets for objectives extending beyond standard scientific law discovery benchmarking. For instance, Level 1 provides an ideal environment for scientific agent training or generalized physical reasoning (where tolerance for structural violations is low), while Level 3 is well-suited for stress-testing symbolic regression models or exploring counterfactual reasoning.
> > > > >
> > > > > Here's some statistics of the categorization result:
> > > > > | Plausibility level \(from high to low\)  | Easy | Medium | Hard | Total |
> > > > > |----|:----:|:------:|:----:|:-----:|
> > > > > | Level 1  | 30   | 23     | 17   | 70    |
> > > > > | Level 2  | 3    | 5      | 6    | 14    |
> > > > > | Level 3 | 3    | 8      | 13   | 24    |
> > > > > | **Total** | 36   | 36     | 36   | 108   |
> > > > >
> > > > > Detailed definitions, discussions, implications, and classification results are provided in **Appendix A.3.4**.
> > > > >
> > > > > ---
> > > > > **Revision 2. Extended discussion on scientific implication and future directions.**
> > > > >
> > > > >
> > > > > We added a "Further Discussions" section in **Appendix E6** to address scientific implication of NewtonBench's Results and highlighted the future directions for discovery benchmarking.
> > > > >
> > > > > **Key Takeaway**: While NewtonBench evaluates discovery within a fixed experiment tool, bridging the gap to authentic discovery requires future benchmarks to demand **autonomous experimental design and model construction from first principles**. NewtonBench is just a starting point.
> > > > >
> > > > > ---
> > > > > **Revision 3. Case Visualization.**
> > > > >
> > > > > As we previously discussed, we added an intuitive visualization case that compares the paradigms of passive observation and active exploration, in **Appendix D**.
> > > > >
> > > > > ---
> > > > > We will selectively incorporate those details into the main content in the final version.
> > > > >
> > > > > Thanks again for your insightful suggestions and recognition!

---

### Official Review · Reviewer_9pai · 2025-10-27

**Soundness:** 2
**Presentation:** 3
**Contribution:** 2
**Rating:** 2
**Confidence:** 4

**Summary:**

The paper proposes NewtonBench, a benchmark for evaluating large language models (LLMs) on discovering physical laws through interactive experimentation. It introduces metaphysical shifts, i.e. a transformation method applied to canonical physics equations (for example, modifying exponents or couplings), to generate 108 shifted laws across 12 physics domains. Each law is instantiated in three configurations: 1) Vanilla Equation: the shifted law alone, 2) Simple System: the law embedded in a minimal experimental setup,
3) Complex System: the law coupled with additional equations. Together these form 324 total tasks. The authors also provide a formal proof that all tasks are finitely solvable.

LLMs interact with each environment by adjusting variables and observing outputs to infer the hidden governing law. Performance is evaluated by symbolic accuracy and robustness to noise. Eleven popular LLMs were tested, showing strong performance on simple cases but substantial drops for complex or noisy systems.

Main contributions include:
1. A large-scale, interactive benchmark for equation discovery.
2. The metaphysical shift method for creating diverse, solvable tasks.
3. A comprehensive evaluation of leading LLMs’ capabilities on the proposed benchmark.

**Strengths:**

1. The paper focus on interactive scientific discovery, which is an important and relatively underexplored area.
2. It proposes an effective method for generating new equations and provides a formal proof of their finite solvability.
3. The evaluation is thorough, covering a wide range of models.

**Weaknesses:**

1. The proposed "metaphysical shifts" fail to create a "physically plausible universe" and ultimately compromise the scientific relevance of the task. The core problem is that the physics descriptions become meaningless once the dimensional structure and proportional dependencies of core quantities are altered. In the laws altered by NewtonBench (such as Newton’s Law of Universal Gravitation, Coulomb’s Law, Heat Transfer, and Hooke’s Law), most are phenomenological in nature and contain at least one parameter or coefficient whose physical identity is defined by its role within that specific law. Examples include the gravitational constant, the permittivity, the thermal conductivity, or the spring constant. These quantities do not possess independent operational definitions outside the context of the laws that introduce them.
Therefore, when the functional form of such a law is altered (for instance, by changing exponents or couplings), those parameters lose their original physical meaning, since their identity is inseparable from the structure of the law itself. The modified equations thus describe mathematically consistent systems but no longer preserve the physical semantics of the original quantities or interactions. This reframes the task from a genuine physics discovery challenge into interactive equation discovery within an abstract, non-physical framework.

2. The so-called “model system discovery” in NewtonBench does not introduce a fundamentally different epistemic or physical level of reasoning. It merely increases equation compositionality (the number of nested or coupled expressions), which can always be flattened into a single symbolic equation. In real physics, a model system is defined by time-evolving state variables, interactions governed by symmetries or conservation laws, and differential equations that capture causality and dynamics. In contrast, the relations in NewtonBench reduce to static algebraic mappings between scalar quantities, making its “complex systems” only syntactically more complicated rather than fundamentally different, as they remain within the same layer of symbolic algebra.

**Questions:**

Please refer to the weaknesses.

---

> ### Author Response · Authors · 2025-11-22
>
> Dear Reviewer 9pai,
>
> Thanks for your thoughtful review and acknowledgement in:
> - Scale and interactive nature of NewtonBench.
> - Our data curation approach of law shifts.
> - Thorough evaluation and analysis.
>
> Regarding the two concerns raised, we view them primarily as **conceptual and philosophical questions** about what constitutes a “physically plausible universe,” rather than critiques of the **technical goals and contributions of NewtonBench**. Nonetheless, we appreciate the reviewer’s perspective, and we have added clarifications to the manuscript to better articulate the intended scope of the benchmark and to prevent potential misunderstandings about its design assumptions.
>
> > Q1(a). the proposed law shift approach failed to create a “physically plausible universe” … (due to the shifted unit in physical constants).
>
> **A1(a):** The notion of a “physically plausible universe” is not a formally defined concept in physics. We also believe that the scientific value of NewtonBench does not depend on such philosophical notions. To avoid confusion, we have revised **Section 3.1** and removed language related to “plausibility” and “metaphysics,” keeping the focus on the benchmark’s concrete and well-defined technical objectives.
>
>
>
>
> ---
>
> > Q1(b). “... and ultimately compromise the scientific relevance”
>
> **A1(b):** We understand the concern, but we do not believe that shifting the units of physical constants diminishes the scientific relevance of the benchmark. The scientific process does not rely on specific numerical values or units of constants, but on the structure of reasoning, experimental design, and inferential rigor.
>
> Compared with existing work on generalizable scientific law discovery (e.g., LSR‑Synth), NewtonBench offers enhanced scientific relevance in two key respects:
>
> - It **emphasizes the scientific method**—hypothesis formation, experimental design, interventional reasoning, and synthesis—rather than mere function fitting or symbolic regression.
>
> - It is **grounded in canonical physical variables and constraints**. Tasks use real physical quantities (mass, charge, temperature, area, distance), enforce dimensional consistency and domain admissibility, and incorporate real assisting laws to structure interventions. Although the target law is altered, LLM agents must still reason with the same categories of variables, units, and confounders encountered in genuine physics experimentation, rather than arbitrary symbolic constructs.
>
> We have included further discussions in **Ethics Statement**.
>
>
>
> ---
> >Q2. In physics, model system is defined by time-evolving state variables…; while newtonbench merely increase equation compositionality that flattened into a single equation.
>
>
> **A2:** It is a **field‑specific convention**—not a necessary condition of modelhood—to require **time‑evolving state variables** for a physical model. Extensive precedent and scholarship [1–4] establish the legitimacy of **static (equilibrium or constitutive) models** within physics. Accordingly, our use of “model” and “model system” in NewtonBench is both standard and appropriate.
>
> Moreover, NewtonBench introduces a **distinct paradigm** in the equation‑discovery landscape: it emphasizes **interventional identifiability under confounding** achieved through active, agent‑driven experimentation, in contrast to the largely passive curve‑fitting used in prior work. Thus, the model‑system setting in NewtonBench is not only **terminologically valid** but also **substantively important** for evaluating scientific reasoning.
>
> [1] R. Frigg and S. Hartmann, “Models in Science,” Stanford Encyclopedia of Philosophy, 2006.
> [2] N. Cartwright, *How the Laws of Physics Lie*, Oxford University Press, 1983.
> [3] H. B. Callen, *Thermodynamics and an Introduction to Thermostatistics*, 2nd ed., Wiley, 1985.
> [4] L. D. Landau and E. M. Lifshitz, *Electrodynamics of Continuous Media*, 2nd ed., Pergamon/Elsevier, 1984.
>
> ---
> We sincerely appreciate your valuable suggestions and thoughtful feedback. If you have any further questions or require additional clarification, we would be glad to provide it. We kindly ask you to consider adjusting your score if our revisions and clarifications adequately address your concerns.

---

> > ### Comment · Reviewer_9pai · 2025-11-23
> >
> > I appreciate the authors' response to my previous comments. However, it seems there are some misunderstandings regarding my concerns, which I would like to clarify as follows.
> >
> > **1. Clarification on "Scientific Relevance" and Conceptual Grounding**
> >
> > My first point is not a philosophical one, but fundamentally concerns the key motivation, claims, and contribution of NewtonBench.
> > In both the introduction and abstract, scientific relevance is highlighted as a key feature of NewtonBench. The paper states that NewtonBench constructs “laws that are conceptually grounded but physically novel” and “maintains the scientific relevance of real-world principles.”
> >
> > How are these laws grounded? According to Appendix A.2, the grounding is primarily through the variable names and their physics descriptions. These descriptions are intended to provide the conceptual basis that ensures the “scientific relevance” of the constructed equations. Thus, these descriptions become central to the validity and meaningfulness of the dataset.
> >
> > However, the variable descriptions in many of the constructed equations do not actually remain “conceptually grounded” nor do they “maintain scientific relevance.”
> >
> > Consider Fourier’s law in the thermodynamics domain (Table 5). The original equation is: $Q = k A\Delta T / d$
> > , where $Q$ is the rate of heat transfer, and $k$ is designated as "thermal conductivity" (Table 7).
> >
> > By **definition**, the rate of heat transfer ($Q$) is determined by the temperature difference ($\Delta T$): it represents energy transfer caused by that temperature difference. Thermal conductivity $k$ is **defined** as the proportionality constant linking $Q/A$ to $\Delta T/d$. This is why its unit is $W/(m \cdot K)$.
> >
> > Now consider a shifted equation, such as $Q = k' (A+\Delta T) / d^2$ (Table 5). This formulation breaks the fundamental relationship. For instance, the equation suggests $Q$ could be non-zero even if the temperature difference $\Delta T=0$ (provided $A \neq 0$), which violates the definition of heat transfer. Moreover, the quantity $k'$ in this new equation is no longer the physical "thermal conductivity". Assigning the physical description of "thermal conductivity" to this new parameter $k'$ thus provides misleading information about its role and physical meaning.
> >
> > Consequently, the claimed "scientific relevance" is weakened, reducing the task to equation discovery with ungrounded/misleading information about the variables.
> >
> > **2. Distinction Between "Model Discovery" and "Equation Discovery"**
> >
> > My second point concerns whether the proposed “model discovery” paradigm is meaningfully distinct from equation discovery. Since the models presented in the paper can be represented as stationary, composed equations, it is unclear what the substantive differences are.
> >
> > ---
> > In conclusion, I acknowledge that this work constructs new equations (similar to LLM-SRBench [1]) and introduces an interactive equation discovery scenario (similar to PhysGym [2]). These are valuable contributions.
> > However, the current claims and positioning, particularly those emphasizing strong "scientific relevance" and "conceptual grounding" of the novel laws via variable descriptions, appear inaccurate and require a major revision to align with the actual nature of the dataset and task.
> >
> > [1] LLM-SRBench: A New Benchmark for Scientific Equation Discovery with Large Language Models, ICML 2025.
> > [2] PhysGym: Benchmarking LLMs in Interactive Physics Discovery with Controlled Priors, arXiv 2025.

---

> > > ### Author Response · Authors · 2025-11-26
> > > **Follow-up by Authors (1/2)**
> > >
> > > Dear Reviewer 9pai,
> > >
> > > Thanks for following up. We would like to address your concerns with following revision and clarifications:
> > >
> > > ---
> > > **[Revision Summary]**
> > > Although NewtonBench's scientific implication lies mainly in incorporating the scientific method and the nature of discovery tasks within a generalization framework, we agree that enhancing its implications for real-world physics is also valuable.
> > >
> > > To this end, we have performed **two substantial revisions**: First, we revised the counterfactual law design to address fallacy under nullification test. Second, we introduced a fine-grained taxonomy for **classification of "physical plausibility levels"** for all shifted laws.
> > >
> > > Moreover, we address your other concerns with further clarifications.
> > >
> > > ---
> > > **Revision 1. Redesigned laws with fallacy under nullification test** (e.g., $Q \neq 0$ while $\Delta T = 0$ in the Fourier's Law example)
> > >
> > > We **redesigned** the laws in NewtonBench that exhibit fallacies under the nullification test and re-evaluated all LLM agents on this revised set. The full set of laws is updated in **Table 5**, and the new result of all LLMs are updated in **Table 2**. We will also update other sections accordingly in the revision.
> > >
> > >
> > > ---
> > > **Revision 2. Introduced classification on "physical plausibility" for enhanced real-world physical implication.**
> > >
> > > To cater to future research that requires more realistic physical environments, we provide a fine-grained classification taxonomy for the "physical plausibility" of each shifted laws. We measure "physical plausibility" with the **compatibility of the shifted law with real-world physics**. A law exhibits lower physical plausibility if it violates more fundamental physical principles in the real world. All 108 laws are categorized into three plausibility levels:
> > >
> > > - Level 1. Admissible Counterfactual Laws (No Structural Violation).
> > >
> > > - Level 2. Counterfactual Laws with Minor Structural Violation.
> > >
> > > - Level 3. Counterfactual Laws with Major Structural Violation.
> > >
> > > The plausibility classification serves as a **practical guideline for future scientific applications** of NewtonBench. It allows researchers to select specific subsets for objectives extending beyond standard scientific law discovery benchmarking. For instance, Level 1 provides an ideal environment for scientific agent training or generalized physical reasoning (where tolerance for structural violations is low), while Level 3 is well-suited for stress-testing symbolic regression models or exploring counterfactual reasoning.
> > >
> > > Here's some statistics of the categorization result:
> > > | Plausibility level \(from high to low\)  | Easy | Medium | Hard | Total |
> > > |----|:----:|:------:|:----:|:-----:|
> > > | Level 1  | 30   | 23     | 17   | 70    |
> > > | Level 2  | 3    | 5      | 6    | 14    |
> > > | Level 3 | 3    | 8      | 13   | 24    |
> > > | **Total** | 36   | 36     | 36   | 108   |
> > >
> > > Detailed definitions, discussions, implications, and classification results are provided in **Appendix A.3.4**.
> > >
> > > ---
> > > **Q3.** "$k$ is designated as "thermal conductivity" (Table 7).... while the quantity in the shifted equation $k'$ is no longer the physical thermal conductivity."
> > >
> > > **A3:** There seems to be a misunderstanding of our content. The sampling range in Table 7 is merely a heuristic setting for numerical stability of the data fidelity evaluation (RMSLE). Even if we sample from arbitrary range, the evaluation is still valid and sound.
> > >
> > > Therefore, the description of $k$ in Table 7 (currently Table 9) does not imply the definition of the updated constant $k'$ in the shifted laws. Physical constants with shifted units do not share the same definitions as constants "in our universe." We have updated the description in the sampling distribution section (Appendix A.4.2) to avoid confusion.
> > >
> > > ---
> > >
> > > **Q4.** Regarding the phrase "conceptually grounded".
> > >
> > > **A4:** We have already discussed this term in our previous response A1(b), noting that our tasks "use real physical quantities (mass, charge, temperature, area, distance), enforce dimensional consistency and incorporate real assisting laws."
> > >
> > > By "conceptually grounded," we refer to the **semantic consistency** of the physical variables with the real world. While the target interaction law (e.g., the specific formula for gravity) is modified, the variables themselves are not arbitrary; they remain constrained by invariant "assisting equations" (standard physical laws like Newton's Second Law or kinematic relations). Consequently, a variable representing "mass" in the simulation continues to govern inertia according to $F=ma$, thereby anchoring the shifted laws to valid real-world physical concepts.
> > >
> > > ---
> > >
> > > (1/2)

---

> > > ### Author Response · Authors · 2025-11-26
> > > **Follow-up by Authors (2/2)**
> > >
> > > **Q5. Why "unit change in physical constants" does not diminish scientific relevance.**
> > >
> > >
> > > **A5:** We respectfully disagree with the reviewer's argument that the unit change in physical constants diminishes scientific relevance. The specific units of a physical constant are not the **causal factors** of a scientific law, but rather the **consequential artifacts** required to satisfy **Dimensional Homogeneity**.
> > >
> > > Our position is grounded in the epistemology of scientific discovery and the principles of Dimensional Analysis, specifically formalized by **the Buckingham $\pi$ theorem** [1] and **P.W. Bridgman’s Operationalism** [2]. According to these established theories, a physical law consists of two distinct components with different roles:
> > >
> > > - **The Structural Relationship** (The Causal Mechanism): This is the functional form identifying how variables interact (e.g., $F \propto m_1m_2/r^2$). This structure represents the actual physical mechanism and the "cause" of the observed behavior.
> > > - **The Dimensional Constant** (The Consequential Offset): This is a scaling parameter derived *ex post facto*. Its primary role is to serve as a "coupling parameter" or dimensional compensator to reconcile the dimensions of the dependent variables with the independent variables.
> > >
> > > Historically, the discovery of physical laws proceeds via **empirical induction where the structure precedes the constant**. For example, Isaac Newton identified the structural relationship of gravity ($F \propto r^{-2}$) in the 17th century, yet the gravitational constant ($G$) and its specific units were not determined until Henry Cavendish’s experiments over a century later. The law existed effectively before the constant was dimensionally defined.
> > >
> > > Therefore, our approach—where the law shifts and the physical constant compensates for the unit change—does not "diminish" scientific relevance. On the contrary, it **perfectly follows the scientific principle** that the structural relationship is the primary discovery, while the units of the constant are a mathematical necessity derived to bridge the gap between input and output dimensions.
> > >
> > > Note: We have **already clarified** the key factors contributing to NewtonBench's scientific relevance (e.g., the scientific method) in our previous response A1(b).
> > >
> > > ---
> > >
> > > **Q6. Distinction Between "Model Discovery" and "Equation Discovery".**
> > >
> > > **A6:** The distinction between the two settings is both **conceptual** and **empirically validated**.
> > > - **Conceptually**, model discovery introduces a **substantive additional inference problem** that standard equation discovery does not face: the agent must **factor the computation graph and isolate the effective input–output map of $f_{\text{target}}$** from system-level interventions and final outputs in the composed model $M = (f_1,\dots,f_k)$ before any symbolic regression step is even well-posed. This identification step is non-trivial and mirrors real-world scientific law discovery [3].
> > >
> > > - **Empirically**, this is not just a notational change: as shown in Table 2, moving from Vanilla Equation to Complex System settings yields large, systematic drops in symbolic accuracy for all models (e.g., Gemini‑2.5‑Pro from 71.5% → 13.9% on hard equations), even though the overall composed law exists in closed form. This consistent gap shows that “model discovery” introduces a **substantively harder regime** than standard equation discovery.
> > >
> > >
> > > ---
> > > **Q7.** Discussions of existing works.
> > >
> > > **A7:** LLM-SRBench [4]: LLM-SRBench have been extensively discussed in our manuscript.
> > >
> > > PhysGym [5]: PhysGym is a concurrent work with our paper. Though both involve interactive experimentation, PhysGym focuses on prior reliance (anonymizing existing discoveries), which fundamentally differs from our emphasis on generalization and the discovery of novel laws. We will add discussion of PhysGym to our paper.
> > >
> > >
> > >
> > >
> > >
> > >
> > >
> > >
> > > ---
> > > [1] Buckingham, E. On Physically Similar Systems; Illustrations of the Use of Dimensional Equations. Physical Review, 1914.
> > > [2] Bridgman, P. W. The Logic of Modern Physics. Macmillan, 1927.
> > > [3] Batterman. R. W. Asymptotics and the Role of Minimal Models. British Journal for the Philosophy of Science, 2002.
> > > [4] Shojaee et al. LLM-SRBench: A New Benchmark for Scientific Equation Discovery with Large Language Models, ICML 2025.
> > > [5] Chen et al. PhysGym: Benchmarking LLMs in Interactive Physics Discovery with Controlled Priors, arXiv 2025.
> > >
> > > ---
> > > (2/2)

---

### Official Review · Reviewer_cQE7 · 2025-10-29

**Soundness:** 3
**Presentation:** 3
**Contribution:** 3
**Rating:** 6
**Confidence:** 4

**Summary:**

This paper introduces **NewtonBench**, a benchmark for evaluating LLMs' scientific law discovery capabilities that addresses fundamental limitations in existing approaches. Current benchmarks suffer from a methodological trilemma (forcing trade-offs between scientific relevance, scalability, and memorization resistance) and oversimplify discovery as static function fitting rather than interactive exploration. NewtonBench resolves these issues through two key innovations: (1) **metaphysical shifts** that systematically mutate canonical physical laws to generate 324 scientifically grounded yet memorization-resistant tasks across 12 physics domains, and (2) **interactive model discovery** requiring agents to actively probe virtual environments to uncover hidden laws embedded in complex systems with confounding variables. Evaluation of 11 state-of-the-art LLMs reveals that while frontier models like GPT-5 and Gemini-2.5-pro demonstrate emerging capability (65-73% accuracy), their performance degrades precipitously with increasing complexity, exhibits extreme sensitivity to observational noise, and paradoxically suffers when given code assistance due to premature exploitation over exploration. The benchmark demonstrates that robust, generalizable scientific discovery in complex, interactive environments remains the core unsolved challenge for AI-driven science.

**Strengths:**

This paper exhibits **high originality** through its novel benchmark design principles, **strong quality** in experimental rigor and scope, **good clarity** in presentation and visualization, and **substantial significance** for both AI research and automated science. The work makes meaningful contributions across all four dimensions, with particularly notable strengths in originality (metaphysical shifts resolution of trilemma) and clarity (outstanding visual communication and narrative structure). The significance is enhanced by the timeliness of addressing LLM scientific reasoning at a critical juncture in model capability development.

**Weaknesses:**

### LLM-as-Judge for Primary Metric Introduces Circularity

**Problem**: Symbolic Accuracy, the main evaluation metric, relies on LLM verification of equation equivalence. Using LLMs to judge LLM-generated discoveries creates potential systematic biases.

**Actionable Fixes**:

1. Provide detailed error analysis: On what types of equations does LLM-as-judge fail? Are these random or systematic?
2. Report inter-rater reliability among human experts
3. Include traditional symbolic equivalence checking (even if imperfect) as a secondary validation
4. Test whether different judge models (GPT vs. Gemini vs. Claude) produce consistent verdicts

### Framing Claims "Scientific Law Discovery" but Evaluates Only Physics Equations

**Problem**: The paper is titled "Benchmarking Generalizable **Scientific Law Discovery**" and makes broad claims about "AI-driven science," "automated science," and "genuine scientific intelligence" throughout. However, the evaluation exclusively covers **physics equations**—12 domains, all closed-form algebraic expressions. This creates a fundamental misalignment between claims and evidence. Scientific discovery encompasses far more than physics equations: chemistry involves molecular structures and reaction mechanisms, biology includes qualitative evolutionary principles and statistical patterns, social sciences rarely use closed-form equations, and even within physics the benchmark excludes quantum mechanics, field theories, differential equations, and computational models. The paper's significance claims about measuring "scientific intelligence" and guiding development of agents "capable of genuine scientific discovery" substantially overreach what physics equation discovery can demonstrate.

**Actionable Fixes**:

(1) **Reframe title and claims** to match actual scope—change to "Benchmarking Physics Equation Discovery" or "Mathematical Law Discovery"; (2) **Add explicit scope limitations** in Abstract/Introduction acknowledging this represents one facet of scientific discovery; (3) **Discuss generalization boundaries**—which findings likely transfer to non-physics domains (e.g., noise sensitivity) versus which are physics-specific (e.g., dimensional consistency); (4) **Provide expansion roadmap** for chemistry (reaction mechanisms), biology (population dynamics), or qualitative theory discovery to validate claimed generalizability. Honest scoping doesn't diminish the contribution—it makes claims defensible and clarifies what the benchmark actually measures: rigorous evaluation of mathematical law discovery from experimental data, a valuable but bounded capability for AI-assisted science.

**Questions:**

Please see the Weaknesses.

---

> ### Author Response · Authors · 2025-11-22
>
> Dear Reviewer cQE7,
>
> Thanks for your thoughtful review and acknowledgement in:
> - Originality and Novelty of NewtonBench.
> - Strong quality in experiment rigor.
> - Clarity in presentation and visualization.
> - Significance and impact across AI and Science domains.
>
> We hope the following revisions and clarifications addresses your concerns!
>
> **[Revision Summary]** In short, (1) we further analyzed the robustness of LLM-as-a-Judge from three perspectives, and (2) we further discussed the impact scope and generalization in scientific law discovery. All revisions have been updated in the manuscript.
>
> ---
>
> >Q1. LLM-as-Judge for Primary Metric Introduces Circularity
>
> **A1:** Using LLM-as-a-Judge as the primary metric for symbolic accuracy follows previous practice in the literature [1]. From our own experience, symbolic approaches are extremely fragile and require strong format restrictions on LLM responses, thus are not applicable to our scenario.
>
> We conducted several experiments/analyses on LLM-as-a-Judge for further validation:
> - **Style Robustness Analysis** (Is the LLM judge robust across different law styles?) Answer: Yes. The LLM judge is quite robust to syntactic variations.
> - **Impact of Few-shot Learning** (Does performance improve with more demonstrations?) Answer: Yes. More demonstrations lead to better performance.
> - **Error Analysis** (Representative cases of false positives/negatives.) Findings: Errors are quite **random** (even with temperature=0). Simply using self-consistency or bootstrapping could resolve these problems. However, current performance (98.3%) is already good enough.
>
> We argue that all frontier LLMs (e.g., GPT-5, Gemini-2.5/3, etc.) should be fully capable of performing judgment. However, regarding detailed performance and settings across all models, we would like to leave this work to LLM-as-a-Judge research.
>
>
> Detailed results and analyses are presented in **Appendix A5**.
>
> ---
> >Q2. Framing Claims "Scientific Law Discovery" but Evaluates Only Physics Equations
>
> **A2:** We appreciate this concern and have clarified the scope of our claims with explicit discussions on generalization boundaries in the revised manuscript.
>
> **Why Physics?** Physics provides the clearest testbed for scientific law discovery due to its well‑established ground truth and canonical equation structures. However, NewtonBench's findings generalizes to other subjects, such as chemistry and biology, because the underlying cognitive process—isolating variables to derive mathematical relationships—remains isomorphic across disciplines. Whether analyzing reaction kinetics or enzyme dynamics, the discovery process relies on the same rigorous interplay of hypothesis testing and symbolic abstraction. Accordingly, NewtonBench serves as a robust indicator of an agent’s scientific discovery potential across the natural sciences.
>
>
>
> **Standard Practice.** Numerous works [2][3] claim "scientific discovery" while evaluating on one or two specific domains, reflecting the field's expectation of generalizability and broader impact.
>
> We have updated the **Ethics Statement** section with expanded discussion on these generalization arguments and future extensions to other scientific domains.
>
>
> ---
>
>
>
> [1] LLM-SRBench: A New Benchmark for Scientific Equation Discovery with Large Language Models,  2025
> [2] Auto-Bench: An Automated Benchmark for Scientific Discovery in LLMs, 2025
> [3] Unlearning as Ablation: Toward a Falsifiable
> Benchmark for Generative Scientific Discovery, 2025
>
> ---
>
> We sincerely appreciate your valuable suggestions and thoughtful feedback. If you have any further questions or require additional clarification, we would be glad to provide it. We kindly ask you to consider adjusting your score if our revisions and clarifications adequately address your concerns.

---

> > ### Comment · Reviewer_cQE7 · 2025-11-23
> >
> > The rebuttal has addressed my concerns, and I am happy to maintain my original positive assessment.

---

### Official Review · Reviewer_KvtF · 2025-11-03

**Soundness:** 3
**Presentation:** 3
**Contribution:** 3
**Rating:** 6
**Confidence:** 3

**Summary:**

The paper introduces NEWTONBENCH, a benchmark to evaluate whether LLM agents can rediscover scientific laws through interactive experimentation rather than static curve fitting. Tasks are formalized via hidden target equations embedded in model systems, with agents probing a virtual environment and optionally using a code interpreter to hypothesize laws. The benchmark spans 324 tasks from 12 physics domains, created by applying metaphysical shifts to canonical laws to ensure novelty and prevent memorization. Performance is measured by Symbolic Accuracy (structure equivalence) and RMSLE (data fidelity). Experiments on 11 LLMs show frontier models succeed on simple systems but degrade with system and equation complexity and under noise.

**Strengths:**

* The benchmark is well-motivated and carefully constructed, with clear task formalization, principled law mutations, and dual metrics. The interactive setup and solvability proof are strong points. The experiments are broad (11 models) and include informative analyses (noise robustness, code-assistance effects), with consistent reporting (four runs, CIs).
* Task specification is clear and formal. The benchmark formalizes Equation and Model with expression-tree semantics and sequential model composition, clarifying what is hidden vs. provided to the agent. Three model settings (Vanilla/Simple/Complex) explicitly encode difficulty via system composition, aligning evaluation with progressive scientific scenarios. This makes difficulty interpretable.
* The agentic evaluation protocol is principled and interactive. Agents must conduct experiments via a standardized ‘<run_experiment>’ interface rather than passively fit given datasets. This better reflects real discovery workflows.
* Experimental suite is comprehensive. 11 diverse LLMs are compared with consistent settings, including “reasoning vs. non-reasoning” categorization and code-use variants. Main results aggregate across difficulty and system complexity with mean and std and additional analyses for noise and code assistance.

**Weaknesses:**

* The paper excludes classical symbolic-regression systems and prior LLM-SR pipelines from evaluation because they do not support the interactive model-system protocol (Ethics), which weakens external validity claims.  Although “Vanilla Equation” tasks exist (Sec. 2), there is no reported attempt to adapt strong SR baselines in this subset to provide a bridge comparison. No direct evidence found in the manuscript. The conclusions about “genuine scientific intelligence” would be stronger if positioned against competitive non-agentic baselines on overlapping subproblems.
* The method relies on an LLM as a judge for symbolic equivalence. While the authors report 98.3% agreement with humans, details like adjudication model identity and calibration procedures are in the appendix without ablations in the main results. Potential bias arises if the judge shares training or inductive biases with evaluated models; the paper does not present stress tests (adversarial equivalences) for the judge. No direct evidence found in the manuscript.
* I am also concerned on the physical plausibility and task realism. The paper acknowledges that some mutated laws, while dimensionally coherent, may be physically implausible in our universe. This could reduce ecological validity for certain claims.There is no user study or expert rating on realism/interpretability of discovered laws beyond dimensional checks. No direct evidence found in the manuscript. The strong claim of resolving the “trilemma” would benefit from quantitative measures of “scientific relevance” beyond coverage of canonical domains.

**Questions:**

* On Vanilla Equation tasks, will you adapt strong symbolic-regression (SR) methods to your submission format so they can output discovered forms and constants, and report SA/RMSLE side‑by‑side with your agent?
* Where exact protocol parity is impossible, can you include partial evaluations (e.g., passive dataset variants) to triangulate your difficulty claims?
* In Sec. 4, can you explicitly separate results that require “interactive discovery” from those a non‑agentic baseline could achieve on overlapping subproblems, and provide a table that maps each claim to the relevant baseline category?

---

> ### Author Response · Authors · 2025-11-22
>
> Dear Reviewer KvtF,
>
> Thanks for your thoughtful review and acknowledgement in:
> - Overall quality of NewtonBench.
> - Clear task specification and setting.
> - Interactive and realistic agentic protocol.
> - Comprehensive experiments and analyses.
>
> We hope the following revisions and clarifications address your concerns!
>
> **[Revision Summary]** In short, (1) we further evaluate and analyze two representative SR baselines (PySR and LLM-SR). (2) we further analyzed the robustness of LLM-as-a-Judge. All revisions are updated in the manuscript.
>
> ---
>
> > Q1 (and questions as well): The paper excludes classical symbolic-regression systems and prior LLM-SR pipelines from evaluation because they do not support the interactive model-system protocol (Ethics), which weakens external validity claims.
>
> **A1:** Thanks for your suggestions. Including prior SR approaches on the compatible subset of NewtonBench is indeed an important addition. To this end, we conducted experiments on two established SR approaches: **PySR** and **LLM-SR**, evaluating them on the vanilla equation tasks of NewtonBench.
>
> **Key Findings:**
> - Benefiting from iterative regression and optimization, both PySR and LLM-SR achieve low RMSLE comparable to our best LLM agents (though consuming 1000× more compute/API calls).
> - However, symbolic accuracy for both approaches remains marginally lower than our LLM agents (with code assistance, for fair comparison).
> - SR methods show high sensitivity to equation difficulty.
>
> Here's an intuitive result snippet:
> | Method                              | LLM Backbone   | SA easy | SA medium | SA hard | SA Avg. | RMSLE        |
> |---|--|---|-----------|---------|---------|--------------|
> | PySR                                | –              | 83.3    | 25.0      | 0.0     | 36.1    | 2.51 × 10⁻⁴  |
> | LLM‑SR*                             | GPT‑4.1‑mini   | 80.0    | 66.7      | 20.0    | 55.6    | **3.89 × 10⁻⁵** |
> | Vanilla Agent (newtonbench)         | GPT‑4.1‑mini   | 41.7    | 0.0       | 0.0     | 13.9    | 2.27         |
> | Agent w/ Code Asst. (newtonbench)   | GPT‑4.1‑mini   | **100.0** | **100.0** | **25.0** | **75.0** | 4.61 × 10⁻²  |
>
> \*LLM-SR uses relaxed judgement in SA due to its noisy output structure.
>
>
> Detailed results and analyses are presented in **Appendix E5**.
>
> ---
>
> >Q2: The method relies on an LLM as a judge for symbolic equivalence... Potential bias arises if the judge shares training or inductive biases with evaluated models; the paper does not present stress tests (adversarial equivalences) for the judge. No direct evidence found in the manuscript.
>
> **A2:** This concern is valid. We conducted several experiments and analyses on LLM-as-a-Judge for further validation:
> - **Style Robustness Analysis** (Is the LLM judge robust across different law styles?) Answer: Yes. The LLM judge is highly robust to syntactic variations.
> - **Impact of Few-shot Learning** (Does performance improve with more demonstrations?) Answer: Yes. More demonstrations yield better performance.
> - **Error Analysis** (Representative cases of false positives/negatives.) Findings: Errors are fairly random (even with temperature=0). Simply using self-consistency or bootstrapping could resolve these issues. However, current performance (98.3%) is already sufficient.
>
>
> Detailed results and analyses are presented in **Appendix A5**.
>
>
> > Q3: The paper acknowledges that some mutated laws, while dimensionally coherent, may be physically implausible in our universe. This could reduce ecological validity for certain claims. The strong claim of resolving the “trilemma” would benefit from quantitative measures of “scientific relevance” beyond coverage of canonical domains.
>
> **A3:** We understand your concern. We believe that "scientific relevance" does not necessarily require the target law to "exist" in our universe (we removed the word "plausible" to avoid confusion).
>
> Compared with existing work on generalizable scientific law discovery (e.g., LSR‑Synth), NewtonBench offers enhanced scientific relevance in two key respects:
> - It **emphasizes the scientific method**—hypothesis formation, experimental design, interventional reasoning, and synthesis—rather than mere function fitting.
> - It is **grounded in canonical quantities and constraints**. Tasks employ real physical variables (mass, charge, temperature, area, distance), enforce dimensional consistency and domain admissibility, and apply real‑world assisting laws to structure interventions. Although the target law is modified, LLM agents must still reason about the same types of variables, units, and confounders that appear in physics laboratories, rather than arbitrary symbols.
>
> ---
>
> We sincerely appreciate your valuable suggestions and thoughtful feedback. If you have any further questions or require additional clarification, we would be glad to provide it. We kindly ask you to consider adjusting your score if our revisions and clarifications adequately address your concerns.

---

> ### Author Response · Authors · 2025-11-26
> **Follow-up by Authors**
>
> Dear Reviewer KvtF,
>
> We are happy to follow-up on our new revision that further addresses your concern regarding scientific relevance and task realism.
>
> ---
> **[Revision]:  Fine-grained Categorization Taxonomy on Physical Plausibility of Shifted Laws**
>
> We agree that further analysis / evaluation on the realism or plausibility of the updated physical laws substantially enhance the physical interpretation of NewtonBench. To this end, we formally introduced a **three-level taxonomy to categorize all 108 shifted physical laws into three plausibility levels**.
>
> We measure "physical plausibility" with the compatibility of the shifted law with real-world physics. A law exhibits lower physical plausibility if it violates more fundamental physical principles in the real world. The three plausibility levels are defined as follows:
>
> - **Level 1 (no structural violation)**: Level 1 consists of admissible counterfactuals that modify the functional form while strictly preserving the original law's finiteness, real-valued outputs, and fundamental symmetries like action-reaction.
> - **Level 2 (minor structural violation)**: Level 2 includes laws that maintain mathematical regularity and finiteness but diverge from standard physical theory by altering essential symmetries or reciprocity relations required for conservation.
> - **Level 3 (major structural violation)**: Level 3 characterizes laws with significant structural anomalies, such as the introduction of singularities or physically invalid value ranges. These deviations challenge the minimal regularity and positivity constraints typically required in real-world physical modeling. However, because dimensional homogeneity is preserved, these candidates remain eligible for scientific law discovery benchmarking.
>
> The plausibility classification **serves as a practical guideline for future applications of NewtonBench**. It allows researchers to select specific subsets for objectives extending beyond standard scientific law discovery benchmarking. For instance, Level 1 provides an ideal environment for scientific agent training or generalized physical reasoning (where tolerance for structural violations is low), while Level 3 is well-suited for stress-testing symbolic regression models or exploring counterfactual reasoning.
>
> Here's some statistics of the categorization result:
> | Plausibility level \(from high to low\)  | Easy | Medium | Hard | Total |
> |----|:----:|:------:|:----:|:-----:|
> | Level 1  | 30   | 23     | 17   | 70    |
> | Level 2  | 3    | 5      | 6    | 14    |
> | Level 3 | 3    | 8      | 13   | 24    |
> | **Total** | 36   | 36     | 36   | 108   |
>
> Detailed definitions, discussions, implications, and classification results are provided in **Appendix A.3.4**.
>
> ---
> We sincerely appreciate your valuable suggestions and thoughtful feedback. If you have any further questions or require additional clarification, we would be glad to provide it. We kindly ask you to consider adjusting your score if our revisions and clarifications adequately address your concerns.

---

### Author Response · Authors · 2025-11-22
**General Response to All Reviewers**

# General Response to All Reviewers

We sincerely thank all reviewers for their thoughtful and constructive feedback. We are encouraged by the broad recognition of NewtonBench's contributions, including:

- **Originality and novelty** of the benchmark design and interactive agentic protocol
- **Strong experimental rigor** with comprehensive evaluations and analyses
- **Clear presentation** of task formulation, methodology, and results
- **Significance and impact** for both AI and scientific discovery research

We have carefully addressed all concerns raised and made substantial revisions to the manuscript. Below we summarize the major changes:

---

## Summary of Revisions

### 1. Evaluation of Classical SR Baselines (→ Reviewer KvtF)

We evaluated **PySR** and **LLM-SR** on vanilla equation tasks to strengthen external validity. Both achieve low RMSLE comparable to our best agents (with 1000× more compute) but lower symbolic accuracy. Results show high sensitivity to equation difficulty.

| Method | SA (easy/medium/hard) | SA Avg. | RMSLE |
|--|-|-|-|
| PySR | 83.3 / 25.0 / 0.0 | 36.1 | 2.51×10⁻⁴ |
| LLM-SR* | 80.0 / 66.7 / 20.0 | 55.6 | **3.89×10⁻⁵** |
| Agent w/ Code | 100.0 / 100.0 / 25.0 | **75.0** | 4.61×10⁻² |a

See **Appendix E5** for details.

---

### 2. Robustness Analysis of LLM-as-a-Judge (→ Reviewers KvtF, cQE7)

We conducted comprehensive validation experiments on our symbolic accuracy metric:

- **Style Robustness:** The LLM judge is highly robust to syntactic variations across different equation representations
- **Few-shot Impact:** Performance improves with more demonstrations, achieving 98.3% accuracy
- **Error Analysis:** Errors are fairly random (even at temperature=0); self-consistency or bootstrapping could further improve reliability

We argue that LLM-as-a-Judge follows established practice in the literature and all frontier LLMs should be capable of performing such judgments. Current performance (98.3%) is already sufficient for reliable evaluation.

See **Appendix A5** for details.

---

### 3. DataBlind Experiment for Memorization-Free Validation (→ Reviewer T5rR)

DataBlind experiments confirm NewtonBench is fully memorization-free via counterfactual shifts:

| Model | LSR-Transform | NewtonBench (no hint) | NewtonBench (hint) |
|-|-|-|-|
| GPT-4o-mini | 7.21 | 0.0 | 0.0 |
| Nemotron-Ultra | 28.83 | 0.0 | 0.0 |
| GPT-4.1 | 35.14 | 0.0 | 0.0 |

See **Appendix E.1** for details.

---

### 4. Clarifications on Scientific Relevance and Scope (→ Reviewers KvtF, cQE7, 9pai)

**Scientific Relevance:** NewtonBench emphasizes the **scientific method** (hypothesis testing, experimental design, interventional reasoning) and is **grounded in canonical physical quantities** (dimensional consistency, real variables). Shifted constants don't diminish relevance—discovery relies on reasoning structure, not numerical values.

**Generalization Scope:** While focused on physics, the cognitive process (isolating variables, deriving relationships) generalizes across natural sciences, following standard practice in scientific discovery works.

See **Section 3.1** and **Ethics Statement** for expanded discussions.

---

### 5. System-Level Complexity Documentation (→ Reviewer T5rR)

We have clarified the curation logic and rationale for system-level complexity tiers:

- Systems are curated by domain experts with explicit paired instructions
- Complexity is measured by system properties (# assisting equations, # operations), not reasoning paths
- Empirical validation: Table 2 shows consistent performance degradation from Vanilla → Simple → Complex

|| Simple Sys. | Complex Sys. |
|-|-|--|
| Avg. # Assisting Eqs. | 1.75 | 2.42 |
| Avg. # Operations | 5.17 | 8.17 |

Details incorporated in **Appendix A2**.

---

### 6. Improved Clarity and Presentation (→ Reviewer T5rR, 9pai)

Removed excessive philosophical discussions, adopted "counterfactual shifts" terminology, and focused on concrete technical objectives.

---

## Closing Remarks

We believe these revisions and clarifications substantially strengthen the manuscript and address all major concerns raised by the reviewers. The additional experiments on classical SR baselines, comprehensive LLM-as-a-Judge validation, and DataBlind memorization tests provide strong empirical support for NewtonBench's design choices and evaluation methodology.

We respectfully ask all reviewers to reconsider their scores in light of these revisions. We remain happy to provide any additional clarification or conduct further experiments if needed.

Thank you again for your valuable feedback and for helping us improve the quality of this work.

ICLR 2026 Conference Submission7266 Authors

---

### Author Response · Authors · 2025-12-03
**Final Comments by Authors**

# Final Comments by Authors

We sincerely thank all reviewers for their thoughtful and constructive feedback throughout this rebuttal period. We are encouraged by the broad recognition of NewtonBench's contributions, specifically highlighting:

- **Originality and Novelty:** The unique benchmark design and interactive agentic protocol. (Reviewers KvtF, cQE7, 9pai, T5rR)
- **Experimental Rigor:** The comprehensive evaluations and detailed analyses. (Reviewers KvtF, cQE7, 9pai, T5rR)
- **Presentation Quality:** The clarity of the task formulation, methodology, and results. (Reviewers KvtF, cQE7, T5rR)
- **Significance and Impact:** The potential value for both AI research and scientific discovery. (Reviewers KvtF, cQE7, 9pai, T5rR)

During the rebuttal period, we substantially enhanced our manuscript based on reviewer comments. While the **core design and contributions remain unchanged**, we incorporated extensive auxiliary experiments, analyses, and clarifications for the highest soundness and scientific rigor. Consequently, our manuscript has **expanded from 60 to 70 pages** to accommodate these improvements.

Regarding the reviewers' final responses, we are delighted to **receive a strong recommendation from Reviewer T5rR, who raised their score from 6 to 8 following multiple rounds of discussion**. Furthermore, **Reviewer cQE7 confirmed that all their concerns were well addressed** and maintained their recommendation for acceptance. While the other two reviewers didn't yet respond to our latest follow-ups, we remain **confident that all raised concerns have been thoroughly addressed** in our revision.

---

## Summary of Revisions

We **fully addressed all reviewer concerns** with revisions and clarifications as follows.

 Note that most concerns pertained to further details rather than the main body of the work (core methodology and conclusions). Consequently, most revisions have been placed in the appendix. We'll selectively integrate into the main text in camera-ready.

---

### 1. Core Revisions and Improvements

**C1.** NewtonBench's practical implication into real-world physics/science can be further improved. (Reviewer KvtF, 9pai, T5rR)

**[Revision 1]:** Introduced a fine-grained classification taxonomy "physical plausibility levels" on our curated laws to better control and measure real-world physical relevance. (**App. A.3.4**)

**[Revision 2]:** Formally discussed NewtonBench's scientific interpretation and highlighted future benchmarking directions. (**Ethics Stat.** & **App. E.6**)

---

**C2.** Baselines from symbolic regression methods can be included for comprehensiveness. (Reviewer KvtF)

**[Revision 3]:** Thoroughly evaluated and analyzed PySR and LLM-SR. (**App. E.5**)

---
**C3.** Justification and analysis for our LLM-as-a-Judge approach. (Reviewer KvtF, cQE7)

**[Revision 4]:** Include analysis on style robustness. (**App. A.5**)

**[Revision 5]:** Include analysis on impact few-shot learning. (**App. A.5**)

**[Revision 6]:** Include analysis on error cases. (**App. A.5**)

Overall, the robustness of our LLM Judge is well-justified.

---

### 2. Miscellenous Revisions and Clarifications

---
**M1.** Reviewer T5rR suggested that the introduction of philosophical concepts may overwhelm readers.

**[Revision 7]:** Removed unnecessary philosophical discussions (**Sec. 3.1**).

---
**M2.** Reviewer T5rR questioned whether NewtonBench is truly memorization-free.

**[Revision 8]:** We conduct further DataBlind experiments (**App. E.1**), proving that NewtonBench is **fully memorization-free**.

---

**M3.** Reviewer 9pai identified an inconsistency in the formulation of Fourier's Law with nullification test.

**[Revision 9]:** While this test is not crucial given Newtonbench's counterfactual nature, we have **revised all curated laws and redesigned a small subset with similar inconsistencies** to ensure rigorous alignment with real-world physical principles. (**App. A.3.3**)

**All experiments and analyses have been re-implemented and updated in the manuscript, and no major conclusions and findings are affected**.

---

**M4.** Reviewer 9pai's main argument: "physical constant's unit is deterministic to physical laws; unit change in physical constants may diminish scientific relevance".

**[Response]:** This argument is **incorrect**. Real-world discovery involves identifying structural relationships between variables first, then **retrospectively defining physical constants' values and units to satisfy dimensional homogeneity** (per the Buckingham $\pi$ theorem). This epistemological order **aligns perfectly with NewtonBench's core design**. (detailed justifications in **Author Follow-up A5**)

---

**M5.** Reviewer 9pai questioned whether the proposed model discovery paradigm is meaningfully distinct from equation discovery.

**[Response]:** The significant distinction is **both conceptual and empirically validated**. We provided detailed justifications in **Author Follow-up A6**.

---

### Meta-Review · Area_Chair_9zUR · 2026-01-08

**Summary:**

This paper presents  NewtonBench, a benchmark for evaluating LLMs' scientific law discovery capabilities that addresses fundamental limitations in existing approaches. Most reviewers responded positively, although one raised concerns regarding the motivation for creating such a benchmark. The area chair finds the work interesting and has therefore decided to recommend its acceptance.

**Reviewer Concerns:**

Motivation. The author's response was reasonable during the rebuttal.

**Reviewer Scores:**

NA

---

### Decision · Program_Chairs · 2026-01-26

Accept (Poster)